# Kernel-Based Robust Markov Subsampling for Regularized Nonparametric Regression with Contaminated Data

## Abstract

Large-scale data with contamination are ubiquitous in biomedicine, economics and social science, but its statistical learning often suffers from computational bottlenecks and robustness. Subsampling offers an efficient solution by sampling a representative subset of uncorrupted data from full dataset, thereby reducing computational costs while enhancing robustness. Existing subsampling methods, like leverage- and gradient-based approaches, focus on parametric models and fail under nonparametric models or severe contamination. To address these limitations, we propose a kernel-based robust Markov subsampling (KRMS) method for nonparametric regression with contaminated data in reproducing kernel Hilbert space (RKHS). By dynamically adjusting Markov sampling probabilities based on the ratio of residuals to kernel norms of predictors, our method simultaneously suppresses contaminated observations and prioritizes informative observations, enabling robust learning from contaminated datasets. Theoretically, we establish the asymptotic properties of the estimators, including consistency and asymptotic normality, and generalization bounds under RKHS regularization, providing the first unified framework for robust subsampling in nonparametric settings. Simulations and real-data applications demonstrate KRMS's superiority over existing methods, particularly for high contamination levels. Our approach bridges a critical gap in scalable and robust statistical learning, with broad applicability to large-scale, non-i.i.d. data.

## 1 Introduction

The rapid development of data collection technologies has ushered in an era of unprecedented data proliferation across nearly all scientific and industrial fields. Data in fields ranging from biomedical imaging and financial risk analysis to environmental monitoring and social network analytics exhibit not only massive scale but also increasingly frequent contamination, including outliers, measurement errors, and systematic biases (Fan et al., 2014). While this data deluge offers unprecedented opportunities for scientific discovery and practical applications, it simultaneously faces critical limitations in conventional statistical learning methods, particularly their inability to scale computationally with massive datasets and their vulnerability to pervasive data contamination. Traditional statistical learning approaches, developed for uncontaminated, or smaller-scale data, frequently fail when applied to the complicated or contaminated data, where contamination is not merely an occasional nuisance but an inherent characteristic. This dual challenge of computational scalability and statistical learning robustness has emerged as a fundamental bottleneck in the era of big data.

To mitigate these challenges posed by massive datasets, subsampling has emerged as a widely used strategy. Specifically, by selecting a representative subset of uncontaminated data from the full dataset with contamination, this subsampling method possesses dual merits: substantial computational efficiency and potentially improving parameter estimation accuracy. However, the effectiveness of subsampling hinges critically on its ability to preserve statistical properties of the full dataset, a non-trivial challenge in practice. Existing solutions to this challenge can be classified as three categories: (i) optimal design-based approaches (Ai et al., 2021; Wang & Ma, 2021) that minimize asymptotic variance of parameter estimator for uncontaminated data; (ii) informative subsampling techniques for the massive data without contamination or with relatively low level of contamination,

including leverage-based subsampling (Ma et al., 2015; Rudi et al., 2018), gradient-based subsampling (Zhu, 2016) and influence function-based subsampling (Ting & Brochu, 2018), however, these methods typically produce biased estimators when applied to highly contaminated data; (iii) robust subsampling methods for corrupted massive data based on the idea of quantile breakdown point for linear regression models (Camponovo et al., 2012), robust gradient-based Markov subsampling (Gong et al., 2020), and low-gradient subsampling (Jing, 2023). Notably, Markov subsampling has shown particular promise by adaptively refining parameter estimate through sequential transitions, and self-correcting for contamination via Metropolis-Hastings (MH) rejection scheme while preserving structural information in the dataset. But existing Markov subsampling methods are fundamentally constrained to parametric models with contaminated data, leaving it ill-equipped for nonparametric regression problems where the target is an infinite-dimensional function rather than finite-dimensional parameters. Three key challenges are encountered in nonparametric regression models with contaminated data. First, existing subsampling methods fail to properly weight observations in RKHS, where contamination distorts both local smoothness and global structure. Second, the "curse of dimensionality" exacerbates contamination effects in high-dimensional function estimation. Third, non-i.i.d. data dependencies, such as those in Markov chains, interact with contamination in ways that linear regression models cannot capture. These limitations become particularly severe under heavy contamination scenarios like Huber's $\varepsilon$-model, where existing subsamplers fail to retain the essential topological properties of the target function.

To overcome these challenges, we propose a Kernel-based robust Markov Subsampling (KRMS), which introduces several key innovations. By mapping the original data to an RKHS, our contamination scoring mechanism combines residual with features similarity to identify contaminated observations through relative data structure rather than absolute values. This kernel-based approach enables effective separation of contaminated observations that would be indistinguishable in the original feature space. The KRMS framework incorporates these scores into a MH sampling process that naturally accommodates non-i.i.d. data dependencies while maintaining computational efficiency. Theoretically, we establish consistency of nonparametric function estimator under mild regularity conditions, while demonstrating robustness to both contamination and high dimensionality. Our approach thus solves what existing methods cannot: simultaneous robustness to severe contamination, computational scalability, and theoretical soundness for nonparametric regression with complex dependencies.

Our work has three key contributions to nonparametric regression with contaminated data. First, we propose a KRMS method in RKHS by dynamically adjusting Markov subsampling probabilities based on the ratio of residuals to kernel norms of predictors, which is the first subsampling method specifically designed for contaminated data in complex nonparametric regression settings. Unlike existing approaches limited to parametric models, KRMS adapts to the intrinsic geometry of function spaces through kernel learning. Second, within the framework of kernel regularized regression with symmetric periodic Gaussian kernels in Sobolev spaces (Zeng & Xia, 2019), we establish rigorous theoretical guarantees for the KRMS estimator of nonparametric function. Based on assumption that the data follow a uniformly ergodic Markov chain (u.e.M.c.), we obtain optimal consistency rates and asymptotic normality of KRMS estimator, and explicit error bound of excess risk under contamination. Third, we extend the theoretical framework to characterize the generalization performance of kernel regularized regression in RKHS, providing new insights into the interaction between subsampling robustness and function space geometry.

The rest of this paper is organized as follows. Section 2 introduces regularized nonparametric regression model and RKHS. Section 3 details the proposed method. Section 4 presents asymptotic properties and generalization bounds for kernel-based regularization regression under Huber $\varepsilon$-contamination for u.e.M.c. samples. Simulation studies are conducted in Section 5. Concluding remarks are given in Section 6. The proofs of theorems, additional simulations and real examples analysis are presented in the Appendices C and D. The convergence analysis and parameter sensitivity analysis are presented in the Appendix E.

## 2 PRELIMINARIES

### 2.1 REGULARIZED NONPARAMETRIC REGRESSION MODEL

Consider learning a continuous function $f(\mathbf{x}) \in \mathcal{H}(\mathbb{X})$ from a dataset $\mathcal{D} = \{z_i = (\mathbf{x}_i, y_i) : i = 1, \ldots, n\}$, where $\mathbf{x}_i = (x_{i1}, \ldots, x_{ip})^\top \in \mathbb{X}$ is the input vector of the $i$-th individual, $y_i \in \mathcal{Y}$ is the corresponding observed output, $\mathbb{X}$ is a compact subset of $\mathbb{R}^p$ ($p \geq 2$), and $\mathcal{H} = \mathcal{H}(\mathbb{X})$ is a space of continuous functions. The relationship between $\mathbf{x}_i$ and $y_i$ is modeled as $y_i = f_0(\mathbf{x}_i) + \epsilon_i$, where $f_0(\mathbf{x}_i) : \mathbb{R}^p \to \mathbb{R}$ is an unknown target function, and the random noise $\epsilon_i$ satisfies $\mathbb{E}(\epsilon_i) = 0$ and $\mathbb{E}(\epsilon_i^2) = \sigma^2$, and is independent of $\mathbf{x}_i$ for $i = 1, \ldots, n$. The goal is to find a function $f(\mathbf{x}) : \mathbb{X} \to \mathcal{Y}$ that approximates $f_0$ well by minimizing the generalization risk: $\mathcal{R}_\mathcal{F}(f) = \mathbb{E}\{\ell(f(\mathbf{x}), y)\} = \int_\mathcal{Z} \ell(f(\mathbf{x}), y) d\mathcal{F}$, where $\ell(f(\mathbf{x}), y)$ is a nonnegative loss function measuring the fitting error when using $f(\mathbf{x})$ to fit the output $y$, $\mathcal{Z} = \mathbb{X} \times \mathcal{Y}$ represents the sample space, $\mathcal{F}$ is an unknown joint distribution of $z = (\mathbf{x}, y) \in \mathcal{Z}$, $\mathbb{E}(\cdot)$ is the expectation taken with respect to distribution function $\mathcal{F}$. It is difficult to directly compute minimizer of $\mathcal{R}_\mathcal{F}(f)$ due to unknown distribution $\mathcal{F}$ involved. To solve the difficulty, we instead minimize the empirical risk (ER) over a function space $\mathcal{H}$: $\mathcal{R}_\mathcal{D}(f) = (2n)^{-1} \sum_{i=1}^n \ell(f(\mathbf{x}_i), y_i)$. Throughout this paper, we consider the following squared-error loss: $\ell(f(\mathbf{x}_i), y_i) = \{y_i - f(\mathbf{x}_i)\}^2$. Thus, for the considered squared-error loss, the ER minimizer is

$$f_\mathcal{D} = \arg\min_{f \in \mathcal{H}} \mathcal{R}_\mathcal{D}(f) = \arg\min_{f \in \mathcal{H}} \frac{1}{2n} \sum_{i=1}^n \{y_i - f(\mathbf{x}_i)\}^2, \tag{1}$$

which is an approximation of function $f_0(\mathbf{x})$. However, when $\mathcal{H}$ is highly complex, the optimization problem (1) becomes ill-posed and prone to overfitting (Zou et al., 2014). To address this issue, we restrict the function space $\mathcal{H}$ to a RKHS and solve the following regularized optimization problem:

$$f_{\mathcal{D},\lambda} = \arg\min_{f \in \mathcal{H}} \{\mathcal{R}_\mathcal{D}(f) + \lambda J(f)\}, \tag{2}$$

where $J(f) : \mathcal{H} \to \mathbb{R}_+$ is a penalty functional with $J(0) = 0$ that controls complexity of $f$, and $\lambda > 0$ is an appropriate regularization parameter depending on the sample size $n$ such that $\lambda = \lambda(n)$ and $\lim_{n \to \infty} \lambda(n) = 0$ as $n \to \infty$. For any estimator $f_{\mathcal{D},\lambda}$ of function $f_0(\mathbf{x})$, its quality is measured by its excess risk (i.e., the difference between the $L_2$ expected risks of $f_{\mathcal{D},\lambda}$ and $f_0$): $\|f_{\mathcal{D},\lambda} - f_0\|_{L^2_{\mathcal{F}_\mathbb{X}}}^2 = \mathcal{R}_\mathcal{F}(f_{\mathcal{D},\lambda}) - \mathcal{R}_\mathcal{F}(f_0)$, where $\mathcal{F}_\mathbb{X}$ is the marginal distribution of $\mathcal{F}$ on $\mathbb{X}$, and $L^2_{\mathcal{F}_\mathbb{X}}$ denotes the space of square-integrable functions with respect to the measure $\mathcal{F}_\mathbb{X}$.

### 2.2 REPRODUCING KERNEL HILBERT SPACE

Following Aronszajn (1950), an RKHS $\mathcal{H}$ is a Hilbert space of functions where all evaluation functionals are continuous and bounded. To wit, for any $f(\mathbf{x}) \in \mathcal{H}$ and $\mathbf{x} \in \mathbb{X}$, there exists a positive constant $C$ such that $\mathcal{L}_\mathbf{x}(f) = |f(\mathbf{x})| \leq C\|f\|_\mathcal{H}$, where $\mathcal{L}_\mathbf{x}$ is the evaluation functional at observation $\mathbf{x}$, and $\|\cdot\|_\mathcal{H}$ is the norm on $\mathcal{H}$. A function $K(\cdot, \cdot) : \mathbb{X} \times \mathbb{X} \to \mathbb{R}$ is called a reproducing kernel (RK) if it is symmetric and positive definite: $\sum_{i=1}^n \sum_{j=1}^n a_i a_j K(\mathbf{x}_i, \mathbf{x}_j) \geq 0$ for any $\mathbf{x}_1, \ldots, \mathbf{x}_n \in \mathbb{X}$ and $a_1, \ldots, a_n \in \mathbb{R}$. By the Moore-Aronszajn Theorem (Aronszajn, 1950), every symmetric positive definite function $K(\cdot, \cdot)$ uniquely defines an RKHS $\mathcal{H}_K$ of real-valued functions. Specifically, $\mathcal{H}$ is the closure of the linear span of kernel functions:

$$\mathcal{H}_K = \left\{ f(\cdot) = \sum_{i=1}^n \alpha_i K(\mathbf{x}_i, \cdot) : \mathbf{x}_i \in \mathbb{X}, \alpha_i \in \mathbb{R} \right\},$$

and the corresponding inner product is defined as $\langle K(\mathbf{x}_i, \cdot), K(\mathbf{x}_j, \cdot) \rangle_{\mathcal{H}_K} = K(\mathbf{x}_i, \mathbf{x}_j)$.

## 3 METHODOLOGIES

The optimization problem in Equation (2) yields an efficient estimator of the target function $f_0$ when the dataset $\mathcal{D}$ is sampled independently from the true distribution $\mathcal{F}$. However, in many applications, $\mathcal{D}$ often contains contaminated observations due to outliers or adversarial contamination. In such cases, $\mathcal{D}$ is instead generated from Huber's contamination model (Huber, 1992): $\mathcal{P} = (1-\theta)\mathcal{F} + \theta\mathcal{Q}$,

where $\mathcal{F}$ is the true (uncontaminated) distribution, $\mathcal{Q}$ is an arbitrary contaminating distribution, and $\theta \in [0, 1/2)$ controls the contamination level. This model captures scenarios where a fraction $\theta$ of the data may be arbitrarily contaminated, while the majority $1 - \theta$ follows the true distribution $\mathcal{F}$.

It is well established that estimators obtained from contaminated datasets can exhibit significant bias and provide poor approximations of the target function $f_0$. To address this challenge, a natural strategy involves identifying and sampling uncontaminated observations from the contaminated dataset $\widetilde{\mathcal{D}} = \{\widetilde{z}_i = (\widetilde{\mathbf{x}}_i, \widetilde{y}_i)\}_{i=1}^n$ to obtain an optimal solution of $f$ for Equation (2). Therefore, our objective is to develop an effective subsampling method that is capable of robustly selecting a representative subset of uncontaminated observations, even in the presence of severe contamination. Unlike conventional linear regression models, we consider a more general setting where contaminated observations reside within a RKHS framework. This approach leverages kernel methods to map the contaminated data into high-dimensional or infinite-dimensional feature spaces, where contaminated observations, which are difficult to distinguish in the original input space, becomes more separable. Building on this insight, we propose a novel kernel-based robust subsampling method for nonparametric models. A key advantage of our approach is its reliance on the relative distance between data points in the kernel-induced feature space, rather than the absolute magnitude-based criteria typically used in linear regression models. This property enables more reliable identification of contamination, particularly in complex and nonlinear settings.

For an uncontaminated dataset $\mathcal{D}$, the squared-error loss in the RKHS $\mathcal{H}_K$ takes the form

$$\mathcal{R}_{\mathcal{D}}(f) = \frac{1}{2n} \sum_{i=1}^n \left\{ y_i - \sum_{j=1}^n \alpha_j K(\mathbf{x}_j, \mathbf{x}_i) \right\}^2.$$

The regularized estimator $f_{\mathcal{D},\lambda} = \arg\min_{f \in \mathcal{H}} \{\mathcal{R}_{\mathcal{D}}(f) + \lambda J(f)\}$ provides an unbiased estimate of function $f_0$. When dealing with a contaminated dataset $\widetilde{\mathcal{D}}$, the squared-error loss becomes

$$\mathcal{R}_{\widetilde{\mathcal{D}}}(f) = \frac{1}{2n} \sum_{i=1}^n \left\{ y_i - \sum_{j=1}^n \alpha_j K(\tilde{\mathbf{x}}_j, \tilde{\mathbf{x}}_i) \right\}^2,$$

where $\widetilde{\mathbf{x}}_i$ is contaminated input vector, and $\widetilde{y}_i$ is contaminated output. Here we consider Huber's contamination model for contaminated data mechanism, i.e., for input vector $\tilde{\mathbf{x}}_i$ and output $\tilde{y}_i$, a proportion $\theta$ of observations follows the arbitrary contaminating distribution $\mathbf{W}$ and $\mathbf{O}$, respectively. Under this mechanism, observations $(\widetilde{\mathbf{x}}_i, \widetilde{y}_i)$ are corrupted with probability $\theta$ and remain uncorrupted with probability $1 - \theta$. The specific forms of $\mathbf{W}$ (e.g., sparse noise, adversarial perturbations) and $\mathbf{O}$ (e.g., outliers, multiplicative errors) characterize the nature of the corruption. The corresponding estimator $f_{\widetilde{\mathcal{D}},\lambda} = \arg\min_{f \in \mathcal{H}} \{\mathcal{R}_{\widetilde{\mathcal{D}}}(f) + \lambda J(f)\}$ is biased when the contamination level $\theta$ is relatively large, and its computation becomes prohibitively expensive for a relatively large sample size $n$. To overcome these challenges, some robust subsampling methods like low-gradient subsampling (Jing, 2023), robust gradient-based Markov subsampling (Gong et al., 2020) and Markov subsampling based on Huber criterion Gong et al. (2022) have been proposed. However, these subsampling methods often yield unstable estimators due to sensitivity to unbalanced sampling probabilities, loss of important gradient information, poor performance with contaminated data. To this end, we propose a novel robust kernel-based Markov subsampling method that operates in the RKHS to better separate contaminated observations, uses modified gradient information for more reliable sampling, maintains computational efficiency while being robust to contamination. The method specifically addresses the limitations of existing subsampling approaches by carefully preserving the geometric structure of the uncontaminated data while downweighting the influence of contaminated observations in the kernel space.

The gradient of the empirical risk $\mathcal{R}_{\widetilde{\mathcal{D}}}(f)$ with respect to the coefficient vector $\alpha = (\alpha_1, \ldots, \alpha_n)^\top$ at the $i$-th observation $(\widetilde{\mathbf{x}}_i, \widetilde{y}_i)$ is

$$g_i(\alpha) = -\frac{1}{n} \left( \widetilde{y}_i - \sum_{j=1}^n \alpha_j K(\widetilde{\mathbf{x}}_j, \widetilde{\mathbf{x}}_i) \right) \begin{bmatrix} K(\widetilde{\mathbf{x}}_1, \widetilde{\mathbf{x}}_i) \\ K(\widetilde{\mathbf{x}}_2, \widetilde{\mathbf{x}}_i) \\ \vdots \\ K(\widetilde{\mathbf{x}}_n, \widetilde{\mathbf{x}}_i) \end{bmatrix}$$

whose norm is given by

$$\|g_i(\alpha)\| = \frac{1}{n} \left| \widetilde{y}_i - \sum_{j=1}^{n} \alpha_j K\left(\widetilde{\mathbf{x}}_j, \widetilde{\mathbf{x}}_i\right) \right| \sqrt{\sum_{j=1}^{n} K\left(\widetilde{\mathbf{x}}_j, \widetilde{\mathbf{x}}_i\right)^2},$$

which implies that the magnitude of $g_i(\alpha)$ depends on $\widetilde{e}_i = \left| \widetilde{y}_i - \sum_{j=1}^{n} \alpha_j K\left(\widetilde{\mathbf{x}}_j, \widetilde{\mathbf{x}}_i\right) \right|$ and $\widetilde{d}_i = \sqrt{\sum_{j=1}^{n} K\left(\widetilde{\mathbf{x}}_j, \widetilde{\mathbf{x}}_i\right)^2}$. The absolute value of prediction error $\widetilde{e}_i$ measures absolute deviation between observed and predicted responses, and large values of $\widetilde{e}_i$ indicate poor model fit, potentially signaling contamination. The quantity $\widetilde{d}_i$ depicts overall similarity of $\widetilde{\mathbf{x}}_i$ to other observations in the RKHS $\mathcal{H}_K$. Generally, large value of $\widetilde{d}_i$ indicates that $\widetilde{\mathbf{x}}_i$ is closely related to most of other observations (i.e., exhibiting high similarity), and small value of $\widetilde{d}_i$ implies that $\widetilde{\mathbf{x}}_i$ deviates considerably from the majority of the dataset $\widetilde{D}$ and can be regarded as a contaminated observation or outlier in feature space. Based on the preceding argument, we define the residual kernel-norm score as

$$w(\widetilde{z}_i, \alpha) = \frac{\left| \widetilde{y}_i - \sum_{j=1}^{n} \alpha_j K\left(\widetilde{\mathbf{x}}_j, \widetilde{\mathbf{x}}_i\right) \right|}{\sqrt{\sum_{j=1}^{n} K\left(\widetilde{\mathbf{x}}_j, \widetilde{\mathbf{x}}_i\right)^2}}. \tag{3}$$

A large value of $w(\widetilde{z}_i, \alpha)$ can be regarded as a strong indicator of contaminated observation or outlier. The score (3) normalizes residuals against the local geometry of $\mathcal{H}_K$, ensuring robust outlier detection regardless of the kernel structure. We can regard $w(\widetilde{z}_i, \alpha)$ as a modified version of the low-gradient subsampling. Similarly to low-gradient subsampling technique, we can utilize $w(\widetilde{z}_i, \alpha)$ to assign subsampling probabilities, i.e., $\pi_i \propto 1/w(\widetilde{z}_i, \alpha)$ is taken as the subsampling probability of observation $\widetilde{z}_i = (\widetilde{\mathbf{x}}_i, \widetilde{y}_i)$ in which the observations with smaller (larger) values of $w(\widetilde{z}_i, \alpha)$ are assigned larger (smaller) subsampling probabilities. This method is effective for moderate sample size $n$, but it faces high computational cost for large sample size $n$ (i.e., large-scale dataset), and sensitivity to highly contaminated data at small subsampling ratios (Gong et al., 2020). To overcome these problems, we develop a robust kernel-based Markov subsampling procedure by utilizing $w(\widetilde{z}_i, \alpha)$ to construct transition acceptance probabilities. This approach ensures that the generated subsamples are uniformly ergodic Markov chain (u.e.M.c.) samples, and maintains robustness while improving computational efficiency. The detailed procedure is summarized in Algorithm 1.

**Remark 1.** *(i) The probabilities $w(\widetilde{z}_i, \alpha)$ in (3) depend on the parameter vector $\alpha = (\alpha_1, \ldots, \alpha_n)^\top$. A key challenge in obtaining a high-performance estimator $\hat{f} = f_{\mathbb{S}, \lambda}$ of function $f_0$ is the need for a good initial estimate of $\alpha$, which is particularly difficult in heavily contaminated data settings. To address this issue, we employ a recursive updating approach, where $\alpha^{(\kappa)}$ is refined iteratively using the $(\kappa - 1)$-th subsample drawn from the contaminated dataset $\widetilde{\mathcal{D}}$ via the proposed subsampling, and used to recompute $w(\widetilde{z}_i, \alpha^{(\kappa-1)})$.*

*(ii) The choice of subsample size $n_0$ balances computational complexity of Algorithm 1 and estimation precision. It should be selected based on available computing resources and desired approximation accuracy.*

*(iii) Steps 6 and 9 of Algorithm 1 can be implemented using some standard subsampling technique, such as Poisson sampling or replacement sampling.*

*(iv) Parameter $\lambda$ can be determined via leave-one-out cross-validation (LOOCV) criterion to optimize model performance.*

*(v) The overall computational complexity of Algorithm 1 is $\mathcal{O}(T_0(nn_0 p + n_0^3))$. The term $nn_0 p$ arises from evaluating the residual kernel-norm scores across the full dataset, while $n_0^3$ corresponds to solving the sub-problem. Crucially, this linear dependence on $n$ represents a substantial improvement over the cubic $\mathcal{O}(n^2 p + n^3)$ complexity of standard kernel regression. Additionally, the space complexity is reduced to $\mathcal{O}(np + n_0^2)$, avoiding the $\mathcal{O}(np + n^2)$ storage required for the full Gram matrix. Thus, when $n_0 \ll n$, our method offers significant computational and spatial advantages.*

---

**Algorithm 1** Robust Kernel-based Markov Subsampling

---

1: **Initialization**: Contaminated data $\tilde{\mathcal{D}} = \{\tilde{z}_i = (\tilde{\mathbf{x}}_i, \tilde{y}_i)\}_{i=1}^{n}$, $\mathbb{S}_\kappa = \emptyset$, subsample size $n_0 < n$, burn-in period $t_0$, maximum number of iterations $T_0$, stopping criterion $\xi_0$ (e.g., 0.001).
2: **Output**: $\hat{f}$
3: Train a pilot estimate $\alpha^{(0)}$ for uniformly drawn observations $\{\tilde{z}_i\}_{i=1}^{n_0}$ from $\tilde{\mathcal{D}}$ via $\alpha^{(0)} = \arg\min_\alpha \sum_{i=1}^{n_0} \{\tilde{y}_i - \sum_{j=1}^{n_0} \alpha_j K(\tilde{\mathbf{x}}_j, \tilde{\mathbf{x}}_i)\}^2 + \lambda J(f)$, and set $\kappa = 1$
4: **while** $\kappa \leq T_0$ or $\|\alpha^{(\kappa)} - \alpha^{(\kappa-1)}\|_2 \geq \xi_0$ **do**
5:     Set $\alpha = \alpha^{(\kappa-1)}$
6:     Randomly draw an observation $\tilde{z}_1$ from $\tilde{\mathcal{D}}$, and compute $w(\tilde{z}_1, \alpha)$ via (3) and set $\mathbb{S}_\kappa = \mathbb{S}_\kappa \cup \tilde{z}_1$
7:     **for** $2 \leq t \leq n_0 + t_0$ **do**
8:         **while** $|\mathbb{S}_\kappa| < t$ **do**
9:             Randomly draw a candidate observation $\tilde{z}^*$ from $\tilde{\mathcal{D}}$ and compute $w(\tilde{z}^*, \alpha)$ via (3)
10:            Calculate acceptance probability: $\pi_\alpha = \min\{1, w(\tilde{z}_{t-1}, \alpha)/w(\tilde{z}^*, \alpha)\}$
11:            Set $\mathbb{S}_\kappa = \mathbb{S}_\kappa \cup \tilde{z}^*$ with probability $\pi_\alpha$
12:            If $\tilde{z}^*$ is accepted, set $\tilde{z}_t = \tilde{z}^*$
13:         **end while**
14:         Set $w(\tilde{z}_t, \alpha) = w(\tilde{z}^*, \alpha)$
15:     **end for**
16:     Denote the last $n_0$ observations of $\mathbb{S}_\kappa$ as $\{(\tilde{\mathbf{x}}_i^*, \tilde{y}_i^*)\}_{i=1}^{n_0}$
17:     Set $\alpha^{(\kappa)} = \arg\min_\alpha \sum_{i=1}^{n_0} \left\{\tilde{y}_i^* - \sum_{j=1}^{n_0} \alpha_j K(\tilde{\mathbf{x}}_j^*, \tilde{\mathbf{x}}_i^*)\right\}^2 + \lambda J(f)$
18:     Set $f_{\mathbb{S},\lambda}^{(\kappa)}(\tilde{\mathbf{x}}) = \sum_{j=1}^{n_0} \alpha_j^{(\kappa)} K(\tilde{\mathbf{x}}_j^*, \tilde{\mathbf{x}})$
19:     Update $\kappa = \kappa + 1$
20: **end while**
21: Return $\hat{f} = f_{\mathbb{S},\lambda}^{(\kappa+1)}$

---

# 4 THEORETICAL RESULTS

## 4.1 VALIDITY OF SUBSAMPLING

We first show that the Markov chain generated by the KRMS Algorithm 1 is uniformly ergodic. This property ensures convergence to a unique stationary distribution in finite time, which is a crucial requirement in establishing our subsequent theoretical properties.

**Theorem 1.** *Let $\hat{\alpha}$ be estimate of parameter vector $\alpha$ obtained with Algorithm 1. Consider the Markov chain $\{\tilde{z}_t\}_{t\geq 0}$ generated by the following process: given the current state $\tilde{z}_t$ together with the $\hat{\alpha}$, a candidate $\tilde{z}^*$ is generated by randomly sampling from $\tilde{\mathcal{D}}$ and accepted with probability $p_a^* = \min\{1, w(\tilde{z}_t, \hat{\alpha})/w(\tilde{z}^*, \hat{\alpha})\}$. Then, the Markov chain $\{\tilde{z}_t\}_{t\geq 0}$ is irreducible and aperiodic on the finite state space $\tilde{\mathcal{D}}$, and is therefore uniformly ergodic. Its unique stationary distribution $\mathcal{P}'$ has the probability mass function:*

$$\pi(\tilde{z}) = \frac{1/w(\tilde{z}, \hat{\alpha})}{\sum_{z' \in \tilde{\mathcal{D}}} 1/w(z', \hat{\alpha})}, \quad \forall \tilde{z} \in \widetilde{D}.$$

*Consequently, the limiting probability of each sample is proportional to the inverse of its kernel residual score $w(\tilde{z}, \hat{\alpha})$.*

By Theorem 1, our subsampling algorithm yields a u.e.M.c sample converging to $\mathcal{P}'$, which represents a "cleaner" version of the initially contaminated distribution $\mathcal{P}$, with a reduced contamination proportion $0 \leq \theta' < \theta$.

**Theorem 2.** *(Contamination Reduction) Let the original distribution be $\mathcal{P} = (1 - \theta)\mathcal{F} + \theta\mathcal{Q}$. The stationary distribution is a mixture $\mathcal{P}' = (1 - \theta')\mathcal{F}' + \theta'\mathcal{Q}'$, where the new contamination proportion is given by:*

$$\theta' = \frac{\theta S_\mathcal{Q}}{(1 - \theta)S_\mathcal{F} + \theta S_\mathcal{Q}},$$

*where $S_\mathcal{F} = \mathbb{E}_\mathcal{F}(1/w)$ and $S_\mathcal{Q} = \mathbb{E}_\mathcal{Q}(1/w)$ are the expected inverse scores for inliers and outliers, respectively. Consequently, $\theta' < \theta$ if and only if $S_\mathcal{Q} < S_\mathcal{F}$.*

**Remark 2.** *Theorem 2 quantifies the robustness gain. The condition $\theta' < \theta$ holds provided $S_{\mathcal{Q}} < S_{\mathcal{F}}$, implying that outliers possess larger average residual kernel-norm scores (i.e., $\mathbb{E}_{\mathcal{Q}}(w) > \mathbb{E}_{\mathcal{F}}(w)$). This aligns with the intuition of residual-based detection: reweighting inversely to residual scores effectively downweights contamination. To ensure $S_{\mathcal{Q}} < S_{\mathcal{F}}$, we rely on the geometric separation in RKHS. Specifically, we assume outliers are incoherent with the kernel structure (see Proposition 1 in Appendix B ).*

### 4.2 ASYMPTOTIC PROPERTIES OF ESTIMATOR

Now we investigate the theoretical properties of the regularized estimator $f_{\mathbb{S},\lambda}$ defined in Algorithm 1. Due to the theoretical challenges posed by the standard Gaussian kernel, we employ instead a symmetric periodic Gaussian kernel (introduced in the Appendix B). This choice enables analytically tractable approximations, facilitating eigen-decomposition and simultaneous diagonalization for asymptotic analysis (Lin & Brown, 2004; Zeng & Xia, 2019). Specifically, the regularization scheme for regression with symmetric periodic Gaussian kernel is given by $J(f) = \|f\|_{\mathcal{H}_\omega}^2 = \langle f, f \rangle_{\mathcal{H}_\omega}$, where the inner product $\langle f, f \rangle_{\mathcal{H}_\omega}$ is defined analogously to $\langle f, f \rangle_{\mathcal{H}_K}$.

The estimator $f_{\mathbb{S},\lambda}$ is obtained by minimizing the regularized empirical risk using samples $\mathbb{S} = \{\widetilde{z}_i = (\widetilde{\mathbf{x}}_i, \widetilde{y}_i)\}_{i=1}^n$ drawn from the distribution $\mathcal{P}'$. Define

$$\mathcal{R}_{\mathbb{S}}(f) = \frac{1}{n} \sum_{i=1}^n \{f(\widetilde{\mathbf{x}}_i) - \widetilde{y}_i\}^2, \quad f_{\mathbb{S},\lambda} = \arg\min_{f \in \mathcal{H}_\omega} \{\mathcal{R}_{\mathbb{S}}(f) + \lambda J(f)\},$$

$$\mathcal{R}_{\mathcal{P}'}(f) = \int_{\mathcal{Z}} \{f(\widetilde{\mathbf{x}}) - \widetilde{y}\}^2 \, \mathrm{d}\mathcal{P}', \quad f_{\mathcal{P}'} = \arg\min_{f \in \mathcal{H}_\omega} \mathcal{R}_{\mathcal{P}'}(f).$$

**Condition 1.** $\{\widetilde{\mathbf{x}}_i\}_{i=1}^n$ *is a uniformly ergodic Markov chain sample of variable $\widetilde{\mathbf{x}}$, exhibiting uniformly mixing ($\phi$-mixing) properties. The density function $p'(\widetilde{\mathbf{x}})$ of $\widetilde{\mathbf{x}}$ is supported on $[0, \pi]$ and satisfies the boundedness: $0 < c \le p'(\widetilde{\mathbf{x}}) \le C < \infty$ for the positive constants $c$ and $C$.*

**Condition 2.** $\{\epsilon_i\}_{i=1}^n$ *is a sequence of i.i.d. random variables that are independent of $\widetilde{\mathbf{x}}$, and satisfy $\mathbb{E}(\epsilon_i) = 0$ and $\mathbb{E}(\epsilon_i^2) = \sigma^2$.*

**Condition 3.** $f_{\mathcal{P}'} \in \mathcal{H}_{\omega[-\pi,\pi]}^\infty$.

**Condition 4.** $f_{\mathcal{P}'} \in \mathcal{H}_{[-\pi,\pi]}^m$.

The explanation of these conditions and the definition of norms and inner products are given in the Appendix B. For $n$-dependent sequences $a_n$ and $b_n$, the notation $a_n \sim b_n$ means $\lim_{n\to\infty} a_n/b_n = c \in (0, \infty)$.

**Theorem 3.** *Suppose that Conditions 1, 2 and 3 hold. If $\lambda \sim (\ln n)^{\frac{1}{2}}/n$ as $n \to \infty$, the regularization estimator $f_{\mathbb{S},\lambda}$ satisfies*

$$\|f_{\mathbb{S},\lambda} - f_{\mathcal{P}'}\|_0^2 = O_p\left(\frac{(\ln n)^{\frac{1}{2}}}{n}\right).$$

**Theorem 4.** *Suppose that Conditions 1, 2 and 4 hold, and $\omega$ is a constant. If $\lambda = o(1)$ and $(-\ln \lambda)^{\frac{1}{2}}/\omega \sim n^{\frac{1}{2m+1}}$ as $n \to \infty$, the regularization estimator $f_{\mathbb{S},\lambda}$ satisfies*

$$\|f_{\mathbb{S},\lambda} - f_{\mathcal{P}'}\|_0^2 = O_p\left(n^{-\frac{2m}{2m+1}}\right).$$

Theorems 3 and 4 establish that for an infinitely or finitely smooth $m$-th order target function, the estimation error tends to zero as the sample size approaches infinity provided the regularization parameter $\lambda$ is appropriately chosen, demonstrating the consistency of the estimator. While we employ the Gaussian kernel to establish logarithmic convergence, our framework accommodates polynomial-decay kernels (e.g., Sobolev) under Conditions 3–4, readily yielding polynomial rates.

To derive the functional Bahadur representation (FBR) of the estimator, a key prerequisite for establishing its asymptotic theory, we first introduce necessary notation. Let $H_{\omega_t} = H_\omega(t, \cdot)$. For any $f, \Delta f \in \mathcal{H}_\omega$, define

$$S_{n\lambda}(f) = -\frac{2}{n} \sum_{i=1}^n (\widetilde{y}_i - f(\widetilde{\mathbf{x}}_i)) H_{\omega_{\widetilde{\mathbf{x}}_i}} + 2\lambda f,$$

$$DS_{n\lambda}(f)\Delta f = \frac{2}{n}\sum_{i=1}^{n}\Delta f\left(\widetilde{\mathbf{x}}_i\right)H_{\omega_{\widetilde{\mathbf{x}}_i}} + 2\lambda\Delta f.$$

Let $DS_\lambda(f)\Delta f = \mathbb{E}_{p'}\{DS_{n\lambda}(f)\Delta f\}$, $S_\lambda(f) = \mathbb{E}_{p'}\{S_{n\lambda}(f)\}$, $\mathcal{R}_{\mathbb{S},\lambda}(f) = \mathcal{R}_{\mathbb{S}}(f) + \lambda J(f)$, $\mathcal{R}_{\mathcal{P}',\lambda}(f) = \mathbb{E}\{\mathcal{R}_{\mathbb{S},\lambda}(f)\}$, $f_{\mathcal{P}',\lambda} = \arg\min_{f\in\mathcal{H}_\omega}\mathcal{R}_{\mathcal{P}',\lambda}(f)$. Thus, we have $f_{\mathbb{S},\lambda} - f_{\mathcal{P}'} = (f_{\mathcal{P}',\lambda} - f_{\mathcal{P}'}) + f_{\mathbb{S},\lambda} - f_{\mathcal{P}',\lambda}$. Denote $\tilde{f} = f_{\mathcal{P}',\lambda} - DS_\lambda^{-1}(f_{\mathcal{P}',\lambda})S_{n\lambda}(f_{\mathcal{P}',\lambda})$, $f_{\mathbb{S},\lambda} - f_{\mathcal{P}',\lambda} = (f_{\mathbb{S},\lambda} - \tilde{f}) + (\tilde{f} - f_{\mathcal{P}',\lambda})$.

**Theorem 5.** *(Functional Bahadur representation) Suppose that Conditions 1, 2 and 3 hold. If $\lambda \sim (\ln n)^{\frac{1}{2}}/n = o(1)$ as $n \to \infty$, we have*

$$\left\| f_{\mathbb{S},\lambda} - f_{\mathcal{P}'} + \{DS_\lambda(f_{\mathcal{P}'})\}^{-1}S_{n\lambda}(f_{\mathcal{P}'}) \right\|_\lambda^2 = O_p\left(\frac{\ln n}{n^2}\right).$$

Theorem 5 shows that the estimation error can be accurately approximated by a leading linear random term, with the remainder term converging to zero at the high-order rate of $O_p\left(\ln n/n^2\right)$. Now we apply this FBR to show pointwise asymptotic normality of estimators in Sobolev spaces.

**Theorem 6.** *Suppose that Conditions 1, 2 and 3 hold. Let $f_{\mathcal{P}'}(\widetilde{\mathbf{x}}) = \sum_{k=0}^{\infty}f_{\mathcal{P}',k}\phi_k(\widetilde{\mathbf{x}})$, where $f_{\mathcal{P}',k} = \int_{\mathbb{X}}f(\widetilde{\mathbf{x}})\phi_k(\widetilde{\mathbf{x}})d\widetilde{\mathbf{x}}$, $f_0(\widetilde{\mathbf{x}}) = \sum_{k=0}^{\infty}\lambda_k f_{\mathcal{P}',k}\phi_k(\widetilde{\mathbf{x}})/(\lambda+\lambda_k)$. If $\lambda = o(1)$ and $(-\ln\lambda)^{\frac{1}{2}}/\omega \sim n^{\frac{1}{2m+1}}$ as $n \to \infty$, for any $\widetilde{\mathbf{x}}_0 \in [-\pi, \pi]$, there exists a constant $\sigma_{\widetilde{\mathbf{x}}_0}^2 > 0$ such that*

$$\lim_{n\to\infty}\frac{\sigma^2}{(\ln n)^{\frac{1}{2}}}\sum_{k=0}^{\infty}\left(1 + \frac{\lambda}{\lambda_k}\right)^{-2}\phi_k^2(\widetilde{\mathbf{x}}_0) = \sigma_{\widetilde{\mathbf{x}}_0}^2,$$

*we have*

$$\sqrt{\frac{n}{(\ln n)^{\frac{1}{2}}}}\{f_{\mathbb{S},\lambda}(\widetilde{\mathbf{x}}_0) - f_0(\widetilde{\mathbf{x}}_0)\} \xrightarrow{d} \mathcal{N}\left(0, \sigma_{\widetilde{\mathbf{x}}_0}^2\right),$$

**Theorem 7.** *Suppose that Conditions 1, 2 and 4 hold. If $\lambda = o(1)$ and $(-\ln\lambda)^{\frac{1}{2}}/\omega \sim n^{\frac{1}{2m+1}}$ as $n \to \infty$, we have*

$$\left\| f_{\mathbb{S},\lambda} - f_{\mathcal{P}'} + \{DS_\lambda(f_{\mathcal{P}'})\}^{-1}S_{n\lambda}(f_{\mathcal{P}'}) \right\|_\lambda^2 = O_p\left(n^{-\frac{4m}{2m+1}}\right).$$

*For any $\widetilde{\mathbf{x}}_0 \in [-\pi, \pi]$, if there exists a constant $\tilde{\sigma}_{\widetilde{\mathbf{x}}_0}^2 > 0$ such that*

$$\lim_{n\to\infty}\frac{\sigma^2}{n^{\frac{1}{2m+1}}}\sum_{k=0}^{\infty}\left(1 + \frac{\lambda}{\lambda_k}\right)^{-2}\phi_k^2(\widetilde{\mathbf{x}}_0) = \tilde{\sigma}_{\widetilde{\mathbf{x}}_0}^2,$$

*we have*

$$n^{\frac{m}{2m+1}}\{f_{\mathbb{S},\lambda}(\widetilde{\mathbf{x}}_0) - f_0(\widetilde{\mathbf{x}}_0)\} \xrightarrow{d} \mathcal{N}\left(0, \tilde{\sigma}_{\widetilde{\mathbf{x}}_0}^2\right).$$

Theorems 6 and 7 establish that for a target function with an infinitely or finite smooth $m$-th order, the estimator achieves a fast convergence rate. Moreover, when centered by its oracle-smoothed counterpart and properly scaled, the estimator's distribution converges to a normal distribution.

### 4.3 GENERALIZATION BOUND

To characterize the generalization ability of Algorithm 1, we evaluate the quality via its excess risk $\mathcal{R}_{\mathcal{F}}(f_{\mathbb{S},\lambda}) - \mathcal{R}_{\mathcal{F}}(f_0)$. In what follows, we discuss non-asymptotic upper bound of the excess risk. We refine estimation error by exploiting the boundedness of the target function, restricting regression function to a pregiven interval. To this end, we assume that there exists a constant $M > 0$ such that $|y| \le M$ for any $y \in \mathcal{Y}$ and $|f(\mathbf{x})| \le M$ for any $\mathbf{x} \in \mathbb{X}$. Given $\mathcal{H}_\omega \subset \mathcal{H}_K$, let $C(\mathbb{X})$ denote the space of continuous function on $\mathbb{X}$ equipped with the norm: $\|f\|_\infty = \sup_{\mathbf{x}\in\mathbb{X}}|f(\mathbf{x})|$. By the continuity of kernel $K(\cdot)$ and compactness of $\mathbb{X}$, we have $\kappa = \sup_{\mathbf{x}\in\mathbb{X}}K(\mathbf{x},\mathbf{x}) < \infty$, which implies the following key inequality: $\|f\|_\infty \le \kappa\|f\|_K^2$ for $\forall f \in \mathcal{H}_K$.

When the sample dataset contains contaminated observations, the traditional error decomposition approach faces additional challenges. To address this issue, we consider a new error decomposition for the excess risk: $\mathcal{R}_{\mathcal{F}}(f_{\mathbb{S},\lambda}) - \mathcal{R}_{\mathcal{F}}(f_0)$, which is given in Propositions 2–5.

**Theorem 8.** *Suppose that* $\mathbb{S} = \{\widetilde{z}_i = (\widetilde{\mathbf{x}}_i, \widetilde{y}_i)\}_{i=1}^n$ *is a u.e.M.c sample. If* $\mathcal{D}(\lambda)$ *satisfies* $\mathcal{D}(\lambda) \leq c_q \lambda^q$ *with* $\lambda = n^{-\vartheta_1}$ *and* $\vartheta_1 = \min\{1/(2-q), 1/((1+s)q), 1/q\}$, *thus, for any* $0 < \delta < 1$, *with probability at least* $1 - \delta$,

$$\mathcal{R}_{\mathcal{F}}(f_{\mathbb{S},\lambda}) - \mathcal{R}_{\mathcal{F}}(f_0) \leq C_2 \ln(2/\delta) n^{-\vartheta_1 q} + 48M^2 \theta',$$

*where* $C_2$ *is a constant independent of* $n$ *and* $\delta$, $\theta'$ *is the proportion of contaminated data in the subsample set after subsampling.*

Theorem 8 establishes asymptotic property of the excess risk, and its convergence rate is $\mathcal{O}(n^{-1})$ as $n \to \infty$ and $\theta' \to 0$, which is consistent with the known optimal rate for regularized least square type algorithms Li et al. (2017).

## 5  SIMULATION STUDIES

Simulation studies are conducted to evaluate the finite-sample performance of the proposed KRMS method for kernel-based regularized least squares regression. Our evaluation focuses on the method's robustness under different data contamination scenarios. The simulation design incorporates two fundamentally distinct data generating processes: a linear regression and a nonlinear regression. For each experimental configuration, we generate a training set with $N = 10000$ observations and an independent test set with $N_{\text{test}} = 2000$ observations. The proposed subsampling algorithm is applied to draw subsamples of sizes $n \in \{500, 1000, 1500\}$. The entire experiment is repeated $M = 100$ times to ensure statistical reliability. For each replication $m$, we compute mean squared error (MSE) for the test set via $\text{MSE}_m = \frac{1}{N_{\text{test}}} \sum_{i=1}^{N_{\text{test}}} (\widetilde{y}_i - \hat{\widetilde{y}}_i)^2$, where $\hat{\widetilde{y}}_i$ is the predictive value of $\widetilde{y}_i$. Our primary performance metric is the average mean squared error (AMSE) for all replications: $\text{AMSE} = \frac{1}{M} \sum_{m=1}^{M} \text{MSE}_m$. To assess method stability and the performance of the proposed algorithm, we report the standard deviation (SD) of the MSEs values among $M$ replications, and the positive screening rate (PSR, %), defined as the proportion of correctly identified uncontaminated observations in each subsample, respectively. For comparison, we consider the following five subsampling algorithms: MS-KLSR–Markov sampling with kernel-based regularized least squares regression (Zou et al., 2014), UNIF-KLSR–uniform subsampling with kernel-based regularized least squares regression, UNIF-LSR–uniform subsampling for linear least squares regression, GMS-LSR–gradient-based Markov sampling for linear least squares regression (Gong et al., 2020), and LGS-LSR–low gradient-based subsampling for linear least squares regression (Jing, 2023). To implement kernel-based regression algorithms , we take the Gaussian kernel $K(x, t) = \exp\{-(x - t)^2/4\}$. The regularization parameter is selected using the LOOCV strategy.

**Experiment 1** (Linear model). Dataset $\{(\mathbf{x}_i, y_i)\}_{i=1}^N$ is generated from linear model $y_i = x_{i1} + 2x_{i2} + 3x_{i3} + 4x_{i4} + \epsilon_i$, where $x_{i1}, \ldots, x_{i4}$ are independently generated from uniform distribution $U(0, 1)$, and $\epsilon_i$'s are independently sampled from the standard normal distribution. To create corrupted observations using the mechanism: for predictors $\mathbf{x}_i$, we replace a proportion $\theta$ of observations with random values drawn from $\mathbf{W}_i$; for corresponding response variable $y_i$, we replace contaminated cases with values drawn from $O_i$ for $\theta \in \{0.1, 0.2, 0.3, 0.4\}$. We assume that $O_i$ follows the normal distribution $\mathcal{N}(0, 10)$, inducing significant fluctuations of contaminated observations, and $\mathbf{W}_i$ follows the following three distributions: (M1) $W_{ij} \sim t(1)$, (M2) $W_{ij} \sim \exp(1)$ and (M3) $W_{ij} \sim F(1, 1)$, where $t(1)$ represents the $t$-distribution with one degree of freedom, $\exp(1)$ denotes standard exponential distribution, $F(\cdot, \cdot)$ is the F-distribution, and $W_{ij}$ is the $j$-th component of $\mathbf{W}_i$, which are designed to investigate robustness to different types of outliers.

Tables 1–3 (Tables 2 and 3 are given in Appendix D) indicate that the KRMS-KLSR method outperforms others in that the former consistently achieves the relatively small AMSE and SD values and maintains high PSR values for nearly all scenarios. For contamination schemes M1 and M3 together with low values of $\theta$ (e.g., $\theta \leq 0.2$), the LGS-LSR method shows marginally superior performance over KRMS-KLSR based on AMSE values, but it exhibits lower SD and higher PSR values, implying poorer stability and reliability compared to KRMS-KLSR. However, under severe contamination (e.g., $\theta > 0.2$) or complex outliers (e.g., M2 mechanism), KRMS-KLSR offers considerable improvements: it exhibits only moderate AMSE increases while LGS-LSR suffers from substantial performance degradation. GMS-LSR demonstrates intermediate performance, bridging the gap between LGS-LSR and conventional methods. Non-robust methods (e.g., UNIF-LSR)

Table 1: Performance comparison of KRMS and five competing subsampling methods for corrupted mechanism M1 in Experiment 1

| $\theta$ | Method | $n = 500$ | | | $n = 1000$ | | | $n = 1500$ | | |
|---|---|---|---|---|---|---|---|---|---|---|
| | | AMSE | SD | PSR | AMSE | SD | PSR | AMSE | SD | PSR |
| 0.1 | UNIF-KLSR | 1.605 | 0.513 | 90.04% | 1.861 | 0.520 | 89.92% | 2.238 | 0.526 | 90.15% |
| | MS-KLSR | 1.364 | 0.324 | 96.75% | 1.500 | 0.346 | 96.60% | 1.857 | 0.476 | 96.76% |
| | KRMS-KLSR | 1.118 | **0.043** | **99.93%** | 1.084 | **0.035** | **99.94%** | 1.070 | **0.033** | **99.93%** |
| | UNIF-LSR | 18.175 | 4.974 | 90.04% | 22.156 | 3.359 | 89.92% | 23.867 | 2.799 | 90.15% |
| | GMS-LSR | 2.975 | 1.681 | 94.38% | 4.085 | 1.760 | 93.61% | 5.076 | 2.323 | 93.56% |
| | LGS-LSR | **1.032** | 0.050 | 99.30% | **1.032** | 0.045 | 99.26% | **1.032** | 0.081 | 99.16% |
| 0.2 | UNIF-KLSR | 2.545 | 0.521 | 80.02% | 2.797 | 0.507 | 80.16% | 3.225 | 0.444 | 80.22% |
| | MS-KLSR | 2.133 | 0.528 | 91.52% | 2.425 | 0.509 | 91.57% | 2.862 | 0.437 | 92.22% |
| | KRMS-KLSR | 1.117 | **0.046** | **99.89%** | 1.082 | **0.035** | **99.87%** | 1.074 | **0.035** | **99.86%** |
| | UNIF-LSR | 25.266 | 2.203 | 80.02% | 26.584 | 1.449 | 80.16% | 27.235 | 0.844 | 80.22% |
| | GMS-LSR | 9.647 | 3.397 | 85.40% | 11.755 | 3.094 | 85.25% | 13.691 | 2.706 | 85.55% |
| | LGS-LSR | **1.083** | 0.092 | 98.42% | **1.065** | 0.083 | 98.30% | **1.068** | 0.089 | 98.14% |
| 0.3 | UNIF-KLSR | 3.337 | 0.702 | 70.41% | 3.859 | 0.543 | 70.08% | 4.434 | 0.556 | 70.00% |
| | MS-KLSR | 2.981 | 0.587 | 84.94% | 3.570 | 0.469 | 85.23% | 4.116 | 0.450 | 86.02% |
| | KRMS-KLSR | **1.127** | **0.043** | **99.76%** | **1.085** | **0.037** | **99.79%** | **1.069** | **0.033** | **99.73%** |
| | UNIF-LSR | 27.075 | 1.154 | 70.41% | 27.835 | 0.644 | 70.08% | 28.102 | 0.507 | 70.00% |
| | GMS-LSR | 16.172 | 3.216 | 76.47% | 18.815 | 2.179 | 76.87% | 19.973 | 2.153 | 77.56% |
| | LGS-LSR | 1.138 | 0.121 | 97.12% | 1.155 | 0.185 | 96.95% | 1.126 | 0.121 | 96.74% |
| 0.4 | UNIF-KLSR | 4.494 | 0.792 | 60.06% | 5.264 | 0.869 | 60.03% | 5.868 | 0.654 | 59.83% |
| | MS-KLSR | 4.341 | 0.699 | 75.30% | 5.055 | 0.705 | 76.09% | 5.570 | 0.535 | 76.84% |
| | KRMS-KLSR | **1.132** | **0.047** | **99.62%** | **1.086** | **0.034** | **99.61%** | **1.069** | **0.041** | **99.57%** |
| | UNIF-LSR | 27.739 | 0.657 | 60.06% | 28.178 | 0.528 | 60.03% | 28.280 | 0.468 | 59.83% |
| | GMS-LSR | 20.541 | 2.332 | 67.39% | 22.434 | 1.816 | 68.01% | 22.913 | 1.351 | 69.14% |
| | LGS-LSR | 1.328 | 0.287 | 95.03% | 1.275 | 0.192 | 94.83% | 1.273 | 0.176 | 94.46% |

demonstrate severe degradation, leading to a relatively large AMSE values and a relatively low PSR values, which confirms the necessity of robust subsampling. Thus, the proposed KRMS-KLSR method retains stable performance with increasing contamination levels and larger subsample sizes. To save space, Experiment 2 for nonlinear regression model are moved to Appendix D.

# 6 CONCLUSION

Corrupted observations from outliers, measurement errors, or multi-source heterogeneity are widely encountered in biomedicine, environmental science, and economics. Traditional statistical inference often faces huge challenges such as computational inefficiency and sensitivity to contamination. Subsampling has emerged as a powerful strategy to select representative subsets while discarding contaminated points. However, existing methods like score-based or low-gradient subsampling mainly focus on parametric models and perform poorly under high contamination.

To address these issues, we propose a KRMS method for nonparametric regression with contaminated data. Our key innovation is to define subsampling probability as the ratio of the absolute residual to the kernel norm of covariates, which dynamically downweights outliers while preserving clean data. Unlike conventional methods, the proposed approach explicitly accounts for both the predictive error and the geometric structure of the data in a RKHS, ensuring robustness even under severe contamination. Theoretical guarantees, including consistency and asymptotic normality and generalization bounds under RKHS regularization, are established under some conditions.

Empirical results demonstrate KRMS's superiority in high-contamination settings, with stable performance across simulations and real-data applications. While the method currently focuses on continuous, fully observed responses, future work will extend it to classification, distributed streaming data, missing data, and high- or ultrahigh-dimensional optimization via deep learning approaches.

## REPRODUCIBILITY STATEMENT

Detailed explanations on Tables 1–9 are given in Appendix D. We also attach our codes to facilitate the reproduction of our experiments.

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

## A  CHATGPT USAGE

During the preparation of this manuscript, we used ChatGPT (GPT-4) solely for the purpose of polishing language and syntax. The tool was employed exclusively for language refinement and was not used to generate any scientific content, ideas, experimental designs, or data interpretations.

## B  MAIN TOOLS FOR THEORETICAL RESULTS

### B.1  MAIN TOOLS FOR VALIDITY OF SUBSAMPLING

We explicitly quantify the robustness of the proposed method by analyzing the stationary distribution of the generated Markov chain. We assume the contaminated dataset $\widetilde{\mathcal{D}}$ is an i.i.d. realization of the contaminated distribution $\mathcal{P} = (1 - \theta)\mathcal{F} + \theta\mathcal{Q}$, with density $p(\tilde{z})$. To ensure technical rigor, we introduce the following regularity assumption.

**Assumption 1** (Non-vanishing Score). *There exists a constant $\delta > 0$ such that the residual kernel-norm score satisfies $w(\tilde{z}, \alpha) \geq \delta$ almost surely for all $\tilde{z} \in \mathrm{supp}(\mathcal{P})$. Although Algorithm 1 operates on a finite dataset $\widetilde{\mathcal{D}}$ (the empirical measure), for our theoretical analysis, we consider its population-level counterpart where proposals are drawn from the underlying contaminated distribution $\mathcal{P}$. This allows us to characterize the distributional robustness of the method.*

First, we identify the exact form of the stationary distribution generated by Algorithm 1. Note that drawing a candidate uniformly from the dataset $\widetilde{\mathcal{D}}$ is empirically equivalent to drawing a proposal from the distribution $\mathcal{P}$.

**Lemma 1** (Stationary Distribution). *The Markov chain generated by Algorithm 1, utilizing the contaminated density $p(\tilde{z}^*)$ as the independent proposal distribution and acceptance probability $\min\{1, w(\tilde{z}, \alpha)/w(\tilde{z}^*, \alpha)\}$, converges to a unique stationary distribution $\mathcal{P}'$ with density:*

$$p'(\tilde{z}) = \frac{1}{Z}\frac{p(\tilde{z})}{w(\tilde{z}, \alpha)},$$

*where $Z = \int [p(\tilde{z})/w(\tilde{z}, \alpha)]d\tilde{z}$ is the normalizing constant.*

*Proof of Lemma 1.* The Algorithm 1 utilizes an Independent Metropolis-Hastings sampler. The proposal distribution is the contaminated distribution $p(\widetilde{z}^*)$. The transition kernel is

$$T(\tilde{z} \to \tilde{z}^*) = p(\tilde{z}^*) \min\left\{1, \frac{w(\widetilde{z}, \alpha)}{w(\tilde{z}^*, \alpha)}\right\} + (1 - r(\tilde{z}))\delta_{\tilde{z}}(\tilde{z}^*).$$

To verify $p'(\tilde{z}) \propto p(\tilde{z})/w(\tilde{z}, \alpha)$ is the stationary density, we check the detailed balance condition: $p'(\tilde{z})T(\tilde{z} \to \tilde{z}^*) = p'(\tilde{z}^*)T(\tilde{z}^* \to \tilde{z})$. Substituting the expressions, we obtain

$$\text{LHS} = \frac{1}{Z}\frac{p(\tilde{z})}{w(\tilde{z}, \alpha)} \cdot p(\tilde{z}^*) \cdot \min\left\{1, \frac{w(\tilde{z}, \alpha)}{w(\tilde{z}^*, \alpha)}\right\}$$

$$= \frac{p(\tilde{z})p(\tilde{z}^*)}{Z} \cdot \frac{1}{w(\tilde{z}, \alpha)} \min\left\{1, \frac{w(\tilde{z}, \alpha)}{w(\tilde{z}^*, \alpha)}\right\}$$

$$= \frac{p(\tilde{z})p(\tilde{z}^*)}{Z} \min\left\{\frac{1}{w(\tilde{z}, \alpha)}, \frac{1}{w(\tilde{z}^*, \alpha)}\right\}.$$

By the symmetry of the RHS, we obtain

$$\text{RHS} = \frac{p(\tilde{z}^*)p(\tilde{z})}{Z} \min\left\{\frac{1}{w(\tilde{z}^*, \alpha)}, \frac{1}{w(\tilde{z}, \alpha)}\right\}.$$

Since LHS = RHS, detailed balance holds. $\qquad\square$

To ensure the condition $S_{\mathcal{Q}} < S_{\mathcal{F}}$ holds, we provide a geometric justification based on the properties of RKHS.

**Proposition 1** (Outlier Incoherence). *Let $\hat{f}_\lambda$ be the pilot estimator minimizing $\mathcal{R}_{\widetilde{\mathcal{D}}}(f) + \lambda\|f\|_K^2$. Assume the target function $f_0$ has a bounded RKHS norm $\|f_0\|_K \leq R$, while the outliers are incoherent with the kernel structure such that fitting them requires a function norm $\|g\|_K \gg R$. If $\lambda$ is sufficiently large, then for any isolated outlier $\hat{z}_{out}$ and clean point $z$, we have $w(\hat{z}, \alpha) \gg w(z, \alpha)$, which implies $S_{\mathcal{Q}} < S_{\mathcal{F}}$.*

*Proof of Proposition 1.* Consider the pilot estimator $\hat{f}_\lambda$. By the Representer Theorem we have $\hat{f}_\lambda(\cdot) = \sum_j c_j K(\widetilde{\mathbf{x}}_j, \cdot)$. The objective function penalizes both the fitting error and the RKHS norm $\|f\|_K^2$. For an outlier $(\hat{\mathbf{x}}, \hat{y})$ that deviates from the smooth manifold of $f_0$ by a distance $\Delta$, forcing the estimator to fit this point (i.e., reducing residual to 0) would require adding a sharp "spike" function. Such a function possesses a large RKHS norm, leading to a significant increase in the penalty term $\lambda\|f\|_K^2$. Since $\lambda$ is chosen to be large (promoting smoothness), the optimization favors minimizing the penalty over fitting the outlier. Consequently, the residual $|\hat{y} - \hat{f}_\lambda(\hat{\mathbf{x}})|$ remains proportional to $\Delta$ (large), whereas inliers are well-approximated with small residuals. Thus, $w(\hat{z}, \alpha) \gg w(z, \alpha)$, which implies $S_{\mathcal{Q}} < S_{\mathcal{F}}$. $\qquad\square$

**Remark 3.** *(Remark on Pilot Estimator.) One might concern that the pilot estimator (using squared loss) could interpolate outliers, vanishing their residuals. However, although the squared-error loss is generally non-robust, the constraint imposed by a large regularization parameter $\lambda$ in the pilot phase acts as a global smoothness prior. This prevents the function from interpolating sparse, high-magnitude outliers, thereby ensuring $w(\hat{z}, \alpha)$ remains large.*

### B.2 MAIN TOOLS FOR ASYMPTOTIC PROPERTIES OF ESTIMATOR

Note that $\mathcal{H}_K$ is an RKHS, by the Riesz representing theorem, functions in $\mathcal{H}_K$ satisfy the reproducing property: $\langle K(\mathbf{x}, \cdot), f(\cdot)\rangle_{\mathcal{H}_K} = f(\mathbf{x})$ for all $f \in \mathcal{H}_K$ and $\mathbf{x} \in \mathbb{X}$. Following Mercer (1909), a reproducing kernel $K(\cdot, \cdot)$ can be expressed as $K(\mathbf{x}_i, \mathbf{x}_j) = \sum_{k=1}^{\infty} \lambda_k \phi_k(\mathbf{x}_i)\phi_k(\mathbf{x}_j)$, where $\lambda_k$'s are the eigenvalues of $K(\cdot, \cdot)$, and $\phi_k(\cdot)$'s are the corresponding eigenfunctions, forming a sequence of orthogonal basis functions in $L^2(\mathbb{X})$ with respect to the inner product: $\langle \phi_i, \phi_j\rangle_{\mathcal{H}_K} = \delta_{ij}/\lambda_i$, where $\delta_{ij}$ is the Kronecker delta. The RKHS can alternatively be defined in terms of these eigenvalues and eigenfunctions:

$$\mathcal{H}_K = \left\{f(\mathbf{x}) = \sum_{k=1}^{\infty} f_k \phi_k(\mathbf{x}) : \sum_{k=1}^{\infty} \frac{f_k^2}{\lambda_k} < \infty\right\},$$

where $f_k = \int_{\mathbb{X}} f(\mathbf{x})\phi_k(\mathbf{x})d\mathbf{x}$. This spectral representation facilitates theoretical analysis, praticularly in studying the asymptotic behaviour of estimator derived from RKHS (Zeng & Xia, 2019). To study the asymptotic performance of estimator, Lin & Brown (2004) introduced two RKHSs: an infinite order Sobolev space with periodic functions

$$\mathcal{S}_{\omega[a,b]}^{\infty} = \Big\{ f \in L^2(a,b) : f \text{ is } (b-a)- \text{ periodic with}$$

$$\sum_{m=0}^{\infty} \frac{\omega^{2m}}{m!2^m} \int_a^b \left[ f^{(m)}(t) \right]^2 dt < \infty \Big\},$$

and an $m$-th order Sobolev space with periodic functions

$$\mathcal{S}_{[a,b]}^{m} = \Big\{ f \in L^2(a,b) : f \text{ is } (b-a)- \text{ periodic with}$$

$$\int_a^b [f(t)]^2 + \left[ f^{(m)}(t) \right]^2 dt < \infty \Big\}.$$

Zeng & Xia (2019) introduced the symmetric periodic Gaussian kernel

$$H_\omega(t,s) = K_{\omega,-\pi,\pi}(s,t) + K_{\omega,-\pi,\pi}(s,-t),$$

where $K_{\omega,-\pi,\pi}(s,t) = \sum_{k=-\infty}^{\infty} K_\omega^0(t-s-2k\pi,0)$ is the periodic Gaussian kernel with period $2\pi$, and

$$K_\omega^0(t,s) = \frac{1}{\sqrt{2\pi}\omega} e^{-(s-t)^2/\omega^2},$$

is the well-known Gaussian reproducing kernel function. Let $\mathcal{H}_{\omega[-\pi,\pi]}^{\infty}$ be the RKHS corresponding to $H_\omega(t,s)$, which is an infinite order Sobolev space with symmetric functions. This RKHS consists of symmetric functions on $[-\pi,\pi]$, and is a subspace of infinite order Sobolev space.

Following Zeng & Xia (2019), $\mathcal{H}_{\omega[-\pi,\pi]}^{\infty}$ can be written as

$$\mathcal{H}_{\omega[-\pi,\pi]}^{\infty} = \left\{ g : g(t) = \sum_{k=0}^{\infty} g_k \xi_k(t), \sum_{k=0}^{\infty} \frac{g_k^2}{\lambda_{k,\omega}} < \infty \right\}$$

$$= \left\{ g : g(-t) = g(t), g \in \mathcal{S}_{\omega[-\pi,\pi]}^{\infty} \right\},$$

where $\lambda_{k,\omega} = \exp(-k^2\omega^2/2)$, $\xi_0(t) = \pi^{-1/2}$, $\xi_k(t) = \sqrt{2/\pi}\cos(kt)$. Also, the $m$-th order Sobolev space with symmetric functions can be expressed as

$$\mathcal{H}_{[-\pi,\pi]}^{m} = \left\{ g : g(t) = \sum_{k=0}^{\infty} g_k \xi_k(t), \sum_{k=0}^{\infty} \frac{g_k^2}{\rho_k} < \infty \right\}$$

$$= \left\{ g : g(-t) = g(t), g \in \mathcal{S}_{[-\pi,\pi]}^{m} \right\}.$$

where $\rho_0 = 1$ and $\rho_k = k^{2m} + 1$. Specifically, in the considered RKHS, every function can be expanded orthogonally in the cosine basis, where each coefficient $g_k$ must be scaled by the reciprocal of its corresponding eigenvalue to ensure finiteness of the induced norm.

In order to study the asymptotic performance of estimator, we need some conditions, which are displayed in the main tex, and we now explain them. Condition 1 is used to ensure that every point in the support set has a specific probability density and is bounded. This guarantees the convergence of the integral and has been utilized in Zeng & Xia (2019). Condition 2 is a standard assumption in classical regression models. Condition 3 postulates a high degree of smoothness for the target function $f_{\mathcal{P}'}$, typically implying that it is infinitely differentiable on the domain $[-\pi,\pi]$. Condition 4 quantifies the smoothness of the target function $f_{\mathcal{P}'}$ by postulating that it belongs to the $m$-th order Sobolev space $\mathcal{H}^m$, meaning that the function and its derivatives up to order $m$ are square-integrable. To facilitate theoretical analysis, we standardize our symmetric periodic Gaussian kernel as

$$\tilde{H}_\omega(t,s) = \frac{H_\omega(t,s)}{\sqrt{f(t)f(s)}},$$

which is simply denoted as $H_\omega(t, s)$. Denote $\|f\|_{\mathcal{H}_\omega}^2 = \langle f, f \rangle_{\mathcal{H}_\omega}$. For asymptotic analysis, we define the following norms and inner products (Zeng & Xia, 2019):

$$\|f\|_0 = \left[ \mathbb{E}_{p'} \left\{ f^2(\widetilde{\mathbf{x}}) \right\} \right]^{\frac{1}{2}} = \left[ \int_0^\pi f^2(t) p'(t) dt \right]^{\frac{1}{2}},$$

$$\|f\|_\lambda = \left( \|f\|_0^2 + \lambda \|f\|_{\mathcal{H}}^2 \right)^{\frac{1}{2}},$$

$$\langle f_1, f_2 \rangle_0 = \frac{1}{4} \left( \|f_1 + f_2\|_0^2 - \|f_1 - f_2\|_0^2 \right),$$

$$\langle f_1, f_2 \rangle_\lambda = \langle f_1, f_2 \rangle_0 + \lambda \langle f_1, f_2 \rangle_{\mathcal{H}_\omega}.$$

### B.3 MAIN TOOLS FOR GENERALIZATION BOUND

Note that

$$\mathcal{R}_{\mathbb{S}}(f) = \frac{1}{n} \sum_{i=1}^n \{ f(\widetilde{\mathbf{x}}_i) - \widetilde{y}_i \}^2, = \arg \min_{f \in \mathcal{H}_\omega} \{ \mathcal{R}_{\mathbb{S}}(f) + \lambda J(f) \}. \tag{4}$$

To bound the excess risk of (4) for u.e.M.c. samples, similarly to Gong et al. (2015), we first define the optimal regularization error $\mathcal{D}(\lambda)$ as

$$\mathcal{D}(\lambda) = \inf_{f \in \mathcal{H}_K} \left\{ \mathcal{R}_{\mathcal{F}}(f) - \mathcal{R}_{\mathcal{F}}(f_0) + \lambda \|f\|_K^2 \right\},$$

which depicts the approximation ability of the hypothesis space $\mathcal{H}_K$ relative to the optimal mapping $f_0$. Thus, we the following relationship

$$f_\lambda = \arg \min_{f \in \mathcal{H}_K} \left\{ \mathcal{R}_{\mathcal{F}}(f) - \mathcal{R}_{\mathcal{F}}(f_0) + \lambda \|f\|_K^2 \right\},$$

respectively. Thus, following Gong et al. (2015), the approximation ability of the target function $f_0$ can be characterized with exponent $0 < q \le 1$ satisfying

$$\mathcal{D}(\lambda) \le c_q \lambda^q \tag{5}$$

for some constant $c_q$ and any $\lambda > 0$. This inequality ensures that the learning algorithm based on the RKHS and regularization methods can approximate the target function at a convergence rate determined by the exponent $q$.

To bound the excess risk, we consider a new error decomposition for the excess risk.

**Proposition 2.** *Let $f_{\mathbb{S},\lambda}$ be the estimator defined in (4) based on the contaminated sample $\mathbb{S} = \{\widetilde{z}_i = (\widetilde{\mathbf{x}}_i, \widetilde{y}_i)\}_{i=1}^n$, where $\widetilde{\mathbf{x}}_i = (\widetilde{x}_{i1}, \dots, \widetilde{x}_{ip})^\top \in \mathbb{X}$'s are drawn from the mixture distribution $\mathcal{P}' = (1 - \theta')\mathcal{F} + \theta'\mathcal{Q}$, where $\mathcal{F}$ is the true distribution and $\mathcal{Q}$ is the contaminated distribution. Similarly, let $f_{\mathcal{D},\lambda}$ be the estimator defined as $f_{\mathcal{D},\lambda} = \arg \min_{f \in \mathcal{H}} \{ \mathcal{R}_{\mathcal{D}}(f) + \lambda J(f) \}$, computed from the uncontaminated sample $\mathcal{D} = \{z_i = (\mathbf{x}_i, y_i)\}_{i=1}^n$, where $\mathbf{x}_i = (x_{i1}, \dots, x_{ip})^\top \in \mathbb{X}$ drawn exclusively from $\mathcal{F}$ with regularization parameter $\lambda > 0$. Thus, we have*

$$\mathcal{R}_{\mathcal{F}}(f_{\mathbb{S},\lambda}) - \mathcal{R}_{\mathcal{F}}(f_0) \le \mathcal{S}(\mathcal{D}, \mathbb{S}, \lambda) + \mathcal{A}(\mathcal{D}, \mathbb{S}) + \mathcal{D}(\lambda),$$

*where $\mathcal{S}(\mathcal{D}, \mathbb{S}, \lambda) = \{ \mathcal{R}_{\mathcal{F}}(f_{\mathbb{S},\lambda}) - \mathcal{R}_{\mathcal{D}}(f_{\mathbb{S},\lambda}) \} + \{ \mathcal{R}_{\mathcal{D}}(f_\lambda) - \mathcal{R}_{\mathcal{F}}(f_\lambda) \}$, $\mathcal{A}(\mathcal{D}, \mathbb{S}) = \mathcal{R}_{\mathcal{D}}(f_{\mathbb{S},\lambda}) - \mathcal{R}_{\mathcal{D}}(f_{\mathcal{D},\lambda})$, and $\mathcal{D}(\lambda) = \{ \mathcal{R}_{\mathcal{F}}(f_\lambda) - \mathcal{R}_{\mathcal{F}}(f_0) + \lambda \|f_\lambda\|_K^2 \}$. Here $\mathcal{S}(\mathcal{D}, \mathbb{S}, \lambda)$, $\mathcal{A}(\mathcal{D}, \mathbb{S})$ and $\mathcal{D}(\lambda)$ denote the sample error, contamination error and regularization error, respectively.*

The covering number provides a natural measure of complexity for hypothesis spaces, quantifying their capacity through metric entropy. For its definition, we refer the reader to Gong et al. (2015). Extensive results exist on covering number bounds (Zhou, 2002; Zhang, 2002). Of particular interest is the RKHS ball: $\mathcal{B}_\varsigma = \{ f \in \mathcal{H}_1 : \|f\| \le \varsigma \} \subset C(\mathbb{X})$ whose covering numbers are well studied. We denote $\mathcal{N}(\epsilon) = \mathcal{N}(\mathcal{B}_1, \epsilon)$ for the unit ball case. Following Samson (2000), we measure variable dependence via the operator norm $\|\Gamma\|$ of the covariance matrix $\Gamma$. This leads to our key decomposition of sample error:

$$\begin{aligned}
\mathcal{S}(\mathcal{D}, \mathbb{S}, \lambda) &= [\mathcal{R}_{\mathcal{F}}(f_{\mathbb{S},\lambda}) - \mathcal{R}_{\mathcal{D}}(f_{\mathbb{S},\lambda})] + [\mathcal{R}_{\mathcal{D}}(f_\lambda) - \mathcal{R}_{\mathcal{F}}(f_\lambda)] \\
&= \{ [\mathcal{R}_{\mathcal{F}}(f_{\mathbb{S},\lambda}) - \mathcal{R}_{\mathcal{F}}(f_0)] - [\mathcal{R}_{\mathcal{D}}(f_{\mathbb{S},\lambda}) - \mathcal{R}_{\mathcal{D}}(f_0)] \} \\
&\quad + \{ [\mathcal{R}_{\mathcal{D}}(f_\lambda) - \mathcal{R}_{\mathcal{D}}(f_0)] - [\mathcal{R}_{\mathcal{F}}(f_\lambda) - \mathcal{R}_{\mathcal{F}}(f_0)] \} \\
&= \mathcal{S}_1(\mathcal{D}, \mathbb{S}, \lambda) + \mathcal{S}_2(\mathcal{D}, \lambda).
\end{aligned}$$

Based on this decomposition, we obtain the following propositions.

**Proposition 3.** *For $\mathcal{H}_K$ with polynomial complexity exponent $s > 0$ and any $0 < \delta < 1$, with probability at least $1 - \delta$,*

$$\mathcal{S}_1(\mathcal{D}, \mathbb{S}, \lambda) \leq \frac{1}{2} \left\{ \mathcal{R}_{\mathcal{F}}(f_{\mathbb{S}, \lambda}) - \mathcal{R}_{\mathcal{F}}(f_0) \right\}$$

$$+ \frac{14336 M^2 \|\Gamma\|^2 \ln(1/\delta)}{n}$$

$$+ \left( \frac{14336 M^2 \|\Gamma\|^2 c_s (64 M R)^s}{n} \right)^{\frac{1}{1+s}}.$$

**Proposition 4.** *If $\|K\|_\infty \leq \kappa$, thus, for any $0 < \delta < 1$, with probability at least $1 - \delta$,*

$$\mathcal{S}_2(\mathcal{D}, \lambda) \leq \frac{1}{2} \mathcal{D}(\lambda) + \frac{56 \|\Gamma\|^2 (\kappa \mathcal{D}(\lambda)/\lambda + 3M)^2 \ln(1/\delta)}{n}.$$

**Proposition 5.** *For contamination proportion $\theta' \in [0, 1/2)$ and any $0 < \delta < 1$, with probability at least $1 - \delta$,*

$$\mathcal{A}(\mathcal{D}, \mathbb{S}) \leq 24 M^2 \theta' + \frac{896 \|\Gamma\|^2 M^2 \ln(2/\delta)}{n} \theta'.$$

By Propositions 2–5, we can establish the bound of the excess risk based on regularization regression for u.e.M.c. samples.

## C  PROOFS OF THEORETICAL RESULTS

### C.1  PROOF OF VALIDITY OF SUBSAMPLING

*Proof of Theorem 1.* To prove that the Markov chain is uniformly ergodic, we will demonstrate that it satisfies three conditions: (i) Finite State Space, (ii) Irreducible, (iii) Aperiodic for a given $\alpha$ (e.g., $\hat{\alpha}$). For a Markov chain on a finite state space, these three conditions are sufficient for uniform ergodicity (Levin & Peres, 2017).

(i) Finite State Space. The state space $\mathcal{S}$ of the Markov chain corresponds to the sample set in the given dataset $\hat{\mathcal{D}} = \{\tilde{z}_1, \ldots, \tilde{z}_n\}$. With $n$ samples in $\hat{\mathcal{D}}$, the state space has finite cardinality $|\mathcal{S}| = n$.

(ii) Irreducibility: Let $\tilde{z}_i$ and $\tilde{z}_j \in \mathcal{S}$ be two arbitrary states. To establish irreducibility, it suffices to show that the one-step transition probability $\Pr(\tilde{z}_{t+1} = \tilde{z}_j \mid \tilde{z}_t = \tilde{z}_i)$ is positive for all $i, j$. Within the 'while' loop in Algorithm 1, a candidate point $\tilde{z}_j$ is drawn randomly from $\tilde{\mathcal{D}}$, the probability of proposing $\tilde{z}_j$ is exactly $1/n > 0$. This candidate is accepted with probability

$$p_a^* = \min \left\{ 1, \frac{w(\tilde{z}_i, \alpha)}{w(\tilde{z}_j, \alpha)} \right\}$$

Since all importance scores $w(\cdot, \alpha)$ are positive, the acceptance probability is bounded away from zero. The product of these positive probabilities ensures $P(\tilde{z}_{t+1} = \tilde{z}_j \mid \tilde{z}_t = \tilde{z}_i) > 0$. Hence, the states constitute an irreducible Markov chain.

(iii) Aperiodicity: To establish aperiodicity, it suffices to prove that the self-transition probability $\Pr(\tilde{z}_{t+1} = \tilde{z}_i \mid \tilde{z}_t = \tilde{z}_i)$ is strictly positive for any state $\tilde{z}_i \in \mathcal{S}$. When the chain is in state $\tilde{z}_i$, a candidate $\tilde{z}^* = \tilde{z}_i$ is drawn uniformly from $\tilde{\mathcal{D}}$ with probability $1/n$. Since the acceptance probability for this candidate is

$$p_a^* = \min \left\{ 1, \frac{w(\tilde{z}_i, \alpha)}{w(\tilde{z}_i, \alpha)} \right\} = 1,$$

the transition is always accepted. Hence, the self-transition probability is bounded below by $1/n > 0$, which implies that the Markov chain is aperiodic.

The Markov chain defined by Algorithm 1 operates on a finite state space $\mathcal{S}$ and satisfies irreducibility, aperiodicity, and uniformly ergodicity. By the fundamental theorem of Markov chains, these properties guarantee existence of a unique stationary distribution $\mathcal{P}'$ on $\mathcal{S}$, and geometric convergence in total variation:

$$\|\mu_t - \mathcal{P}'\|_{\text{TV}} \leq M \gamma^t$$

for some positive constants $M > 0$ and $0 < \gamma < 1$, where $\mu_t$ denotes the distribution at time $t$. This completes the proof of theorem. $\qquad\square$

*Proof of Theorem 2.* From Lemma 1, the stationary density is $p'(\tilde{z}) = \frac{1}{Z}p(\tilde{z})/w(\tilde{z}, \alpha)$. By $p(\tilde{z}) = (1-\theta)f(\tilde{z}) + \theta q(\tilde{z})$, the normalizing constant $Z$ can be written as

$$Z = (1-\theta)\int \frac{f(z)}{w(z,\alpha)}dz + \theta\int \frac{q(z)}{w(z,\alpha)}dz = (1-\theta)S_{\mathcal{F}} + \theta S_{\mathcal{Q}}.$$

The total probability mass assigned to the contamination distribution $\mathcal{Q}$ in the stationary distribution is

$$\theta' = \int \frac{1}{Z}\frac{\theta q(\tilde{z})}{w(\tilde{z},\alpha)}d\tilde{z} = \frac{\theta}{Z}S_{\mathcal{Q}} = \frac{\theta S_{\mathcal{Q}}}{(1-\theta)S_{\mathcal{F}} + \theta S_{\mathcal{Q}}}.$$

The condition for contamination reduction $\theta' < \theta$ simplifies to

$$\frac{\theta S_{\mathcal{Q}}}{(1-\theta)S_{\mathcal{F}} + \theta S_{\mathcal{Q}}} < \theta \iff S_{\mathcal{Q}} < (1-\theta)S_{\mathcal{F}} + \theta S_{\mathcal{Q}} \quad (\text{since } Z > 0, \theta > 0)$$

$$\iff (1-\theta)S_{\mathcal{Q}} < (1-\theta)S_{\mathcal{F}}$$

$$\iff S_{\mathcal{Q}} < S_{\mathcal{F}}.$$

$\qquad\square$

## C.2 PROOF OF ASYMPTOTIC PROPERTIES

**Lemma 2.** *Suppose that Condition 1 hold. If the tuning parameter $\lambda$ satisfies $(-\ln \lambda)^{\frac{1}{2}}/(n\omega) = o(1)$, and $\omega$ is fixed or changes with $n$, we have*

$$\|f_{\mathbb{S},\lambda} - f_{\mathcal{P}',\lambda}\|_0^2 = O_p\left(\|\tilde{f} - f_{\mathcal{P}',\lambda}\|_0^2\right).$$

*Proof of Lemma 2.* Note that $S_{n\lambda}(f) = -\frac{2}{n}\sum_{i=1}^n (\widetilde{y}_i - f(\widetilde{\mathbf{x}}_i))H_{\omega_{\widetilde{\mathbf{x}}_i}} + 2\lambda f$, and $DS_{n\lambda}(f)\Delta f = \frac{2}{n}\sum_{i=1}^n \Delta f(\widetilde{\mathbf{x}}_i)H_{\omega_{\widetilde{\mathbf{x}}_i}} + 2\lambda\Delta f$. Then, we have

$$S_{n\lambda}(f_{\mathbb{S},\lambda}) - S_{n\lambda}(f_{\mathcal{P}',\lambda}) = \frac{2}{n}\sum_{i=1}^n \{f_{\mathbb{S},\lambda}(\widetilde{\mathbf{x}}_i) - f_{\mathcal{P}',\lambda}(\widetilde{\mathbf{x}}_i)\}H_{\omega_{\widetilde{\mathbf{x}}_i}} + 2\lambda(f_{\mathbb{S},\lambda} - f_{\mathcal{P}',\lambda})$$

$$= DS_{n\lambda}(f_{\mathcal{P}',\lambda})(f_{\mathbb{S},\lambda} - f_{\mathcal{P}',\lambda}).$$

By the definition of $\tilde{f}$: $\tilde{f} - f_{\mathcal{P}',\lambda} = -DS_\lambda^{-1}(f_{\mathcal{P}',\lambda})S_{n,\lambda}(f_{\mathcal{P}',\lambda})$, we obtain $S_{n\lambda}(f_{\mathcal{P}',\lambda}) = DS_\lambda(f_{\mathcal{P}',\lambda})(f_{\mathcal{P}',\lambda} - \tilde{f})$. Since $f_{\mathbb{S},\lambda}$ is the optimal solution of $\mathcal{R}_{\mathbb{S}}(f) + \lambda J(f)$, we have $S_{n\lambda}(f_{\mathbb{S},\lambda}) = 0$ and

$$DS_\lambda(f_{\mathcal{P}',\lambda})(f_{\mathbb{S},\lambda} - \tilde{f}) = DS_\lambda(f_{\mathcal{P}',\lambda})(f_{\mathbb{S},\lambda} - f_{\mathcal{P}',\lambda}) + DS_\lambda(f_{\mathcal{P}',\lambda})(f_{\mathcal{P}',\lambda} - \tilde{f})$$

$$= DS_\lambda(f_{\mathcal{P}',\lambda})(f_{\mathbb{S},\lambda} - f_{\mathcal{P}',\lambda}) - DS_{n\lambda}(f_{\mathcal{P}',\lambda})(f_{\mathbb{S},\lambda} - f_{\mathcal{P}',\lambda}).$$

Combining the above equations leads to

$$\mathbb{E}_{p'}\|f_{\mathbb{S},\lambda} - \tilde{f}\|_0^2 = \mathbb{E}_{p'}\left\{\left\|\left(DS_\lambda(f_{\mathcal{P}',\lambda})\right)^{-1}\left[DS_\lambda(f_{\mathcal{P}',\lambda})(f_{\mathbb{S},\lambda} - f_{\mathcal{P}',\lambda})\right.\right.\right.$$

$$\left.\left.\left. - DS_{n\lambda}(f_{\mathcal{P}',\lambda})(f_{\mathbb{S},\lambda} - f_{\mathcal{P}',\lambda})\right]\right\|_0^2\right\}$$

$$= \mathbb{E}_{p'}\left\{\sum_{k=0}^\infty \left\langle \left(DS_\lambda(f_{\mathcal{P}',\lambda})\right)^{-1}\left[DS_\lambda(f_{\mathcal{P}',\lambda})(f_{\mathbb{S},\lambda} - f_{\mathcal{P}',\lambda})\right.\right.\right.$$

$$\left.\left.\left. - DS_{n\lambda}(f_{\mathcal{P}',\lambda})(f_{\mathbb{S},\lambda} - f_{\mathcal{P}',\lambda})\right], \phi_k\right\rangle_0^2\right\}$$

$$= \mathbb{E}_{p'}\left\{\sum_{k=0}^\infty \left(1 + \frac{\lambda}{\lambda_k}\right)^{-2} \left\langle \left(DS_\lambda(f_{\mathcal{P}',\lambda})\right)^{-1}\left[DS_\lambda(f_{\mathcal{P}',\lambda})(f_{\mathbb{S},\lambda} - f_{\mathcal{P}',\lambda})\right.\right.\right.$$

$$\left.\left.\left. - DS_{n\lambda}(f_{\mathcal{P}',\lambda})(f_{\mathbb{S},\lambda} - f_{\mathcal{P}',\lambda})\right], \phi_k\right\rangle_\lambda^2\right\}$$

$$= \frac{1}{4}\mathbb{E}_{p'}\left\{\sum_{k=0}^\infty \left(1 + \frac{\lambda}{\lambda_k}\right)^{-2} \left\langle DS_\lambda(f_{\mathcal{P}',\lambda})(f_{\mathbb{S},\lambda} - f_{\mathcal{P}',\lambda})\right.\right.$$

$$\left.\left. - DS_{n\lambda}(f_{\mathcal{P}',\lambda})(f_{\mathbb{S},\lambda} - f_{\mathcal{P}',\lambda}), \phi_k\right\rangle_{\mathcal{H}_\omega}^2\right\}$$

$$= \frac{1}{4}\sum_{k=0}^\infty \left(1 + \frac{\lambda}{\lambda_k}\right)^{-2} \mathbb{E}_{p'}\left\{\left[\frac{2}{n}\sum_{i=1}^n \left(f_{\mathbb{S},\lambda}(\widetilde{\mathbf{x}}_i) - f_{\mathcal{P}',\lambda}(\widetilde{\mathbf{x}}_i)\right)\phi_k(\widetilde{\mathbf{x}}_i)\right.\right.$$

$$\left.\left. - 2\mathbb{E}_{p'}(f_{\mathbb{S},\lambda}(\widetilde{\mathbf{x}}) - f_{\mathcal{P}',\lambda}(\widetilde{\mathbf{x}}))\phi_k(\widetilde{\mathbf{x}})\right]^2\right\}.$$

Let $W_k(\widetilde{\mathbf{x}}) = (f_{\mathbb{S},\lambda}(\widetilde{\mathbf{x}}) - f_{\mathcal{P}',\lambda}(\widetilde{\mathbf{x}}))\phi_k(\widetilde{\mathbf{x}})$, $\mu_k = \mathbb{E}_{p'}[W_k(\widetilde{\mathbf{x}})]$, $W_k(\widetilde{\mathbf{x}}_i) = \{f_{\mathbb{S},\lambda}(\widetilde{\mathbf{x}}_i) - f_{\mathcal{P}',\lambda}(\widetilde{\mathbf{x}}_i)\}\phi_k(\widetilde{\mathbf{x}}_i)$, and $\overline{W}_{n,k} = \frac{1}{n}\sum_{i=1}^n W_k(\widetilde{\mathbf{x}}_i)$. Thus, we have

$$\mathbb{E}_{p'}\left\|f_{\mathbb{S},\lambda} - \tilde{f}\right\|_0^2 = \sum_{k=0}^\infty \left(1 + \frac{\lambda}{\lambda_k}\right)^{-2} \mathbb{E}_{p'}\left\{\left[\frac{1}{n}\sum_{i=1}^n W_k(\widetilde{\mathbf{x}}_i) - \mathbb{E}_{p'}[W_k(\widetilde{\mathbf{x}})]\right]^2\right\}$$

$$= \sum_{k=0}^\infty \left(1 + \frac{\lambda}{\lambda_k}\right)^{-2} \mathbb{E}_{p'}\left[(\mu_k - \overline{W}_{n,k})^2\right] = \sum_{k=0}^\infty \left(1 + \frac{\lambda}{\lambda_k}\right)^{-2} \text{Var}(\overline{W}_{n,k}).$$

Note that $\{\widetilde{\mathbf{x}}_i\}_{i=1}^n$ is a u.e.M.c sample. Uniform ergodicity of the Markov chain is equivalent to uniform $\phi$-mixing with a geometric rate (Jones, 2004), i.e., there exist constants $C_\phi > 0$ and $\mathcal{F}_\phi \in [0,1)$ such that $\phi(n) \leq C_\phi \mathcal{F}_\phi^n$ for every $n \geq 1$. Since $\{W_k(\widetilde{\mathbf{x}}_i)\}_{i=1}^n$ is stationary under $\mathcal{P}'$, we have

$$\text{Var}(\overline{W}_{n,k}) = \frac{1}{n^2}\sum_{i=1}^n \sum_{j=1}^n \text{Cov}(W_k(\widetilde{\mathbf{x}}_i), W_k(\widetilde{\mathbf{x}}_j))$$

$$= \frac{1}{n^2}\left(n\gamma_W(0) + 2\sum_{h=1}^{n-1}(n-h)\gamma_W(h)\right)$$

$$= \frac{1}{n}\gamma_W(0) + \frac{2}{n}\sum_{h=1}^{n-1}\left(1 - \frac{h}{n}\right)\gamma_W(h),$$

where $\gamma_W(h) = \text{Cov}(W_k(\widetilde{\mathbf{x}}_i), W_k(\widetilde{\mathbf{x}}_{i+h}))$ and $\gamma_W(0) = \text{Var}(W_k(\widetilde{\mathbf{x}}_1)) = E_{p'}\{(W_k(\widetilde{\mathbf{x}}_1) - \mu_k)^2\}$. By the $\phi$-mixing covariance inequality (Doukhan, 1995), we obtain $|\gamma_W(h)| \leq 2\|W_k(\widetilde{\mathbf{x}})\|_2^2\sqrt{\phi(n)}$. Due to $\sum_n \sqrt{\phi(n)} < \infty$, we have $\text{Var}(\overline{W}_{n,k}) \leq \frac{1}{n}C_1\|W_k(\widetilde{\mathbf{x}})\|_2^2 = \frac{C_1}{n}\mathbb{E}_{p'}[W_k^2(\widetilde{\mathbf{x}})]$. Thus, we

obtain

$$
\begin{aligned}
\mathbb{E}_{p'}\left\|f_{\mathbb{S},\lambda} - \tilde{f}\right\|_0^2 &= \sum_{k=0}^{\infty}\left(1 + \frac{\lambda}{\lambda_k}\right)^{-2}\mathrm{Var}(\overline{W}_{n,k}) \leq \frac{C_1}{n}\sum_{k=0}^{\infty}\left(1 + \frac{\lambda}{\lambda_k}\right)^{-2}\mathbb{E}_{p'}[W_k^2(\widetilde{\mathbf{x}})] \\
&= \frac{C_1}{n}\sum_{k=0}^{\infty}\left(1 + \frac{\lambda}{\lambda_k}\right)^{-2}\mathbb{E}_{p'}\left[(f_{\mathbb{S},\lambda}(\widetilde{\mathbf{x}}) - f_{\mathcal{P}',\lambda}(\widetilde{\mathbf{x}}))\,\phi_k(\widetilde{\mathbf{x}})\right]^2 \\
&\leq \frac{C_2}{n}\|f_{\mathbb{S},\lambda} - f_{\mathcal{P}',\lambda}\|_0^2\sum_{k=0}^{\infty}\left(1 + \frac{\lambda}{\lambda_k}\right)^{-2} \\
&\leq \frac{C_2(-\ln\lambda)^{\frac{1}{2}}}{n\omega}\|f_{\mathbb{S},\lambda} - f_{\mathcal{P}',\lambda}\|_0^2 \\
&= o\left(\|f_{\mathbb{S},\lambda} - f_{\mathcal{P}',\lambda}\|_0^2\right).
\end{aligned}
$$

The second inequality holds since $\phi_k(\widetilde{\mathbf{x}})$ is bounded for any $k$. The third inequality is given by Lemma 3.3 of Zeng (2019). Combining the above equations yields

$$
\|\tilde{f} - f_{\mathcal{P}',\lambda}\|_0^2 \geq \|f_{\mathbb{S},\lambda} - f_{\mathcal{P}',\lambda}\|_0^2 - \|f_{\mathbb{S},\lambda} - \tilde{f}\|_0^2 = (1 - o_p(1))\,\|f_{\mathbb{S},\lambda} - f_{\mathcal{P}',\lambda}\|_0^2.
$$

Thus, we obtain $\|f_{\mathbb{S},\lambda} - f_{\mathcal{P}',\lambda}\|_0^2 = O_p\left(\|\tilde{f} - f_{\mathcal{P}',\lambda}\|_0^2\right)$. $\qquad\square$

**Lemma 3.** *Suppose that Condition 1 hold. If the tuning parameters $\lambda$ and $\omega$ satisfy $\lambda = o(1)$ and $(-\ln\lambda)^{\frac{1}{2}}/(n\omega) = o(1)$, respectively, we have $\|\tilde{f} - f_{\mathcal{P}',\lambda}\|_0^2 = O_p\left((-\ln\lambda)^{\frac{1}{2}}/(n\omega)\right)$.*

*Proof of Lemma 3.* Since $S_\lambda(f_{\mathcal{P}',\lambda}) = 0$, similarly to the proof of Lemma 2, we obtain

$$
\begin{aligned}
\mathbb{E}_{p'}\langle S_{n\lambda}(f_{\mathcal{P}',\lambda}), \phi_k\rangle_{\mathcal{H}_\omega}^2 &= \mathbb{E}_{p'}\langle S_{n\lambda}(f_{\mathcal{P}',\lambda}) - S_\lambda(f_{\mathcal{P}',\lambda}), \phi_k\rangle_{\mathcal{H}_\omega}^2 \\
&= \mathbb{E}_{p'}\left\{\frac{2}{n}\sum_{i=1}^{n}(\widetilde{y}_i - f_{\mathcal{P}',\lambda}(\widetilde{\mathbf{x}}_i))\,\phi_k(\widetilde{\mathbf{x}}_i) - 2\mathbb{E}_{p'}(\widetilde{y} - f_{\mathcal{P}',\lambda}(\widetilde{\mathbf{x}})\phi_k(\widetilde{\mathbf{x}}))\right\}^2 \\
&= 4\,\mathrm{Var}\left(\frac{1}{n}\sum_{i=1}^{n}(\widetilde{y}_i - f_{\mathcal{P}',\lambda}(\widetilde{\mathbf{x}}_i))\phi_k(\widetilde{\mathbf{x}}_i)\right) \\
&\leq \frac{4C_1}{n}\mathbb{E}_{p'}\left[(\widetilde{y} - f_{\mathcal{P}',\lambda}(\widetilde{\mathbf{x}}))\phi_k(\widetilde{\mathbf{x}})\right]^2 \\
&= \frac{4C_1}{n}\mathbb{E}_{p'}\left[(\epsilon + f_{\mathcal{P}'}(\widetilde{\mathbf{x}}) - f_{\mathcal{P}',\lambda}(\widetilde{\mathbf{x}}))\phi_k(\widetilde{\mathbf{x}})\right]^2 \\
&= \frac{4C_1}{n}\sigma^2\mathbb{E}_{p'}\phi_k^2 + \frac{4C_1}{n}\mathbb{E}_{p'}\left[(f_{\mathcal{P}'}(\widetilde{\mathbf{x}}) - f_{\mathcal{P}',\lambda}(\widetilde{\mathbf{x}}))\phi_k(\widetilde{\mathbf{x}})\right]^2.
\end{aligned}
$$

By proof of Lemma 2, we have

$$\mathbb{E}_{p'}\|\tilde{f} - f_{\mathcal{P}',\lambda}\|_0^2 = \mathbb{E}_{p'} \left\| \left(DS_\lambda(f_{\mathcal{P}',\lambda})\right)^{-1} S_{n\lambda}(f_{\mathcal{P}',\lambda}) \right\|_0^2$$

$$= \mathbb{E}_{p'} \left\{ \sum_{k=0}^{\infty} \left\langle \left(DS_\lambda(f_{\mathcal{P}',\lambda})\right)^{-1} S_{n\lambda}(f_{\mathcal{P}',\lambda}), \phi_k \right\rangle_0^2 \right\}$$

$$= \frac{1}{4}\mathbb{E}_{p'} \left\{ \sum_{k=0}^{\infty} \left(1 + \frac{\lambda}{\lambda_k}\right)^{-2} \left\langle \left(\frac{1}{2}DS_\lambda(f_{\mathcal{P}',\lambda})\right)^{-1} S_{n\lambda}(f_{\mathcal{P}',\lambda}), \phi_k \right\rangle_\lambda^2 \right\}$$

$$= \frac{1}{4}\mathbb{E}_{p'} \left\{ \sum_{k=0}^{\infty} \left(1 + \frac{\lambda}{\lambda_k}\right)^{-2} \langle S_{n\lambda}(f_{\mathcal{P}',\lambda}), \phi_k \rangle_{\mathcal{H}_\omega}^2 \right\}$$

$$\leq \frac{C}{n} \sum_{k=0}^{\infty} \left(1 + \frac{\lambda}{\lambda_k}\right)^{-2} + \frac{1}{n} \sum_{k=0}^{\infty} \left(1 + \frac{\lambda}{\lambda_k}\right)^{-2}$$

$$\mathbb{E}_{p'} \left[ \left(f_{\mathcal{P}'}(\widetilde{\mathbf{x}}) - f_{\mathcal{P}',\lambda}(\widetilde{\mathbf{x}})\right)^2 \phi_k(\widetilde{\mathbf{x}})^2 \right]$$

$$= \frac{C}{n} \sum_{k=0}^{\infty} \left(1 + \frac{\lambda}{\lambda_k}\right)^{-2} + \frac{C' \|f_{\mathcal{P}'} - f_{\mathcal{P}',\lambda}\|_0^2}{n} \sum_{k=0}^{\infty} \left(1 + \frac{\lambda}{\lambda_k}\right)^{-2}$$

$$= O\left(\frac{(-\ln\lambda)^{\frac{1}{2}}}{n\omega}\right) + O\left(\frac{\lambda(-\ln\lambda)^{\frac{1}{2}}}{n\omega}\right) = O\left(\frac{(-\ln\lambda)^{\frac{1}{2}}}{n\omega}\right).$$

$\square$

*Proof of Theorem 3.* We adopt the commonly used technique in studying consistency like Zeng & Xia (2019). Different from Zeng & Xia (2019), we consider a u.e.M.c samples rather than i.i.d samples. Let $f_{\mathcal{P}'}(\mathbf{x}) = \sum_{k=0}^{\infty} f_{\mathcal{P}',k}\phi_k(\mathbf{x})$, $f(\mathbf{x}) = \sum_{k=0}^{\infty} f_k\phi_k(\mathbf{x})$ and $f_{\mathcal{P}',\lambda}(\mathbf{x}) = \sum_{k=0}^{\infty} f_{\mathcal{P}',\lambda,k}\phi_k(\mathbf{x})$. It follows from Theorem 2 of Zeng & Xia (2019) that

$$\mathcal{R}_{\mathcal{P}',\lambda}(f) = \mathbb{E}_{p'}\left[(\widetilde{y} - f(\widetilde{\mathbf{x}}))^2\right] + \lambda J(f)$$

$$= \mathbb{E}_{p'}\left[(\epsilon + f_{\mathcal{P}'}(\widetilde{\mathbf{x}}) - f(\widetilde{\mathbf{x}}))^2\right] + \lambda J(f)$$

$$= \sigma^2 + \sum_{k=0}^{\infty}(f_k - f_{\mathcal{P}',k})^2 + \lambda \sum_{k=0}^{\infty}\frac{f_k^2}{\lambda_k}.$$

As $f_{\mathcal{P}',\lambda}$ is the minimizer of $\mathcal{R}_{\mathcal{P}',\lambda}(f)$, we have $f_{\mathcal{P}',\lambda,k} = f_{\mathcal{P}',k}\lambda_k/(\lambda + \lambda_k)$. Combining the equations yields

$$\|f_{\mathcal{P}',\lambda} - f_{\mathcal{P}'}\|_0^2 = \sum_{k=0}^{\infty}\frac{\lambda^2}{(\lambda + \lambda_k)^2}f_{\mathcal{P}',k}^2 = \sum_{k=0}^{\infty}\frac{\lambda^2\lambda_k}{(\lambda + \lambda_k)^2}\frac{f_{\mathcal{P}',k}^2}{\lambda_k}$$

$$\leq \sup_k \frac{\lambda^2\lambda_k}{(\lambda + \lambda_k)^2} J(f_{\mathcal{P}'})$$

$$\leq \lambda^2 \sup_{x>0} \frac{x}{\left(x^{\frac{1}{2}} + \lambda x^{-\frac{1}{2}}\right)^2} J(f_{\mathcal{P}'})$$

$$= \frac{\lambda}{4} J(f_{\mathcal{P}'}) = O(\lambda).$$

By Lemma 2 and Lemma 3, we have

$$\|f_{\mathbb{S},\lambda} - f_{\mathcal{P}',\lambda}\|_0^2 = O_p\left(\frac{(-\ln\lambda)^{\frac{1}{2}}}{n\omega}\right),$$

It follows that

$$\|f_{\mathbb{S},\lambda} - f_{\mathcal{P}'}\|_0^2 = O_p(\lambda) + O_p\left(\frac{(-\ln\lambda)^{\frac{1}{2}}}{n\omega}\right).$$

When $\omega$ is fixed, we choose $\lambda \sim (-\ln n)^{\frac{1}{2}}/n$, yielding $\lambda = o(1)$ and $(-\ln \lambda)^{\frac{1}{2}}/n = o(1)$ as $n \to \infty$. When Assumptions of Lemmas 2-3 hold, we have

$$\|f_{\mathbb{S},\lambda} - f_{\mathcal{P}'}\|_0^2 = O_p(\lambda) + O_p\left(\frac{(-\ln \lambda)^{\frac{1}{2}}}{n\omega}\right) = O_p\left(\frac{(\ln n)^{\frac{1}{2}}}{n}\right).$$

$\square$

*Proof of Theorem 4.* Following the proof of Theorem 3, we can show

$$\mathbb{E}_{p'}\|f_{\mathbb{S},\lambda} - f_{\mathcal{P}',\lambda}\|_0^2 = O\left(\frac{(-\ln \lambda)^{\frac{1}{2}}}{n\omega}\right).$$

Thus, we consider $f_{\mathcal{P}',\lambda} - f_{\mathcal{P}'}$. Since $f_{\mathcal{P}'} \in \mathcal{H}_{[-\pi,\pi]}^m$, we get $f_{\mathcal{P}'} = \sum_{k=0}^{\infty} f_{\mathcal{P}',k}\phi_k(\mathbf{x})$ with $\sum_{k=0}^{\infty} f_{\mathcal{P}',k}^2/\mathcal{F}_k < \infty$. Similarly to the proof of Theorem 3, we can show $f_{\mathcal{P}',\lambda,k} = f_{\mathcal{P}',k}\lambda_k/\lambda+\lambda_k$. Thus, we obtain

$$\|f_{\mathcal{P}',\lambda} - f_{\mathcal{P}'}\|_0^2 = \sum_{k=0}^{\infty} \frac{\lambda^2}{(\lambda + \lambda_k)^2} f_{\mathcal{P}',k}^2 = \sum_{k=0}^{\infty} \frac{\lambda^2 \mathcal{F}_k}{(\lambda + \lambda_k)^2} \frac{f_{\mathcal{P}',k}^2}{\mathcal{F}_k}$$

$$\leq C \sup_k \frac{\lambda^2 \mathcal{F}_k}{(\lambda + \lambda_k)^2}$$

$$\leq C \sup_{s>0} \frac{\lambda^2 \left(s^{2m} + 1\right)^{-1}}{\left(\lambda + e^{-\frac{s^2\omega^2}{2}}\right)^2}.$$

Now we find the maximum value of $q(x) = \lambda^2 \left(x^{2m} + 1\right)^{-1} / \left(\lambda + e^{-\frac{x^2\alpha^2}{2}}\right)^2$ with $x > 0$. On the boundary, $q(0) = O\left(\lambda^2\right)$ and $q(\infty) = 0$. For the inner points, it follows from $q'(x) = 0$ that $\omega^2 \left(x^2 + x^{-(2m-2)}\right) = m\left(1 + \lambda e^{\frac{\alpha^2\omega^2}{2}}\right)$ whose solution is denoted as $\hat{x}$. Since $q'(x) > 0$ as $\lambda \to 0$, we have $\hat{x} \to \infty$ as $\lambda \to 0$. As a result, we obtain $\omega^2\hat{x}^2 \sim \lambda e^{\frac{\alpha^2\omega^2}{2}}$. Then, we have $\omega^2\hat{x}^2 \sim -\ln \lambda$, and $q(\hat{x}) = O\left(\frac{\omega^{2m}}{(-\ln \lambda)^m}\right)$. When $\lambda = o(1)$, we have $\lambda^2 = o\left(\omega^{2m}/(-\ln \lambda)^m\right)$. Thus, we obtain

$$\sup_{s>0} \frac{\lambda^2 \left(s^{2m} + 1\right)^{-1}}{\left(\lambda + e^{-\frac{x^2\omega^2}{2}}\right)^2} = O\left(\frac{\omega^{2m}}{(-\ln \lambda)^m}\right).$$

It follows from $\mathbb{E}_{p'}\|f_{\mathbb{S},\lambda} - f_{\mathcal{P}',\lambda}\|_0^2 = O\left(\frac{(-\ln \lambda)^{\frac{1}{2}}}{n\omega}\right)$, and $(-\ln \lambda)^{\frac{1}{2}}/\omega \sim n^{\frac{1}{2m+1}}$ that

$$\|f_{\mathbb{S},\lambda} - f_{\mathcal{P}'}\|_0^2 = O_p\left(n^{-\frac{2m}{2m+1}}\right).$$

$\square$

*Proof of Theorem 5.* Note that for any $f(\mathbf{x}) = \sum_{k=0}^{\infty} f_k\phi_k(\mathbf{x})$, we have

$$\begin{aligned}
\|f\|_\lambda^2 &= \langle f, f \rangle_\lambda^2 = \|f\|_0^2 + \lambda\|f\|_{\mathcal{H}_\omega}^2 \\
&= \sum_{k=0}^{\infty} \left(1 + \frac{\lambda}{\lambda_k}\right) f_k^2 = \sum_{k=0}^{\infty} \left(1 + \frac{\lambda}{\lambda_k}\right)^{-1} \langle f, \phi_k \rangle_\lambda^2.
\end{aligned}$$

Then, we obtain

$$\mathbb{E}_{p'}\left\|f_{\mathbb{S},\lambda} - f_{\mathcal{P}'} + \left(DS_\lambda\left(f_{\mathcal{P}'}\right)\right)^{-1} S_{n\lambda}\left(f_{\mathcal{P}'}\right)\right\|_\lambda^2$$

$$=\mathbb{E}_{p'}\left\{\|\left(DS_\lambda\left(f_{\mathcal{P}'}\right)\right)^{-1}\left[DS_\lambda\left(f_{\mathcal{P}'}\right)\left(f_{\mathbb{S},\lambda} - f_{\mathcal{P}'}\right) - DS_{n\lambda}\left(f_{\mathcal{P}'}\right)\left(f_{\mathbb{S},\lambda} - f_{\mathcal{P}'}\right)\right]\|_\lambda^2\right\}$$

$$=\mathbb{E}_{p'}\left\{\sum_{k=0}^\infty\left(1+\frac{\lambda}{\lambda_k}\right)^{-1}\left(\left(DS_\lambda\left(f_{\mathcal{P}'}\right)\right)^{-1}\left[DS_\lambda\left(f_{\mathcal{P}'}\right)\left(f_{\mathbb{S},\lambda} - f_{\mathcal{P}'}\right)\right.\right.$$

$$\left.\left.-DS_{n\lambda}\left(f_{\mathcal{P}'}\right)\left(f_{\mathbb{S},\lambda} - f_{\mathcal{P}'}\right)\right],\phi_k\right)_\lambda^2\right\}$$

$$=\frac{1}{4}\mathbb{E}_{p'}\left\{\sum_{k=0}^\infty\left(1+\frac{\lambda}{\lambda_k}\right)^{-1}\langle DS_\lambda\left(f_{\mathcal{P}'}\right)\left(f_{\mathbb{S},\lambda} - f_{\mathcal{P}'}\right) - DS_{n\lambda}\left(f_{\mathcal{P}'}\right)\left(f_{\mathbb{S},\lambda} - f_{\mathcal{P}'}\right),\phi_k\rangle_{\mathcal{H}_\omega}^2\right\}$$

$$=\sum_{k=0}^\infty\left(1+\frac{\lambda}{\lambda_k}\right)^{-1}\mathrm{Var}\left[\frac{1}{n}\sum_{i=1}^n\left(f_{\mathbb{S},\lambda}(\widetilde{\mathbf{x}}_i) - f_{\mathcal{P}'}(\widetilde{\mathbf{x}}_i)\right)\phi_k(\widetilde{\mathbf{x}}_i)\right]$$

$$\leq\frac{C_1}{n}\sum_{k=0}^\infty\left(1+\frac{\lambda}{\lambda_k}\right)^{-1}\mathbb{E}_{p'}\left[\left(f_{\mathbb{S},\lambda}(\widetilde{\mathbf{x}}) - f_{\mathcal{P}'}(\widetilde{\mathbf{x}})\right)^2\phi_k(\widetilde{\mathbf{x}})^2\right]$$

$$\leq\frac{C_2}{n}\|f_{\mathbb{S},\lambda}(\widetilde{\mathbf{x}}) - f_{\mathcal{P}'}(\widetilde{\mathbf{x}})\|_0^2\sum_{k=0}^\infty\left(1+\frac{\lambda}{\lambda_k}\right)^{-1}$$

$$=O\left(\frac{\ln n}{n^2}\right).$$

$\square$

*Proof of Theorem 6.* From Theorem 5, we have the FBR:

$$f_{\mathbb{S},\lambda} - f_{\mathcal{P}'} = -\left(DS_\lambda\left(f_{\mathcal{P}'}\right)\right)^{-1} S_{n\lambda}\left(f_{\mathcal{P}'}\right) + \Delta',$$

where $\|\Delta'\|_\lambda^2 = O_p(\frac{\ln n}{n^2})$. For a fixed $\widetilde{\mathbf{x}}_0$, $f(\widetilde{\mathbf{x}}_0) = \langle f, R_{\lambda_{\widetilde{\mathbf{x}}_0}}\rangle_\lambda$ (Zeng, 2019). Then, we have

$$f_{\mathbb{S},\lambda}(\widetilde{\mathbf{x}}_0) - f_{\mathcal{P}'}(\widetilde{\mathbf{x}}_0) = -\left\langle\left(DS_\lambda\left(f_{\mathcal{P}'}\right)\right)^{-1} S_{n\lambda}\left(f_{\mathcal{P}'}\right), R_{\lambda_{\widetilde{\mathbf{x}}_0}}\right\rangle_\lambda + \langle\Delta', R_{\lambda_{\widetilde{\mathbf{x}}_0}}\rangle_\lambda.$$

The second term (remainder) can be bounded using Cauchy-Schwarz in the $\langle\cdot,\cdot\rangle_\lambda$ inner product space, i.e.,

$$|\langle\Delta', R_{\lambda_{\widetilde{\mathbf{x}}_0}}\rangle_\lambda| \leq \|\Delta'\|_\lambda\|R_{\lambda_{\widetilde{\mathbf{x}}_0}}\|_\lambda,$$

It follows from Lemma 3.1 of Zeng (2019) that $R_{\lambda_{\widetilde{\mathbf{x}}_0}} = (\frac{1}{2}DS_\lambda(f_{\mathcal{P}'}))^{-1}H_{\omega_{\widetilde{\mathbf{x}}_0}}$. Using the identity $\langle(\frac{1}{2}DS_\lambda(f_{\mathcal{P}'}))^{-1}f_1, f_2\rangle_\lambda = \langle f_1, f_2\rangle_{\mathcal{H}_\omega}$ and $\langle\phi_k, \phi_j\rangle_{\mathcal{H}_\omega} = \delta_{kj}/\lambda_k$ yields

$$\|R_{\lambda_{\widetilde{\mathbf{x}}_0}}\|_\lambda^2 = \langle R_{\lambda_{\widetilde{\mathbf{x}}_0}}, R_{\lambda_{\widetilde{\mathbf{x}}_0}}\rangle_\lambda = \langle(\frac{1}{2}DS_\lambda(f_{\mathcal{P}'}))^{-1}H_{\omega_{\widetilde{\mathbf{x}}_0}}, R_{\lambda_{\widetilde{\mathbf{x}}_0}}\rangle_\lambda$$

$$= \langle H_{\omega_{\widetilde{\mathbf{x}}_0}}, R_{\lambda_{\widetilde{\mathbf{x}}_0}}\rangle_{\mathcal{H}_\omega} = \left\langle\sum_{k=0}^\infty\lambda_k\phi_k(\widetilde{\mathbf{x}}_0)\phi_k(\cdot), \sum_{j=0}^\infty(1+\frac{\lambda}{\lambda_j})^{-1}\phi_j(\widetilde{\mathbf{x}}_0)\phi_j(\cdot)\right\rangle_{\mathcal{H}_\omega}$$

$$= \sum_{k=0}^\infty\sum_{j=0}^\infty\lambda_k\phi_k(\widetilde{\mathbf{x}}_0)(1+\frac{\lambda}{\lambda_j})^{-1}\phi_j(\widetilde{\mathbf{x}}_0)\frac{\delta_{kj}}{\lambda_k}$$

$$= \sum_{k=0}^\infty(1+\frac{\lambda}{\lambda_k})^{-1}\phi_k^2(\widetilde{\mathbf{x}}_0) = O(\frac{(\ln n)^{\frac{1}{2}}}{\omega}).$$

Now bound the remainder term, scaled by the normalization factor. Thus, we have

$$\left|\sqrt{\frac{n}{(\ln n)^{1/2}}}\langle\Delta', R_{\lambda_{\widetilde{\mathbf{x}}_0}}\rangle_\lambda\right| \leq \sqrt{\frac{n}{(\ln n)^{1/2}}}\|\Delta'\|_\lambda\|R_{\lambda_{\widetilde{\mathbf{x}}_0}}\|_\lambda$$

$$= \sqrt{\frac{n}{(\ln n)^{1/2}}}O_p\left(\sqrt{\frac{\ln n}{n^2}}\right)O\left(\sqrt{\frac{(\ln n)^{1/2}}{\omega}}\right)$$

$$= o_p(1).$$

By Slutsky's theorem, the asymptotic distribution of $\sqrt{\frac{n}{(\ln n)^{1/2}}}(f_{\mathbb{S},\lambda}(\widetilde{\mathbf{x}}_0) - f_{\mathcal{P}'}(\widetilde{\mathbf{x}}_0))$ is the same as that of

$$-\sqrt{\frac{n}{(\ln n)^{1/2}}}\left\langle (DS_\lambda(f_{\mathcal{P}'}))^{-1} S_{n\lambda}(f_{\mathcal{P}'}), R_{\lambda_{\widetilde{\mathbf{x}}_0}}\right\rangle_\lambda,$$

which implies that we only analyze the leading term

$$-\sqrt{n/(\ln n)^{1/2}}\left\langle (DS_\lambda(f_{\mathcal{P}'}))^{-1} S_{n\lambda}(f_{\mathcal{P}'}), R_{\lambda_{\widetilde{\mathbf{x}}_0}}\right\rangle_\lambda.$$

Using the property $\langle (DS_\lambda(f_{\mathcal{P}'}))^{-1} f_1, f_2\rangle_\lambda = \langle \frac{1}{2}f_1, f_2\rangle_{\mathcal{H}_\omega}$, we obtain

$$-\left\langle (DS_\lambda(f_{\mathcal{P}'}))^{-1} S_{n\lambda}(f_{\mathcal{P}'}), R_{\lambda_{\widetilde{\mathbf{x}}_0}}\right\rangle_\lambda = -\left\langle \frac{1}{2} S_{n\lambda}(f_{\mathcal{P}'}), R_{\lambda_{\widetilde{\mathbf{x}}_0}}\right\rangle_{\mathcal{H}_\omega}$$

$$= -\left\langle -\frac{1}{n}\sum_{i=1}^n \epsilon_i H_{\omega_{\widetilde{\mathbf{x}}_i}} + \lambda f_{\mathcal{P}'}, R_{\lambda_{\widetilde{\mathbf{x}}_0}}\right\rangle_{\mathcal{H}_\omega}$$

$$= \frac{1}{n}\sum_{i=1}^n \epsilon_i \langle H_{\omega_{\widetilde{\mathbf{x}}_i}}, R_{\lambda_{\widetilde{\mathbf{x}}_0}}\rangle_{\mathcal{H}_\omega} - \lambda\langle f_{\mathcal{P}'}, R_{\lambda_{\widetilde{\mathbf{x}}_0}}\rangle_{\mathcal{H}_\omega}.$$

For the first term, it follows from $H_{\omega_{\widetilde{\mathbf{x}}_i}}(\cdot) = \sum_k \lambda_k \phi_k(\widetilde{\mathbf{x}}_i)\phi_k(\cdot)$ and $R_{\lambda_{\widetilde{\mathbf{x}}_0}}(\cdot) = \sum_j (1 + \frac{\lambda}{\lambda_j})^{-1}\phi_j(\widetilde{\mathbf{x}}_0)\phi_j(\cdot)$ that

$$\langle H_{\omega_{\widetilde{\mathbf{x}}_i}}, R_{\lambda_{\widetilde{\mathbf{x}}_0}}\rangle_{\mathcal{H}_\omega} = \sum_{k,j} \lambda_k \phi_k(\widetilde{\mathbf{x}}_i)(1 + \frac{\lambda}{\lambda_j})^{-1}\phi_j(\widetilde{\mathbf{x}}_0)\langle \phi_k, \phi_j\rangle_{\mathcal{H}_\omega}$$

$$= \sum_{k,j} \lambda_k \phi_k(\widetilde{\mathbf{x}}_i)(1 + \frac{\lambda}{\lambda_j})^{-1}\phi_j(\widetilde{\mathbf{x}}_0)\frac{\delta_{kj}}{\lambda_k}$$

$$= \sum_k (1 + \frac{\lambda}{\lambda_k})^{-1}\phi_k(\widetilde{\mathbf{x}}_i)\phi_k(\widetilde{\mathbf{x}}_0)$$

$$= R_\lambda(\widetilde{\mathbf{x}}_i, \widetilde{\mathbf{x}}_0).$$

Then, we obtain $\frac{1}{n}\sum_{i=1}^n \epsilon_i \langle H_{\omega_{\widetilde{\mathbf{x}}_i}}, R_{\lambda_{\widetilde{\mathbf{x}}_0}}\rangle_{\mathcal{H}_\omega} = \frac{1}{n}\sum_{i=1}^n \epsilon_i R_\lambda(\widetilde{\mathbf{x}}_i, \widetilde{\mathbf{x}}_0)$.

For the second term $\lambda\langle f_{\mathcal{P}'}, R_{\lambda_{\widetilde{\mathbf{x}}_0}}\rangle_{\mathcal{H}_\omega}$, it follows from $f_{\mathcal{P}'}(\cdot) = \sum_k f_{\mathcal{P}',k}\phi_k(\cdot)$ that

$$\lambda\langle f_{\mathcal{P}'}, R_{\lambda_{\widetilde{\mathbf{x}}_0}}\rangle_{\mathcal{H}_\omega} = \lambda\sum_{k,j} f_{\mathcal{P}',k}(1 + \frac{\lambda}{\lambda_j})^{-1}\phi_j(\widetilde{\mathbf{x}}_0)\langle \phi_k, \phi_j\rangle_{\mathcal{H}_\omega}$$

$$= \lambda\sum_{k,j} f_{\mathcal{P}',k}(1 + \frac{\lambda}{\lambda_j})^{-1}\phi_j(\widetilde{\mathbf{x}}_0)\frac{\delta_{kj}}{\lambda_k}$$

$$= \sum_k \frac{\lambda}{\lambda_k} f_{\mathcal{P}',k}(1 + \frac{\lambda}{\lambda_k})^{-1}\phi_k(\widetilde{\mathbf{x}}_0)$$

$$= \sum_k \frac{\lambda}{\lambda + \lambda_k} f_{\mathcal{P}',k}\phi_k(\widetilde{\mathbf{x}}_0).$$

The leading term is

$$\frac{1}{n}\sum_{i=1}^n \epsilon_i R_\lambda(\widetilde{\mathbf{x}}_i, \widetilde{\mathbf{x}}_0) - \sum_k \frac{\lambda}{\lambda + \lambda_k} f_{\mathcal{P}',k}\phi_k(\widetilde{\mathbf{x}}_0),$$

and the asymptotic distribution is

$$\sqrt{\frac{n}{(\ln n)^{1/2}}} \left( f_{\mathbb{S},\lambda}(\widetilde{\mathbf{x}}_0) - f_{\mathcal{P}'}(\widetilde{\mathbf{x}}_0) - \left[ -\sum_k \frac{\lambda}{\lambda + \lambda_k} f_{\mathcal{P}',k}\phi_k(\widetilde{\mathbf{x}}_0) \right] \right)$$

$$= \sqrt{\frac{n}{(\ln n)^{1/2}}} \left( f_{\mathbb{S},\lambda}(\widetilde{\mathbf{x}}_0) - \left[ \sum_k f_{\mathcal{P}',k}\phi_k(\widetilde{\mathbf{x}}_0) - \sum_k \frac{\lambda}{\lambda + \lambda_k} f_{\mathcal{P}',k}\phi_k(\widetilde{\mathbf{x}}_0) \right] \right)$$

$$= \sqrt{\frac{n}{(\ln n)^{1/2}}} \left( f_{\mathbb{S},\lambda}(\widetilde{\mathbf{x}}_0) - \sum_k \frac{\lambda_k}{\lambda + \lambda_k} f_{\mathcal{P}',k}\phi_k(\widetilde{\mathbf{x}}_0) \right)$$

$$= \sqrt{\frac{n}{(\ln n)^{1/2}}} \left( f_{\mathbb{S},\lambda}(\widetilde{\mathbf{x}}_0) - f^*(\widetilde{\mathbf{x}}_0) \right).$$

Therefore, the asymptotic distribution is determined by the term involving the sum of errors:

$$T_n := \sqrt{\frac{n}{(\ln n)^{1/2}}} \left( \frac{1}{n} \sum_{i=1}^n \epsilon_i R_\lambda(\widetilde{\mathbf{x}}_i, \widetilde{\mathbf{x}}_0) \right) = \frac{1}{\sqrt{n(\ln n)^{1/2}}} \sum_{i=1}^n \epsilon_i R_\lambda(\widetilde{\mathbf{x}}_i, \widetilde{\mathbf{x}}_0).$$

Consider the sequence $V_i = \epsilon_i R_\lambda(\widetilde{\mathbf{x}}_i, \widetilde{\mathbf{x}}_0)$, which is a stationary sequence under $\mathcal{P}'$ and is $\phi$-mixing with $\mathbb{E}_{p'}(V_i) = 0$. It follows from the central limit theorem for $\phi$-mixing sequences (Jones, 2004) that if $\sum \sqrt{\phi(n)} < \infty$ and $\mathbb{E}(V_0^2) < \infty$, $\frac{1}{\sqrt{n}} \sum_{i=1}^n V_i \xrightarrow{d} \mathcal{N}(0, \sigma_{LTV}^2)$, where $\sigma_{LTV}^2 = \text{Var}(V_0) + 2\sum_{j=1}^\infty \mathbb{E}_{p'}(V_0 V_j)$ is the long-term variance. Thus, we have

$$\lim_{n\to\infty} \text{Var}(T_n) = \lim_{n\to\infty} \frac{1}{(\ln n)^{1/2}} \text{Var}\left( \frac{1}{\sqrt{n}} \sum_{i=1}^n V_i \right),$$

$$\text{Cov}(V_0, V_j) = \mathbb{E}_{p'}(V_0 V_j) = \delta_{0j}\sigma^2 \mathbb{E}_{p'}\{R_\lambda(\widetilde{\mathbf{x}}_0, \widetilde{\mathbf{x}})^2\}$$

$$\lim_{n\to\infty} \text{Var}\left( \frac{1}{\sqrt{n}} \sum_{i=1}^n V_i \right) = \sigma_{LTV}^2$$

$$= \sum_{j=-\infty}^\infty \text{Cov}(V_0, V_j) = \text{Cov}(V_0, V_0)$$

$$= \sigma^2 \|R_{\lambda_{\widetilde{\mathbf{x}}_0}}\|_0^2$$

$$= \sigma^2 \sum_k \left( 1 + \frac{\lambda}{\lambda_k} \right)^{-2} \phi_k(\widetilde{\mathbf{x}}_0)^2.$$

Hence, the asymptotic variance of $T_n$ is

$$\lim_{n\to\infty} \text{Var}(T_n) = \lim_{n\to\infty} \frac{\sigma^2}{(\ln n)^{1/2}} \sum_k \left( 1 + \frac{\lambda}{\lambda_k} \right)^{-2} \phi_k(\widetilde{\mathbf{x}}_0)^2.$$

Combining the FBR approximation and the central limit thorem for the leading term leads to

$$\sqrt{\frac{n}{(\ln n)^{\frac{1}{2}}}} \{f_{\mathbb{S},\lambda}(\widetilde{\mathbf{x}}_0) - f^\star(\widetilde{\mathbf{x}}_0)\} \xrightarrow{d} \mathcal{N}\left( 0, \sigma_{\widetilde{\mathbf{x}}_0}^2 \right),$$

where $\sigma_{\widetilde{\mathbf{x}}_0}^2 = \lim_{n\to\infty} \sigma^2 (\ln n)^{-\frac{1}{2}} \sum_{k=0}^\infty (1 + \lambda/\lambda_k)^{-2} \phi_k^2(\widetilde{\mathbf{x}}_0)$.

$\square$

*Proof of Theorem 7.* By the proof of Theorem 6, we obtain

$$\mathbb{E}_{p'}\{\|f_{\mathbb{S},\lambda} - f_{\mathcal{P}'} + (DS_\lambda(f_{\mathcal{P}'}))^{-1} S_{n\lambda}(f_{\mathcal{P}'})\|_\lambda^2\}$$

$$= \sum_{k=0}^\infty \left(1 + \frac{\lambda}{\lambda_k}\right)^{-1} \text{Var}\left\{\frac{1}{n}\sum_{i=1}^n (f_{\mathbb{S},\lambda}(\widetilde{\mathbf{x}}_i) - f_{\mathcal{P}',0}(\widetilde{\mathbf{x}}_i))\phi_k(\widetilde{\mathbf{x}}_i)\right\}$$

$$\leq \frac{C_2}{n}\|f_{\mathbb{S},\lambda}(\widetilde{\mathbf{x}}) - f_{\mathcal{P}',0}(\widetilde{\mathbf{x}})\|_0^2 \sum_{k=0}^\infty \left(1 + \frac{\lambda}{\lambda_k}\right)^{-1}$$

$$= O(\frac{1}{n})O(n^{-\frac{2m}{2m+1}})O\left(\frac{(-\ln\lambda)^{1/2}}{\omega}\right)$$

$$= O\left(n^{-\frac{4m}{2m+1}}\right).$$

Similarly to proof of Theorem 6, we have

$$\left|n^{\frac{m}{2m+1}}\langle\Delta', R_{\lambda_{\widetilde{\mathbf{x}}_0}}\rangle_\lambda\right| \leq n^{\frac{m}{2m+1}}\|\Delta'\|_\lambda\|R_{\lambda_{\widetilde{\mathbf{x}}_0}}\|_\lambda,$$

where

$$\|\Delta'\|_\lambda = \sqrt{O_p(n^{-\frac{4m}{2m+1}})} = O_p(n^{-\frac{2m}{2m+1}}),$$

$$\|R_{\lambda_{\widetilde{\mathbf{x}}_0}}\|_\lambda^2 = \sum_k (1 + \frac{\lambda}{\lambda_k})^{-1}\phi_k^2(\widetilde{\mathbf{x}}_0) = O(\frac{(-\ln\lambda)^{1/2}}{\omega}) = O(n^{\frac{1}{2m+1}}),$$

$$\|R_{\lambda_{\widetilde{\mathbf{x}}_0}}\|_\lambda = O(\sqrt{n^{\frac{1}{2m+1}}}) = O(n^{\frac{1}{2(2m+1)}}).$$

Thus, we have

$$n^{\frac{m}{2m+1}}O_p(n^{-\frac{2m}{2m+1}})O(n^{\frac{1}{2(2m+1)}}) = O_p\left(n^{\frac{m}{2m+1} - \frac{2m}{2m+1} + \frac{1}{2(2m+1)}}\right) = O_p\left(n^{\frac{-2m+1/2}{2m+1}}\right) = o_p(1).$$

For leading term

$$T'_n := n^{\frac{m}{2m+1}}\left(\frac{1}{n}\sum_{i=1}^n \epsilon_i R_\lambda(\widetilde{\mathbf{x}}_i, \widetilde{\mathbf{x}}_0)\right) = \frac{n^{\frac{m}{2m+1}}}{n}\sum_{i=1}^n V_i = n^{-\frac{m+1}{2m+1}}\sum_{i=1}^n V_i,$$

where $V_i = \epsilon_i R_\lambda(\widetilde{\mathbf{x}}_i, \widetilde{\mathbf{x}}_0)$, its variance is

$$\text{Var}(T'_n) = \text{Var}\left(n^{-\frac{m+1}{2m+1}}\sum_{i=1}^n V_i\right) = n^{\frac{1}{2m+1}}\text{Var}\left(\frac{1}{\sqrt{n}}\sum_{i=1}^n V_i\right),$$

and the limitation of $\text{Var}(T'_n)$ is given by

$$\lim_{n\to\infty}\text{Var}(T'_n) = \lim_{n\to\infty}\frac{\sigma^2}{(n^{\frac{1}{2m+1}})}\sum_k \left(1 + \frac{\lambda}{\lambda_k}\right)^{-2}\phi_k(\widetilde{\mathbf{x}}_0)^2.$$

Then, by the Markov chain's central limit theorem and Slutsky's theorem, we obtain

$$n^{\frac{m}{2m+1}}\{f_{\mathbb{S},\lambda}(\widetilde{\mathbf{x}}_0) - f^\star(\widetilde{\mathbf{x}}_0)\} \xrightarrow{d} \mathcal{N}\left(0, \tilde{\sigma}_{\widetilde{\mathbf{x}}_0}^2\right).$$

$\square$

## C.3   PROOF OF GENERALIZATION BOUND

Based on Lemma 3 of Li et al. (2017), we can obtain the following Lemma for u.e.M.c. samples.

**Lemma 4.** *For any bounded measurable functions $f$ and u.e.M.c. samples $\widetilde{z}_1, \cdots, \widetilde{z}_n$, we assume that there exists a constant $C$ satisfying $0 \leq f(z) \leq C, \forall \widetilde{z}_i \in \widetilde{z}$. Thus for any $\varepsilon > 0$, we have*

$$\Pr\left\{\left|\frac{\frac{1}{n}\sum_{i=1}^n f(z_i) - \mathbb{E}(f)}{\sqrt{(\mathbb{E}(f) + \varepsilon)}}\right| \geq \sqrt{\varepsilon}\right\} \leq 2\exp\left\{\frac{-n\varepsilon}{56C\|\Gamma\|^2}\right\}, \tag{6}$$

$$\Pr\left\{\frac{\frac{1}{n}\sum_{i=1}^n f(z_i) - \mathbb{E}(f)}{\sqrt{(\mathbb{E}(f) + \varepsilon)}} \geq \sqrt{\varepsilon}\right\} \leq \exp\left\{\frac{-n\varepsilon}{56C\|\Gamma\|^2}\right\}, \tag{7}$$

*where $\|\Gamma\| = \sqrt{2}/\left(1 - \beta_0^{1/2n_1}\right)$, and $\mathbb{E}(f)$ is the expectation of function $f$.*

*Proof of Lemma 4.* Taking $\varepsilon = \sqrt{\varepsilon\{\mathbb{E}(f) + \varepsilon\}}$ in Lemma 3 of (Li et al., 2017) leads to

$$P\left\{\frac{\frac{1}{n}\sum_{i=1}^{n} f(z_i) - \mathbb{E}(f)}{\sqrt{(\mathbb{E}(f) + \varepsilon)}} \geq \sqrt{\varepsilon}\right\} \leq \exp\left\{\frac{-n(\varepsilon^2 + \varepsilon E(f))}{56C \|\Gamma\|^2 \mathbb{E}(f)}\right\}$$

$$= \exp\left\{\frac{-n\varepsilon}{56C \|\Gamma\|^2}\left(\frac{\varepsilon}{\mathbb{E}(f)} + 1\right)\right\}$$

$$\leq \exp\left\{\frac{-n\varepsilon}{56C \|\Gamma\|^2}\right\}.$$

$\square$

*Proof of Proposition 2.* According to the definition of excess risk, we have

$$\mathcal{R}_{\mathcal{F}}(f_{\mathbb{S},\lambda}) - \mathcal{R}_{\mathcal{F}}(f_0) \leq \mathcal{R}_{\mathcal{F}}(f_{\mathbb{S},\lambda}) - \mathcal{R}_{\mathcal{F}}(f_0) + \lambda\|f_{\mathcal{D},\lambda}\|_K^2$$

$$= \{\mathcal{R}_{\mathcal{F}}(f_{\mathbb{S},\lambda}) - \mathcal{R}_{\mathcal{D}}(f_{\mathbb{S},\lambda})\} + \{\mathcal{R}_{\mathcal{D}}(f_\lambda) - \mathcal{R}_{\mathcal{F}}(f_\lambda)\}$$

$$+ \mathcal{R}_{\mathcal{D}}(f_{\mathbb{S},\lambda}) - \mathcal{R}_{\mathcal{D}}(f_{\mathcal{D},\lambda}) + \{\mathcal{R}_{\mathcal{D}}(f_{\mathcal{D},\lambda}) + \lambda\|f_{\mathcal{D},\lambda}\|_K^2\}$$

$$- \{\mathcal{R}_{\mathcal{D}}(f_\lambda) + \lambda\|f_\lambda\|_K^2\} + \mathcal{R}_{\mathcal{F}}(f_\lambda) - \mathcal{R}_{\mathcal{F}}(f_0) + \lambda\|f_\lambda\|_K^2,$$

$$\mathcal{S}(\mathcal{D}, \mathbb{S}, \lambda) = [\mathcal{R}_{\mathcal{F}}(f_{\mathbb{S},\lambda}) - \mathcal{R}_{\mathcal{D}}(f_{\mathbb{S},\lambda})] + [\mathcal{R}_{\mathcal{D}}(f_\lambda) - \mathcal{R}_{\mathcal{F}}(f_\lambda)],$$

$$\mathcal{A}(\mathcal{D}, \mathbb{S}) = \mathcal{R}_{\mathcal{D}}(f_{\mathbb{S},\lambda}) - \mathcal{R}_{\mathcal{D}}(f_{\mathcal{D},\lambda}),$$

$$\mathcal{H}(\mathcal{D}, \lambda) = \left[\mathcal{R}_{\mathcal{D}}(f_{\mathcal{D},\lambda}) + \lambda\|f_{\mathcal{D},\lambda}\|_K^2\right] - \left[\mathcal{R}_{\mathcal{D}}(f_\lambda) + \lambda\|f_\lambda\|_K^2\right],$$

$$\mathcal{D}(\lambda) = \mathcal{R}_{\mathcal{F}}(f_\lambda) - \mathcal{R}_{\mathcal{F}}(f_0) + \lambda\|f_\lambda\|_K^2.$$

The definition of $f_{\mathcal{D},\lambda}$ implies that $\mathcal{H}(\mathcal{D}, \lambda)$ is at most zero. Hence, we obtain

$$\mathcal{R}_{\mathcal{F}}(f_{\mathbb{S},\lambda}) - \mathcal{R}_{\mathcal{F}}(f_0) \leq \mathcal{S}(\mathcal{D}, \mathbb{S}, \lambda) + \mathcal{A}(\mathcal{D}, \mathbb{S}) + \mathcal{D}(\lambda),$$

where $\mathcal{S}(\mathcal{D}, \mathbb{S}, \lambda)$, $\mathcal{A}(\mathcal{D}, \mathbb{S})$ and $\mathcal{D}(\lambda)$ denote the sample error, contamination error and regularization error, respectively. $\square$

*Proof of Proposition 3.* We utilize the idea of ER minimizer and probability inequality to bound this term by means of a covering number. For $R > 0$, we define $\mathcal{F}_R$ as the set of functions $\mathcal{F}_R := \left\{(f(\mathbf{x}) - y)^2 - (f_0(\mathbf{x}) - y)^2 : f_0 \in B_R\right\}$. Each function $g \in \mathcal{F}_R$ has the form $g(z) = (f_{\mathbb{S},\lambda}(\mathbf{x}) - y)^2 - (f_0(\mathbf{x}) - y)^2$ with $f \in B_R$. Hence, we obtain $\mathbf{E}(g) = \mathcal{R}_{\mathcal{F}}(f_{\mathbb{S},\lambda}) - \mathcal{R}_{\mathcal{F}}(f_0) \geq 0$, $\frac{1}{n}\sum_{i=1}^{n} g(z_i) = \mathcal{R}_{\mathcal{D}}(f_{\mathbb{S},\lambda}) - \mathcal{R}_{\mathcal{D}}(f_0)$, and

$$g(z) = \{f_{\mathbb{S},\lambda}(\mathbf{x}) - f_0(\mathbf{x})\}\{(f_{\mathbb{S},\lambda}(\mathbf{x}) - y) + (f_0(\mathbf{x}) - y)\}.$$

Since $|f_{\mathbb{S},\lambda}(\mathbf{x})| \leq M$ and $|f_0(\mathbf{x})| \leq M$, it is easily shown that $|g(z)| \leq \left|(f_{\mathbb{S},\lambda}(\mathbf{x}) - y)^2\right| + \left|(f_0(\mathbf{x}) - y)^2\right| \leq 8M^2$, and $\|g(z)\|_\infty \leq 8M^2$. By Lemma 3 of Li et al. (2017), for any $\varepsilon > 0$, we have

$$\Pr\left\{\sup_{f \in \mathcal{B}_R} \frac{[\mathcal{R}_{\mathcal{F}}(f_{\mathbb{S},\lambda}) - \mathcal{R}_{\mathcal{F}}(f_0)] - [\mathcal{R}_{\mathcal{D}}(f_{\mathbb{S},\lambda}) - \mathcal{R}_{\mathcal{D}}(f_0)]}{\sqrt{\mathcal{R}_{\mathcal{F}}(f_{\mathbb{S},\lambda}) - \mathcal{R}_{\mathcal{F}}(f_0) + \varepsilon}} \geq 4\sqrt{\varepsilon}\right\}$$

$$= \Pr\left\{\sup_{g \in \mathcal{F}_R} \frac{\mathbb{E}(g) - \frac{1}{n}\sum_{i=1}^{n} g(z_i)}{\sqrt{\mathbb{E}(g) + \varepsilon}} \geq 4\sqrt{\varepsilon}\right\}$$

$$\leq \mathcal{N}(\mathcal{F}_R, \varepsilon)\exp\left\{\frac{-\varepsilon n}{448M^2 \|\Gamma\|^2}\right\}. \tag{8}$$

For any $f_1, f_2 \in \mathcal{B}_R$, we have

$$|g_1(z) - g_2(z)| = \left|(f_1(\mathbf{x}) - y)^2 - (f_2(\mathbf{x}) - y)^2\right| \leq 4M |f_1(\mathbf{x}) - f_2(\mathbf{x})|.$$

Thus, for any $\varepsilon > 0$, an $\frac{\varepsilon}{4MR}$-covering of $\mathcal{B}_1$ provides an $\varepsilon$-covering of $\mathcal{F}_R$, i.e.,

$$\mathcal{N}\left(\mathcal{F}_R, \varepsilon\right) \leq \mathcal{N}\left(\mathcal{B}_{\mathcal{R}}, \frac{\varepsilon}{4M}\right) \leq \mathcal{N}\left(\mathcal{B}_1, \frac{\varepsilon}{4MR}\right). \tag{9}$$

Generally, $\mathcal{H}_1$ has polynomial complexity exponent $s > 0$ if there is some constant $c_s$ such that

$$\log\left(\mathcal{N}\left(\mathcal{H}_1, \varepsilon\right)\right) \leq c_s \varepsilon^{-s}, \forall \varepsilon > 0. \tag{10}$$

Combining inequality (8) and inequality (10) leads to

$$\Pr\left\{\sup_{f \in \mathcal{B}_R} \frac{[\mathcal{R}_{\mathcal{F}}(f_{\mathbb{S},\lambda}) - \mathcal{R}_{\mathcal{F}}(f_0)] - [\mathcal{R}_{\mathcal{D}}(f_{\mathbb{S},\lambda}) - \mathcal{R}_{\mathcal{D}}(f_0)]}{\sqrt{\mathcal{R}_{\mathcal{F}}(f_{\mathbb{S},\lambda}) - \mathcal{R}_{\mathcal{F}}(f_0) + \frac{1}{16}\varepsilon}} \geq \sqrt{\varepsilon}\right\}$$

$$= \Pr\left\{\sup_{g \in \mathcal{F}_R} \frac{\mathbb{E}(g) - \frac{1}{n}\sum_{i=1}^{n} g(z_i)}{\sqrt{\mathbb{E}(g) + \frac{1}{16}\varepsilon}} \geq \sqrt{\varepsilon}\right\}$$

$$\leq \mathcal{N}\left(\mathcal{F}_R, \frac{1}{16}\varepsilon\right) \exp\left\{\frac{-\varepsilon n}{7168 M^2 \|\Gamma\|^2}\right\}$$

$$\leq \mathcal{N}\left(\mathcal{B}_1, \frac{\varepsilon}{64MR}\right) \exp\left\{\frac{-\varepsilon n}{7168 M^2 \|\Gamma\|^2}\right\}.$$

Taking

$$\delta = \mathcal{N}\left(\mathcal{B}_1, \frac{\varepsilon}{64MR}\right) \exp\left\{\frac{-\varepsilon n}{7168 M^2 \|\Gamma\|^2}\right\},$$

leads to

$$\ln \delta = \ln \mathcal{N}\left(\mathcal{B}_1, \frac{\varepsilon}{64MR}\right) - \frac{\varepsilon n}{7168 M^2 \|\Gamma\|^2} \leq c_s \left(\frac{64MR}{\varepsilon}\right)^s - \frac{\varepsilon n}{7168 M^2 \|\Gamma\|^2},$$

which yields

$$\frac{\varepsilon n}{7168 M^2 \|\Gamma\|^2} - c_s \left(\frac{64MR}{\varepsilon}\right)^s - \ln\left(\frac{1}{\delta}\right) \leq 0.$$

It follows that

$$\varepsilon^{s+1} - \frac{7168 M^2 \|\Gamma\|^2 \ln(\frac{1}{\delta})}{n} \cdot \varepsilon^s - \frac{7168 M^2 \|\Gamma\|^2 c_s (64MR)^s}{n} \leq 0.$$

By Lemma 7 of Cucker & Smale (2002), we have

$$\varepsilon^* \leq \max\left\{\frac{14336 M^2 \|\Gamma\|^2 \ln(\frac{1}{\delta})}{n}, \left(\frac{14336 M^2 \|\Gamma\|^2 c_s (64MR)^s}{n}\right)^{\frac{1}{1+s}}\right\}$$

$$\leq \frac{14336 M^2 \|\Gamma\|^2 \ln(\frac{1}{\delta})}{n} + \left(\frac{14336 M^2 \|\Gamma\|^2 c_s (64MR)^s}{n}\right)^{\frac{1}{1+s}}.$$

It follows that:

$$\Pr\left\{\frac{[\mathcal{R}_{\mathcal{F}}(f_{\mathbb{S},\lambda}) - \mathcal{R}_{\mathcal{F}}(f_0)] - [\mathcal{R}_{\mathcal{D}}(f_{\mathbb{S},\lambda}) - \mathcal{R}_{\mathcal{D}}(f_0)]}{\sqrt{\mathcal{R}_{\mathcal{F}}(f_{\mathbb{S},\lambda}) - \mathcal{R}_{\mathcal{F}}(f_0) + \frac{1}{16}\varepsilon^*}} \leq \sqrt{\varepsilon^*}\right\} \geq 1 - \delta,$$

$$\Pr\left\{\mathcal{S}_1(\mathcal{D}, \mathbb{S}, \lambda) \leq \sqrt{\varepsilon^*}\sqrt{\mathcal{R}_{\mathcal{F}}(f_{\mathbb{S},\lambda}) - \mathcal{R}_{\mathcal{F}}(f_0) + \varepsilon^*}\right\} \geq 1 - \delta.$$

By Young's Inequality, we obtain

$$\sqrt{\varepsilon^*}\sqrt{\mathcal{R}_{\mathcal{F}}(f_{\mathbb{S},\lambda}) - \mathcal{R}_{\mathcal{F}}(f_0) + \varepsilon^*} \leq \frac{1}{2}\{\mathcal{R}_{\mathcal{F}}(f_{\mathbb{S},\lambda}) - \mathcal{R}_{\mathcal{F}}(f_0)\} + \varepsilon^*.$$

Thus, for any $\delta > 0$, with confidence at least $1 - \delta$,

$$\mathcal{S}_1(\mathcal{D}, \mathbb{S}, \lambda) \leq \frac{1}{2} \left[ \mathcal{R}_{\mathcal{F}}(f_{\mathbb{S},\lambda}) - \mathcal{R}_{\mathcal{F}}(f_0) \right] + \frac{14336 M^2 \|\Gamma\|^2 \ln(\frac{1}{\delta})}{n}$$
$$+ \left( \frac{14336 M^2 \|\Gamma\|^2 c_s (64 M R)^s}{n} \right)^{\frac{1}{1+s}}.$$

$\square$

*Proof of Proposition 4.* By the definition of $f_\lambda$, we obtain $\|f_\lambda\| \leq \mathcal{D}(\lambda)/\lambda$. Also, according to the condition of $K$, we have $\|f_\lambda\|_\infty \leq \kappa \|f_\lambda\| \leq \frac{\kappa \mathcal{D}(\lambda)}{\lambda}$. Taking

$$V = (f_\lambda(\mathbf{x}) - y)^2 - (f_0(\mathbf{x}) - y)^2 = (f_\lambda(\mathbf{x}) - f_0(\mathbf{x}))\{(f_\lambda(\mathbf{x}) - y) + (f_0(\mathbf{x}) - y)\}$$

yields

$$\begin{aligned}
|V| &= |(f_\lambda(\mathbf{x}) - f_0(\mathbf{x}))\{(f_\lambda(\mathbf{x}) - y) + (f_0(\mathbf{x}) - y)\}| \\
&\leq \{|f_\lambda(\mathbf{x})| + |f_0(\mathbf{x})|\} \{|f_\lambda(\mathbf{x})| + |f_0(\mathbf{x})| + |2y|\} \\
&\leq \left( \frac{\kappa \mathcal{D}(\lambda)}{\lambda} + M \right) \left( \frac{\kappa \mathcal{D}(\lambda)}{\lambda} + 3M \right) \\
&\leq \left( \frac{\kappa \mathcal{D}(\lambda)}{\lambda} + 3M \right)^2.
\end{aligned}$$

By Lemma 4, we have

$$\Pr \left\{ \frac{\frac{1}{n} \sum_{i=1}^n V(z_i) - \mathbb{E}(V)}{\sqrt{(\mathbb{E}(V) + \varepsilon)}} \geq \sqrt{\varepsilon} \right\} \leq \exp \left\{ \frac{-n\varepsilon}{56 \left( \frac{\kappa \mathcal{D}(\lambda)}{\lambda} + 3M \right)^2 \|\Gamma\|^2} \right\}.$$

Thus, for any $\varepsilon > 0$, we obtain

$$\Pr \left\{ \frac{(\mathcal{R}_{\mathcal{D}}(f_\lambda) - \mathcal{R}_{\mathcal{D}}(f_0)) - (\mathcal{R}_{\mathcal{F}}(f_\lambda) - \mathcal{R}_{\mathcal{F}}(f_0))}{\sqrt{(\mathcal{R}_{\mathcal{F}}(f_\lambda) - \mathcal{R}_{\mathcal{F}}(f_0)) + \varepsilon}} \geq \sqrt{\varepsilon} \right\} \leq \exp \left\{ \frac{-n\varepsilon}{56 \left( \frac{\kappa \mathcal{D}(\lambda)}{\lambda} + 3M \right)^2 \|\Gamma\|^2} \right\}.$$

Taking

$$\delta = \exp \left\{ \frac{-n\varepsilon}{56 \left( \frac{\kappa \mathcal{D}(\lambda)}{\lambda} + 3M \right)^2 \|\Gamma\|^2} \right\}, \quad \varepsilon = \frac{56 \|\Gamma\|^2 \left( \frac{\kappa \mathcal{D}(\lambda)}{\lambda} + 3M \right)^2 \ln(\frac{1}{\delta})}{n}.$$

It follows from the inequality $2\sqrt{ab} \leq a + b \,\, \forall a, b \geq 0$ that for any $0 < \delta < 1$, with confidence at least $1 - \delta$, we have

$$\begin{aligned}
\mathcal{S}_2(\mathbb{S}, \lambda) &\leq \frac{1}{2} \left[ \mathcal{R}_{\mathcal{F}}(f_\lambda) - \mathcal{R}_{\mathcal{F}}(f_0) \right] + \frac{56 \|\Gamma\|^2 \left( \frac{\kappa \mathcal{D}(\lambda)}{\lambda} + 3M \right)^2 \ln(\frac{1}{\delta})}{n} \\
&\leq \frac{1}{2} \mathcal{D}(\lambda) + \frac{56 \|\Gamma\|^2 \left( \frac{\kappa \mathcal{D}(\lambda)}{\lambda} + 3M \right)^2 \ln(\frac{1}{\delta})}{n}.
\end{aligned} \tag{11}$$

$\square$

*Proof of Proposition 5.* We assume that the sample set $\mathcal{M} = \{\hat{z}_i = (\hat{\mathbf{x}}_i, \hat{y}_i) : i = 1, \ldots, n\}$ is generated from distribution $\mathcal{Q}$. Then, we obtain

$$\mathcal{A}(\mathcal{D}, \mathbb{S}) = \mathcal{R}_{\mathcal{D}}(f_{\mathbb{S},\lambda}) - \mathcal{R}_{\mathcal{D}}(f_{\mathcal{D},\lambda}) \tag{12}$$
$$= [\mathcal{R}_{\mathcal{D}}(f_{\mathbb{S},\lambda}) - \mathcal{R}_{\mathbb{S}}(f_{\mathbb{S},\lambda})] + [\mathcal{R}_{\mathbb{S}}(f_{\mathbb{S},\lambda}) - \mathcal{R}_{\mathbb{S}}(f_{\mathcal{D},\lambda})] + [\mathcal{R}_{\mathbb{S}}(f_{\mathcal{D},\lambda}) - \mathcal{R}_{\mathcal{D}}(f_{\mathcal{D},\lambda})]. \tag{13}$$

Obviously, the second term $\mathcal{R}_{\mathbb{S}}(f_{\mathbb{S},\lambda}) - \mathcal{R}_{\mathbb{S}}(f_{\mathcal{D},\lambda}) \leq 0$, which yields

$$\mathcal{A}(\mathcal{D}, \mathbb{S}) \leq \{\mathcal{R}_{\mathcal{D}}(f_{\mathbb{S},\lambda}) - \mathcal{R}_{\mathbb{S}}(f_{\mathbb{S},\lambda})\} + \{\mathcal{R}_{\mathbb{S}}(f_{\mathcal{D},\lambda}) - \mathcal{R}_{\mathcal{D}}(f_{\mathcal{D},\lambda})\}.$$

For the first item of (13), we have

$$
\begin{aligned}
\mathcal{R}_{\mathcal{D}}(f_{\mathbb{S},\lambda}) - \mathcal{R}_{\mathbb{S}}(f_{\mathbb{S},\lambda}) &= \{\mathcal{R}_{\mathcal{D}}(f_{\mathbb{S},\lambda}) - \mathcal{R}_{\mathbb{S}}(f_{\mathbb{S},\lambda})\} - \{\mathcal{R}_{\mathcal{F}}(f_{\mathbb{S},\lambda}) - \mathcal{R}_{\mathcal{P}'}(f_{\mathbb{S},\lambda})\} \\
&\quad + \{\mathcal{R}_{\mathcal{F}}(f_{\mathbb{S},\lambda}) - \mathcal{R}_{\mathcal{P}'}(f_{\mathbb{S},\lambda})\} \\
&= \theta' \left[ \{\mathcal{R}_{\mathcal{D}}(f_{\mathbb{S},\lambda}) - \mathcal{R}_{\mathcal{M}}(f_{\mathbb{S},\lambda})\} - \{\mathcal{R}_{\mathcal{F}}(f_{\mathbb{S},\lambda}) - \mathcal{R}_{\mathcal{Q}}(f_{\mathbb{S},\lambda})\} \right] \\
&\quad + \{\mathcal{R}_{\mathcal{F}}(f_{\mathbb{S},\lambda}) - \mathcal{R}_{\mathcal{P}'}(f_{\mathbb{S},\lambda})\} \\
&\leq \theta' \left| \mathcal{R}_{\mathcal{D}}(f_{\mathbb{S},\lambda}) - \mathcal{R}_{\mathcal{F}}(f_{\mathbb{S},\lambda}) \right| + \theta' \left| \mathcal{R}_{\mathcal{Q}}(f_{\mathbb{S},\lambda}) - \mathcal{R}_{\mathcal{M}}(f_{\mathbb{S},\lambda}) \right| \\
&\quad + \left| \mathcal{R}_{\mathcal{F}}(f_{\mathbb{S},\lambda}) - \mathcal{R}_{\mathcal{P}'}(f_{\mathbb{S},\lambda}) \right|.
\end{aligned}
$$

Set random variable $\zeta = (f_{\mathbb{S},\lambda}(\mathbf{x}) - y)^2$, leading to $|\zeta| \leq 4M^2$. By lemma 4, we have

$$\Pr\left\{ \frac{|\mathcal{R}_{\mathcal{D}}(f_{\mathbb{S},\lambda}) - \mathcal{R}_{\mathcal{F}}(f_{\mathbb{S},\lambda})|}{\sqrt{(\mathcal{R}_{\mathcal{F}}(f_{\mathbb{S},\lambda}) + \varepsilon)}} \geq \sqrt{\varepsilon} \right\} \leq 2\exp\left\{ \frac{-n\varepsilon}{224M^2 \|\Gamma\|^2} \right\}.$$

Taking

$$\delta = 2\exp\left\{ \frac{-n\varepsilon}{224M^2 \|\Gamma\|^2} \right\}$$

yields

$$\varepsilon = \frac{224 \|\Gamma\|^2 M^2 \ln(\frac{2}{\delta})}{n}.$$

Thus, for any $0 < \delta < 1$, with confidence at least $1 - \delta$,

$$|\mathcal{R}_{\mathcal{D}}(f_{\mathbb{S},\lambda}) - \mathcal{R}_{\mathcal{F}}(f_{\mathbb{S},\lambda})| \leq 2M^2 + \frac{224 \|\Gamma\|^2 M^2 \ln(\frac{2}{\delta})}{n}, \tag{14}$$

Similarly, we obtain

$$|\mathcal{R}_{\mathcal{Q}}(f_{\mathbb{S},\lambda}) - \mathcal{R}_{\mathcal{M}}(f_{\mathbb{S},\lambda})| \leq 2M^2 + \frac{224 \|\Gamma\|^2 M^2 \ln(\frac{2}{\delta})}{n}. \tag{15}$$

$$
\begin{aligned}
|\mathcal{R}_{\mathcal{F}}(f_{\mathbb{S},\lambda}) - \mathcal{R}_{\mathcal{P}'}(f_{\mathbb{S},\lambda})| &= \left| \int_{\mathcal{Z}} (f_{\mathbb{S},\lambda}(\mathbf{x}) - y)^2 d(\mathcal{F} - \mathcal{P}') \right| \\
&\leq 2 \sup_{(\mathbf{x},y) \in \mathcal{Z}} (f_{\mathbb{S},\lambda}(\mathbf{x}) - y)^2 \|\mathcal{F} - \mathcal{P}'\|_{\mathrm{TV}} \\
&\leq 8M^2 \|\mathcal{F} - \mathcal{P}'\|_{\mathrm{TV}} \\
&\leq 8M^2 \theta'. \tag{16}
\end{aligned}
$$

Thus, combining (14)-(16), for any $0 < \delta < 1$, with confidence at least $1 - \delta$,

$$\mathcal{R}_{\mathcal{D}}(f_{\mathbb{S},\lambda}) - \mathcal{R}_{\mathbb{S}}(f_{\mathbb{S},\lambda}) \leq 12M^2\theta' + \frac{448 \|\Gamma\|^2 M^2 \ln(\frac{2}{\delta})}{n}\theta'.$$

Similarly, we obtain

$$\mathcal{R}_{\mathbb{S}}(f_{\mathcal{D},\lambda}) - \mathcal{R}_{\mathcal{D}}(f_{\mathcal{D},\lambda}) \leq 12M^2\theta' + \frac{448 \|\Gamma\|^2 M^2 \ln(\frac{2}{\delta})}{n}\theta'.$$

Thus, we have

$$\mathcal{A}(\mathcal{D}, \mathbb{S}) \leq 24M^2\theta' + \frac{896 \|\Gamma\|^2 M^2 \ln(\frac{2}{\delta})}{n}\theta'$$

$\square$

*Proof of Theorem 8.* By Propositions 2–5, we obtain the following inequality that holds with probability at least $1 - \delta$:

$$
\mathcal{R}_{\mathcal{F}}(f_{\mathbb{S},\lambda}) - \mathcal{R}_{\mathcal{F}}(f_0) \leq 2\left\{ \frac{14336M^2 \|\Gamma\|^2 \ln(\frac{1}{\delta})}{n} + \left( \frac{14336M^2 \|\Gamma\|^2 c_s (64MR)^s}{n} \right)^{\frac{1}{1+s}} \right\}
$$

$$
+ 3\mathcal{D}(\lambda) + \frac{112 \|\Gamma\|^2 \left( \frac{\kappa \mathcal{D}(\lambda)}{\lambda} + 3M \right)^2 \ln(\frac{1}{\delta})}{n}
$$

$$
+ 48M^2 \theta' + \frac{1792 \|\Gamma\|^2 M^2 \ln(\frac{2}{\delta})}{n} \theta'
$$

$$
\leq 48M^2 \theta' + C_2 \log(\frac{2}{\delta}) \left[ \lambda^q + \frac{\lambda^{2q-2}}{n} + \left( \frac{1}{n} \right)^{\frac{1}{1+s}} + \frac{1}{n} \right].
$$

By the choice of $\lambda$, we can be easily shown that

$$
\frac{\lambda^{2q-2}}{n} \leq \lambda^q, \qquad \left( \frac{1}{n} \right)^{\frac{1}{1+s}} \leq \lambda^q, \qquad \frac{1}{n} \leq \lambda^q.
$$

Taking $\vartheta_1 = \min\left\{ \frac{1}{2-q}, \frac{1}{(1+s)q}, \frac{1}{q} \right\}$ yields the desired result. The result establishes the asymptotic property of the excess risk: $\mathcal{R}_{\mathcal{F}}(f_{\mathbb{S},\lambda}) - \mathcal{R}_{\mathcal{F}}(f_0) \to 48M^2\theta'$ as $n \to \infty$, demonstrating that the KRMS estimator achieves consistency up to the contamination level $\theta'$. Through careful algorithm design that minimizes $\theta'$, the residual term $48M^2\theta'$ becomes negligible when $\theta'$ is sufficiently small. Consequently, as $n \to \infty$ and $\theta' \to 0$, the excess error $\mathcal{R}_{\mathcal{F}}(f_{\mathbb{S},\lambda}) - \mathcal{R}_{\mathcal{F}}(f_0) \to 0$, we obtain the consistency of the estimator. Moreover, the result provides an explicit learning rate of $O\left(n^{-\vartheta_1 q}\right)$. Notably, as $s \to 0$ and $\theta' \to 0$, this convergence rate approaches $O\left(n^{-1}\right)$, recovering the optimal convergence rate of the regularized least square Li et al. (2017).

$\square$

## D EXPERIMENTS

### D.1 THE RESULTS OF EXPERIMENT 1 IN SIMULATION STUDIES

The remaining resultes of Experiment 1 in simulation studies are presented in Table 2–3.

### D.2 THE RESULTS OF EXPERIMENT 2 IN SIMULATION STUDIES

**Expreiment 2** (Nonlinear model). In this experiment, we generate dataset $\{(\mathbf{x}_i, y_i)\}_{i=1}^N$ from the following nonlinear regression: $y_i = 2\exp(-x_{i1}) + 3\sin(\pi x_{i2}) + 2x_{i3}^2 + 3x_{i4} + \epsilon_i$, where $\epsilon_i$'s are independently drawn from the standard normal distribution, and $x_{i1}, \ldots, x_{i4}$ are independently sampled from the uniform distribution $U(0,1)$. The contaminated observations are created with contaminated data mechanism given in Experiment 1 for $\theta \in \{0.1, 0.2, 0.3, 0.4\}$. To generate contaminated data, we consider three cases for specifying $\mathbf{W}_i$ and $O_i$: (M1) (Background Noise): $W_{ij} \sim U(-10, 10)$ and $O_i \sim \mathcal{N}(0, 5)$; (M2) (Negative contamination with centered design): $W_{ij} \sim \mathcal{N}(-5, 10)$ and $O_i \sim \mathcal{N}(0, 5)$; (M3) (Mixed design): $W_{ij} \sim 0.5\mathcal{N}(-10, 5) + 0.5\mathcal{N}(10, 5)$ and $O_i \sim \mathcal{N}(0, 10)$. For comparison, we evaluate a baseline method, KRMS-Linear, which applies the same residual-based subsampling as the proposed KRMS-RKHS but differs only in its use of the linear kernel $K(\tilde{\mathbf{x}}_j, \tilde{\mathbf{x}}_i) = \tilde{\mathbf{x}}_j^\top \tilde{\mathbf{x}}_i$ within the Euclidean space to compute $w(\tilde{z}, \alpha)$ in Equation (3). The corresponding results for (M1)–(M3) are given in Tables 4–6, respectively.

By Tables 4–6, we have the following findings. First, the proposed KRMS-KLSR method outperforms others for all scenarios in that it consistently has the smallest AMSE and SD values regardless of contaminated schemes, contamination proportions, and subsample sizes, and maintains near-perfect PSR values (almost 100%), demonstrating exceptional robustness in identifying uncontaminated observations. Second, exception for contaminated scheme M3 together with low $\theta$ (e.g., $\theta = 0.1$), MS-KLSR and UNIF-KLSR show marginally better AMSE, and KRMS-KLSR

Table 2: Performance comparison of KRMS and five competing subsampling methods for corrupted mechanism M2 in Experiment 1

| $\theta$ | Method | $n = 500$ | | | $n = 1000$ | | | $n = 1500$ | | |
|---|---|---|---|---|---|---|---|---|---|---|
| | | AMSE | SD | PSR | AMSE | SD | PSR | AMSE | SD | PSR |
| 0.1 | UNIF-KLSR | 1.257 | 0.091 | 89.95% | 1.193 | 0.061 | 89.96% | 1.167 | 0.051 | 90.10% |
| | MS-KLSR | 1.151 | 0.055 | 97.34% | 1.110 | 0.043 | 96.99% | 1.096 | 0.039 | 97.09% |
| | KRMS-KLSR | **1.142** | **0.045** | **99.22%** | **1.089** | **0.037** | **99.26%** | **1.066** | **0.036** | **99.26%** |
| | UNIF-LSR | 4.199 | 0.986 | 89.95% | 4.229 | 0.684 | 89.96% | 4.125 | 0.516 | 90.10% |
| | GMS-LSR | 1.519 | 0.240 | 96.74% | 1.466 | 0.173 | 96.70% | 1.438 | 0.119 | 96.88% |
| | LGS-LSR | 1.303 | 0.144 | 98.98% | 1.207 | 0.093 | 98.94% | 1.147 | 0.060 | 98.93% |
| 0.2 | UNIF-KLSR | 1.450 | 0.131 | 80.03% | 1.389 | 0.078 | 79.97% | 1.387 | 0.070 | 79.88% |
| | MS-KLSR | 1.250 | 0.078 | 93.02% | 1.221 | 0.055 | 92.89% | 1.229 | 0.051 | 92.95% |
| | KRMS-KLSR | **1.171** | **0.048** | **98.01%** | **1.106** | **0.046** | **98.26%** | **1.083** | **0.035** | **98.26%** |
| | UNIF-LSR | 8.677 | 1.101 | 80.03% | 8.840 | 0.941 | 79.97% | 9.006 | 0.680 | 79.88% |
| | GMS-LSR | 3.079 | 0.648 | 88.21% | 3.047 | 0.566 | 88.69% | 3.074 | 0.446 | 88.73% |
| | LGS-LSR | 1.348 | 0.188 | 97.60% | 1.308 | 0.161 | 97.48% | 1.240 | 0.100 | 97.44% |
| 0.3 | UNIF-KLSR | 1.778 | 0.193 | 70.15% | 1.744 | 0.133 | 69.79% | 1.700 | 0.119 | 70.09% |
| | MS-KLSR | 1.463 | 0.130 | 87.09% | 1.437 | 0.094 | 87.17% | 1.422 | 0.080 | 87.75% |
| | KRMS-KLSR | **1.209** | **0.074** | **96.70%** | **1.128** | **0.048** | **96.63%** | **1.115** | **0.036** | **96.80%** |
| | UNIF-LSR | 12.826 | 1.411 | 70.15% | 13.187 | 0.768 | 69.79% | 13.242 | 0.648 | 70.09% |
| | GMS-LSR | 6.696 | 1.218 | 77.12% | 7.034 | 0.861 | 77.36% | 6.918 | 0.704 | 78.16% |
| | LGS-LSR | 1.499 | 0.305 | 95.80% | 1.462 | 0.219 | 95.50% | 1.425 | 0.133 | 95.26% |
| 0.4 | UNIF-KLSR | 2.290 | 0.249 | 60.37% | 2.196 | 0.190 | 60.19% | 2.188 | 0.177 | 60.13% |
| | MS-KLSR | 1.806 | 0.179 | 79.15% | 1.753 | 0.140 | 80.19% | 1.763 | 0.131 | 80.85% |
| | KRMS-KLSR | **1.255** | **0.093** | **94.31%** | **1.185** | **0.056** | **94.42%** | **1.163** | **0.051** | **94.79%** |
| | UNIF-LSR | 16.373 | 1.309 | 60.37% | 16.604 | 0.890 | 60.19% | 16.66 | 0.740 | 60.13% |
| | GMS-LSR | 11.193 | 1.208 | 64.90% | 11.426 | 0.808 | 66.01% | 11.561 | 0.714 | 66.83% |
| | LGS-LSR | 1.961 | 0.598 | 92.33% | 2.041 | 0.385 | 91.39% | 2.027 | 0.379 | 90.81% |

Table 3: Performance comparison of KRMS and five competing subsampling methods for corrupted mechanism M3 in Experiment 1

| $\theta$ | Method | $n = 500$ | | | $n = 1000$ | | | $n = 1500$ | | |
|---|---|---|---|---|---|---|---|---|---|---|
| | | AMSE | SD | PSR | AMSE | SD | PSR | AMSE | SD | PSR |
| 0.1 | UNIF-KLSR | 3.596 | 0.337 | 90.10% | 3.693 | 0.162 | 90.04% | 3.712 | 0.118 | 90.12% |
| | MS-KLSR | 3.213 | 0.459 | 96.66% | 3.518 | 0.233 | 96.73% | 3.573 | 0.181 | 97.25% |
| | KRMS-KLSR | 1.117 | **0.043** | **99.78%** | 1.086 | **0.036** | **99.76%** | 1.076 | **0.034** | **99.71%** |
| | UNIF-LSR | 28.159 | 1.042 | 90.10% | 28.461 | 0.461 | 90.04% | 28.498 | 0.391 | 90.12% |
| | GMS-LSR | 11.08 | 5.483 | 95.04% | 14.301 | 5.22 | 95.07% | 17.638 | 4.482 | 95.34% |
| | LGS-LSR | **1.031** | 0.093 | 99.48% | **1.029** | 0.057 | 99.44% | **1.034** | 0.066 | 99.47% |
| 0.2 | UNIF-KLSR | 4.141 | 0.264 | 79.67% | 4.234 | 0.218 | 80.07% | 4.335 | 0.212 | 80.02% |
| | MS-KLSR | 3.993 | 0.227 | 91.74% | 4.096 | 0.193 | 92.27% | 4.163 | 0.183 | 93.12% |
| | KRMS-KLSR | 1.129 | **0.041** | **99.39%** | 1.079 | **0.036** | **99.42%** | **1.063** | **0.035** | **99.38%** |
| | UNIF-LSR | 28.515 | 0.417 | 79.67% | 28.485 | 0.382 | 80.07% | 28.48 | 0.463 | 80.02% |
| | GMS-LSR | 20.024 | 4.049 | 89.55% | 23.360 | 2.811 | 89.79% | 24.081 | 2.512 | 90.14% |
| | LGS-LSR | **1.080** | 0.124 | 98.80% | **1.062** | 0.107 | 98.77% | 1.069 | 0.160 | 98.76% |
| 0.3 | UNIF-KLSR | 5.124 | 0.495 | 69.94% | 5.370 | 0.404 | 69.92% | 5.549 | 0.316 | 69.88% |
| | MS-KLSR | 4.876 | 0.383 | 84.13% | 5.073 | 0.318 | 85.28% | 5.189 | 0.264 | 86.31% |
| | KRMS-KLSR | 1.125 | **0.043** | **99.04%** | 1.078 | **0.037** | **99.00%** | 1.072 | **0.036** | **99.00%** |
| | UNIF-LSR | 28.405 | 0.486 | 69.94% | 28.407 | 0.486 | 69.92% | 28.408 | 0.486 | 69.88% |
| | GMS-LSR | 24.036 | 2.644 | 83.26% | 25.839 | 1.477 | 83.59% | 26.503 | 1.053 | 84.12% |
| | LGS-LSR | **1.097** | 0.111 | 97.98% | 1.177 | 0.318 | 97.93% | 1.148 | 0.227 | 97.82% |
| 0.4 | UNIF-KLSR | 6.665 | 0.671 | 60.35% | 7.063 | 0.553 | 60.36% | 7.250 | 0.413 | 60.37% |
| | MS-KLSR | 6.383 | 0.580 | 73.44% | 6.669 | 0.444 | 74.53% | 6.745 | 0.344 | 76.32% |
| | KRMS-KLSR | **1.133** | **0.041** | **98.37%** | **1.089** | **0.037** | **98.42%** | **1.079** | **0.036** | **98.42%** |
| | UNIF-LSR | 28.618 | 0.411 | 60.35% | 28.618 | 0.411 | 60.36% | 28.618 | 0.411 | 60.37% |
| | GMS-LSR | 26.822 | 1.159 | 76.08% | 27.265 | 0.872 | 76.71% | 27.658 | 0.636 | 77.65% |
| | LGS-LSR | 1.273 | 0.343 | 96.59% | 1.272 | 0.278 | 96.49% | 1.291 | 0.273 | 96.32% |

method still maintains significantly higher PSR values and competitive SD values. This deviation

likely reflects M3's milder perturbation effect at low proportion, where random sampling may occasionally succeed. Third, as $\theta$ increases, KRMS-KLSR's AMSE values remain stable, while others show severe degradation. Fourth, MS-KLSR method outperforms UNIF-KLSR but remains inferior to KRMS-KLSR method. Fifth, a critical limitation of KRMS-Linear is its severe performance degradation with nonlinear function $f_0$. Empirical evidence from the M1 contamination scheme at $\theta = 0.4$ shows a PSR of merely 16.69%. This validates the model misspecification bias hypothesis: the linear estimator's failure to represent the nonlinear trend causes universally large residuals. Consequently, the residual-based score fails to reliably separate outliers from the model's own structural errors, invalidating its discriminative power.

Table 4: Performance comparison of KRMS-KLSR and six competing subsampling methods for corrupted mechanism M1 in Experiment 2

| $\theta$ | Method | $n = 500$ | | | $n = 1000$ | | | $n = 1500$ | | |
|---|---|---|---|---|---|---|---|---|---|---|
| | | AMSE | SD | PSR | AMSE | SD | PSR | AMSE | SD | PSR |
| 0.1 | UNIF-KLSR | 1.617 | 0.093 | 89.94% | 1.720 | 0.117 | 90.05% | 1.821 | 0.173 | 90.10% |
| | MS-KLSR | 1.526 | 0.126 | 96.81% | 1.498 | 0.094 | 96.77% | 1.768 | 0.087 | 97.70% |
| | KRMS-KLSR | **1.137** | **0.041** | **100.00%** | **1.098** | **0.036** | **100.00%** | **1.087** | **0.035** | **100.00%** |
| | KRMS-Linear | 25.631 | 0.964 | 75.58% | 25.245 | 0.900 | 78.07% | 25.055 | 0.711 | 79.82% |
| | UNIF-LSR | 20.074 | 2.044 | 89.94% | 20.093 | 1.330 | 90.05% | 20.241 | 1.115 | 90.10% |
| | GMS-LSR | 4.981 | 1.692 | 97.62% | 5.001 | 0.976 | 97.62% | 4.860 | 0.856 | 97.68% |
| | LGS-LSR | 2.917 | 0.156 | 99.87% | 2.947 | 0.134 | 99.86% | 2.945 | 0.126 | 99.85% |
| 0.2 | UNIF-KLSR | 1.880 | 0.068 | 79.90% | 1.908 | 0.063 | 79.89% | 1.955 | 0.076 | 80.01% |
| | MS-KLSR | 1.839 | 0.067 | 88.69% | 1.906 | 0.064 | 88.41% | 1.973 | 0.123 | 88.70% |
| | KRMS-KLSR | **1.153** | **0.044** | **100.00%** | **1.099** | **0.038** | **100.00%** | **1.091** | **0.035** | **100.00%** |
| | KRMS-Linear | 29.833 | 0.548 | 41.96% | 29.746 | 0.547 | 47.15% | 29.708 | 0.472 | 51.71% |
| | UNIF-LSR | 25.317 | 1.045 | 79.90% | 25.403 | 0.802 | 79.89% | 25.472 | 0.658 | 80.01% |
| | GMS-LSR | 12.806 | 2.468 | 91.74% | 13.372 | 2.049 | 91.75% | 13.526 | 1.489 | 91.95% |
| | LGS-LSR | 3.046 | 0.238 | 99.66% | 3.051 | 0.213 | 99.64% | 3.062 | 0.203 | 99.58% |
| 0.3 | UNIF-KLSR | 2.119 | 0.142 | 70.31% | 2.120 | 0.090 | 69.98% | 2.134 | 0.088 | 70.04% |
| | MS-KLSR | 2.071 | 0.116 | 79.82% | 2.136 | 0.097 | 79.99% | 2.161 | 0.090 | 80.24% |
| | KRMS-KLSR | **1.145** | **0.050** | **100.00%** | **1.104** | **0.041** | **100.00%** | **1.101** | **0.038** | **100.00%** |
| | KRMS-Linear | 30.734 | 0.499 | 25.19% | 30.712 | 0.450 | 30.98% | 30.732 | 0.475 | 36.01% |
| | UNIF-LSR | 27.574 | 0.686 | 70.31% | 27.769 | 0.558 | 69.98% | 27.76 | 0.545 | 70.04% |
| | GMS-LSR | 20.876 | 1.810 | 83.44% | 21.228 | 1.275 | 83.98% | 21.37 | 0.922 | 84.27% |
| | LGS-LSR | 3.191 | 0.418 | 99.41% | 3.171 | 0.330 | 99.35% | 3.177 | 0.295 | 99.26% |
| 0.4 | UNIF-KLSR | 2.500 | 0.159 | 60.04% | 2.418 | 0.121 | 59.97% | 2.407 | 0.092 | 60.07% |
| | MS-KLSR | 2.443 | 0.129 | 69.94% | 2.427 | 0.111 | 70.94% | 2.444 | 0.098 | 71.87% |
| | KRMS-KLSR | **1.149** | **0.045** | **100.00%** | **1.104** | **0.042** | **100.00%** | **1.106** | **0.042** | **100.00%** |
| | KRMS-Linear | 31.086 | 0.518 | 16.69% | 31.087 | 0.483 | 22.57% | 31.084 | 0.475 | 27.13% |
| | UNIF-LSR | 29.034 | 0.587 | 60.04% | 29.128 | 0.471 | 59.97% | 29.056 | 0.463 | 60.07% |
| | GMS-LSR | 25.317 | 0.984 | 74.34% | 25.612 | 0.739 | 74.69% | 25.524 | 0.654 | 75.42% |
| | LGS-LSR | 3.469 | 0.577 | 99.02% | 3.665 | 0.582 | 98.63% | 3.536 | 0.465 | 98.60% |

To further demonstrate the effectiveness of KRMS-KLSR, we present a visual analysis of its performance. Figure 1 depicts the density distribution of the sampling metric $\log(w)$ for clean versus contaminated samples. The results show a more distinct separation between inliers and outliers under nonlinear settings, demonstrating that our kernel-induced residual score more effectively distinguishes anomalies. In contrast, the sampling metric distribution of KRMS-Linear exhibits substantial overlap, which hinders its ability to filter out contaminated data points during subsampling. As illustrated in Figure 2, the linear constraints of KRMS-Linear lead to a fitted curve that is severely distorted by outliers. However, our method accurately captures the underlying nonlinear structure, resists the influence of outliers, and recovers a smooth curve that aligns well with the true curve.

In conclusion, our method offers two key advantages. First, it excels at capturing nonlinear features, enabling the separation of data patterns including outliers that are linearly inseparable in the original input space by mapping them into a higher-dimensional RKHS. Second, it achieves enhanced robustness through a more precise assessment of local data structure in the feature space, quantified by the kernel term $\sqrt{\sum K\left(\tilde{x}_i, \tilde{x}_j\right)^2}$. This facilitates more reliable outlier identification and suppression, leading to consistently stronger statistical performance on complex datasets compared to linear baselines.

Table 5: Performance comparison of KRMS-KLSR and six competing subsampling methods for corrupted mechanism M2 in Experiment 2

| $\theta$ | Method | $n = 500$ | | | $n = 1000$ | | | $n = 1500$ | | |
|---|---|---|---|---|---|---|---|---|---|---|
| | | AMSE | SD | PSR | AMSE | SD | PSR | AMSE | SD | PSR |
| 0.1 | UNIF-KLSR | 1.792 | 0.075 | 89.78% | 1.809 | 0.122 | 89.81% | 1.796 | 0.212 | 90.09% |
| | MS-KLSR | 1.767 | 0.071 | 94.88% | 1.818 | 0.069 | 94.84% | 1.863 | 0.123 | 94.96% |
| | KRMS-KLSR | **1.140** | **0.044** | **100.00%** | **1.092** | **0.037** | **100.00%** | **1.088** | **0.037** | **100.00%** |
| | KRMS-Linear | 29.846 | 0.547 | 71.92% | 29.721 | 0.496 | 75.58% | 29.664 | 0.415 | 77.34% |
| | UNIF-LSR | 27.001 | 0.899 | 89.78% | 26.966 | 0.648 | 89.81% | 26.966 | 0.587 | 90.09% |
| | GMS-LSR | 14.325 | 3.736 | 97.05% | 14.540 | 2.587 | 97.29% | 14.854 | 2.066 | 97.32% |
| | LGS-LSR | 2.875 | 0.138 | 99.98% | 2.888 | 0.122 | 99.98% | 2.922 | 0.130 | 99.97% |
| 0.2 | UNIF-KLSR | 1.913 | 0.069 | 80.23% | 1.919 | 0.073 | 80.02% | 1.921 | 0.076 | 80.02% |
| | MS-KLSR | 1.933 | 0.067 | 87.20% | 1.961 | 0.076 | 87.12% | 1.971 | 0.093 | 87.50% |
| | KRMS-KLSR | **1.141** | **0.042** | **100.00%** | **1.104** | **0.039** | **100.00%** | **1.095** | **0.042** | **100.00%** |
| | KRMS-Linear | 31.015 | 0.485 | 41.27% | 31.004 | 0.453 | 46.19% | 30.999 | 0.458 | 51.25% |
| | UNIF-LSR | 29.314 | 0.532 | 80.23% | 29.444 | 0.445 | 80.02% | 29.445 | 0.482 | 80.02% |
| | GMS-LSR | 25.457 | 1.205 | 90.18% | 25.764 | 0.847 | 90.28% | 25.869 | 0.759 | 90.33% |
| | LGS-LSR | 2.964 | 0.209 | 99.96% | 3.072 | 0.296 | 99.92% | 2.997 | 0.254 | 99.93% |
| 0.3 | UNIF-KLSR | 2.005 | 0.096 | 70.23% | 1.979 | 0.069 | 69.85% | 1.997 | 0.069 | 70.01% |
| | MS-KLSR | 2.046 | 0.092 | 78.97% | 2.047 | 0.078 | 79.59% | 2.073 | 0.079 | 80.20% |
| | KRMS-KLSR | **1.145** | **0.043** | **100.00%** | **1.105** | **0.045** | **100.00%** | **1.109** | **0.041** | **100.00%** |
| | KRMS-Linear | 31.324 | 0.428 | 24.64% | 31.314 | 0.427 | 31.36% | 31.338 | 0.424 | 36.16% |
| | UNIF-LSR | 30.36 | 0.493 | 70.23% | 30.324 | 0.494 | 69.85% | 30.325 | 0.442 | 70.0%1 |
| | GMS-LSR | 28.614 | 0.587 | 81.81% | 28.552 | 0.547 | 82.15% | 28.616 | 0.469 | 82.52% |
| | LGS-LSR | 3.420 | 0.827 | 99.88% | 3.515 | 0.709 | 99.83% | 3.507 | 0.698 | 99.81% |
| 0.4 | UNIF-KLSR | 2.086 | 0.117 | 60.29% | 2.052 | 0.075 | 60.19% | 2.041 | 0.076 | 60.00% |
| | MS-KLSR | 2.156 | 0.108 | 70.12% | 2.151 | 0.085 | 71.15% | 2.163 | 0.087 | 72.16% |
| | KRMS-KLSR | **1.148** | **0.048** | **100.00%** | **1.110** | **0.034** | **100.00%** | **1.103** | **0.037** | **100.00%** |
| | KRMS-Linear | 31.374 | 0.418 | 16.08% | 31.387 | 0.418 | 22.33% | 31.395 | 0.402 | 27.26% |
| | UNIF-LSR | 30.687 | 0.470 | 60.29% | 30.799 | 0.386 | 60.19% | 30.788 | 0.443 | 60.00% |
| | GMS-LSR | 29.648 | 0.483 | 72.89% | 29.767 | 0.401 | 73.42% | 29.771 | 0.465 | 74.41% |
| | LGS-LSR | 6.807 | 4.843 | 98.86% | 6.526 | 2.817 | 99.23% | 7.526 | 2.775 | 98.96% |

Table 6: Performance comparison of KRMS-KLSR and six competing subsampling methods for corrupted mechanism M3 in Experiment 2

| $\theta$ | Method | $n = 500$ | | | $n = 1000$ | | | $n = 1500$ | | |
|---|---|---|---|---|---|---|---|---|---|---|
| | | AMSE | SD | PSR | AMSE | SD | PSR | AMSE | SD | PSR |
| 0.1 | UNIF-KLSR | 1.087 | 0.046 | 89.93% | 1.064 | 0.036 | 89.90% | 1.063 | 0.035 | 89.92% |
| | MS-KLSR | **1.085** | **0.040** | 97.14% | **1.058** | **0.032** | 97.00% | **1.051** | 0.032 | 97.04% |
| | KRMS-KLSR | 1.142 | 0.042 | **99.96%** | 1.091 | 0.038 | **99.96%** | 1.072 | **0.029** | **99.94%** |
| | KRMS-Linear | 3.773 | 0.404 | 93.56% | 3.781 | 0.299 | 93.90% | 3.875 | 0.315 | 93.83% |
| | UNIF-LSR | 4.489 | 0.515 | 89.93% | 4.515 | 0.423 | 89.90% | 4.501 | 0.362 | 89.92% |
| | GMS-LSR | 3.279 | 0.218 | 96.81% | 3.246 | 0.169 | 96.94% | 3.239 | 0.134 | 96.94% |
| | LGS-LSR | 3.105 | 0.192 | 98.27% | 3.000 | 0.124 | 98.26% | 2.957 | 0.090 | 98.18% |
| 0.2 | UNIF-KLSR | 1.276 | 0.107 | 80.15% | 1.269 | 0.077 | 79.86% | 1.277 | 0.066 | 80.20% |
| | MS-KLSR | 1.166 | 0.059 | 92.26% | 1.149 | 0.050 | 92.23% | 1.158 | 0.049 | 92.30% |
| | KRMS-KLSR | **1.146** | **0.038** | **99.86%** | **1.096** | **0.036** | **99.89%** | **1.080** | **0.035** | **99.88%** |
| | KRMS-Linear | 6.982 | 0.806 | 80.35% | 7.080 | 0.663 | 80.86% | 7.230 | 0.588 | 81.24% |
| | UNIF-LSR | 7.103 | 0.865 | 80.15% | 7.273 | 0.675 | 79.86% | 7.179 | 0.541 | 80.20% |
| | GMS-LSR | 3.890 | 0.363 | 90.92% | 4.029 | 0.270 | 90.74% | 3.938 | 0.224 | 91.15% |
| | LGS-LSR | 3.087 | 0.187 | 96.11% | 3.098 | 0.173 | 95.98% | 3.028 | 0.130 | 95.91% |
| 0.3 | UNIF-KLSR | 1.605 | 0.121 | 69.87% | 1.534 | 0.092 | 70.04% | 1.526 | 0.088 | 70.14% |
| | MS-KLSR | 1.351 | 0.085 | 85.77% | 1.333 | 0.062 | 86.16% | 1.369 | 0.068 | 86.23% |
| | KRMS-KLSR | **1.145** | **0.043** | **99.82%** | **1.095** | **0.041** | **99.81%** | **1.078** | **0.039** | **99.79%** |
| | KRMS-Linear | 12.433 | 1.402 | 61.88% | 12.228 | 0.962 | 64.13% | 12.099 | 0.911 | 65.66% |
| | UNIF-LSR | 10.294 | 1.182 | 69.87% | 10.420 | 0.799 | 70.04% | 10.349 | 0.676 | 70.14% |
| | GMS-LSR | 5.219 | 0.607 | 81.37% | 5.241 | 0.449 | 81.78% | 5.162 | 0.358 | 82.38% |
| | LGS-LSR | 3.242 | 0.243 | 93.29% | 3.222 | 0.188 | 93.11% | 3.212 | 0.162 | 92.71% |
| 0.4 | UNIF-KLSR | 1.807 | 0.139 | 60.53% | 1.757 | 0.088 | 59.98% | 1.724 | 0.074 | 60.04% |
| | MS-KLSR | 1.549 | 0.118 | 78.56% | 1.579 | 0.075 | 79.05% | 1.581 | 0.059 | 79.70% |
| | KRMS-KLSR | **1.147** | **0.050** | **99.71%** | **1.096** | **0.037** | **99.72%** | **1.077** | **0.034** | **99.69%** |
| | KRMS-Linear | 17.919 | 1.361 | 44.68% | 17.667 | 1.024 | 47.61% | 17.570 | 0.927 | 49.64% |
| | UNIF-LSR | 13.681 | 1.439 | 60.53% | 13.844 | 0.879 | 59.98% | 13.81 | 0.705 | 60.04% |
| | GMS-LSR | 7.529 | 0.839 | 69.89% | 7.523 | 0.629 | 70.76% | 7.553 | 0.527 | 71.67% |
| | LGS-LSR | 3.532 | 0.422 | 89.19% | 3.642 | 0.370 | 88.12% | 3.595 | 0.266 | 87.69% |

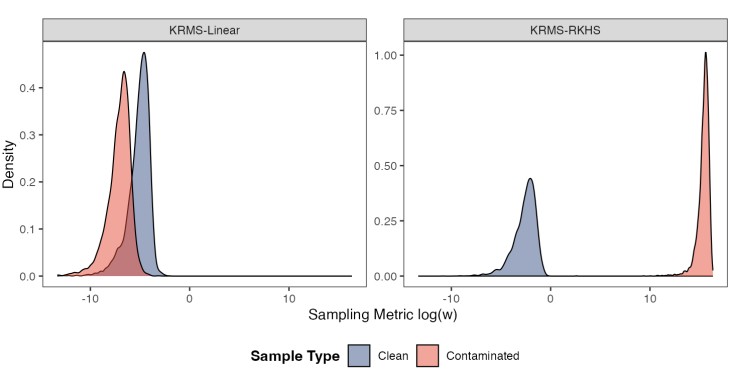

Figure 1: Sampling Metric Distributions of KRMS-RKHS and KRMS-Linear methods

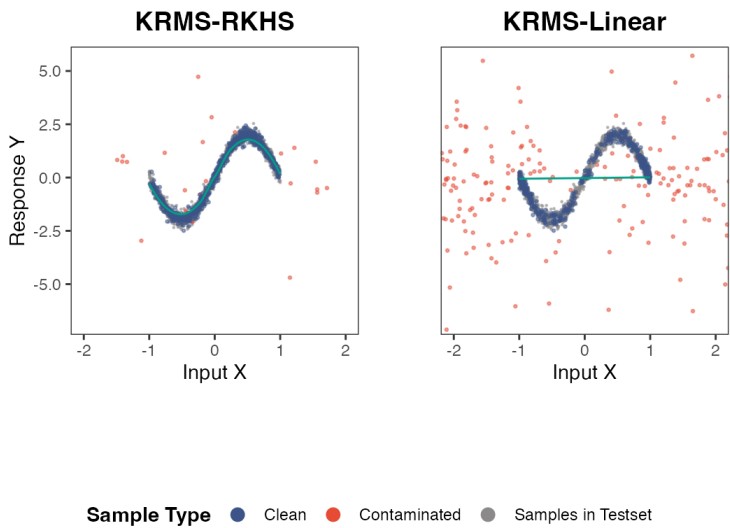

Figure 2: Scatter plot and fitted curve of the subsamples of KRMS-RKHS and KRMS-Linear methods

To validate the scalability regarding data size and dimensionality, we extend the experimental setting to $N = 20,000$ and $p = 50$ under case M1, with $n \in \{1000, 2000, 3000\}$. In addition to the five methods compared earlier, we include a robust nonparametric regression method, Support vector regression (SVR) (Karatzoglou et al., 2004). The results are shown in Table 7. As shown in Table 7, even with a large sample size and higher dimensionality, the KRMS method maintains its effectiveness and robustness, achieving a PSR of $100\%$ across all the considered contamination levels ($\theta \in [0.1, 0.4]$) and subsample sizes. In terms of estimation accuracy, the AMSE of KRMS remains stable under contamination ($\theta > 0$) and is comparable to the uncontaminated baseline ($\theta = 0$). Moreover, KRMS consistently yields lower AMSE values than all benchmark methods, including SVR.

We also assess the computational complexity of the proposed method with respect to sample size $N$ and dimensionality $p$. To ensure a fair comparison, the analysis is limited to kernel-based competitors. As shown in Figures 3 and 4, KRMS-KLSR incurs the highest computational cost among the evaluated methods, which is mainly due to the iterative sampling step required for robust estimation. Empirically, the runtime of KRMS-KLSR scales approximately linearly with $N$ when $p$ is

fixed. In the most challenging scenario ($N = 20,000, p = 100$), the average runtime is about 600 seconds. While computationally more intensive, this trade-off is justified by the significant gains in robustness and estimation accuracy demonstrated in Table 7.

Table 7: Performance comparison of KRMS-KLSR and six competing methods for corrupted mechanism M1 in Experiment 2

| $\theta$ | Method | $n = 1000$ | | | $n = 2000$ | | | $n = 3000$ | | |
|---|---|---|---|---|---|---|---|---|---|---|
| | | AMSE | SD | PSR | AMSE | SD | PSR | AMSE | SD | PSR |
| | UNIF-KLSR | **1.958** | 0.054 | 100.00% | **1.783** | **0.054** | 100.00% | 1.745 | **0.047** | 100.00% |
| | MS-KLSR | 1.996 | **0.049** | 100.00% | 1.848 | **0.054** | 100.00% | 1.830 | 0.050 | 100.00% |
| | KRMS-KLSR | 1.987 | 0.053 | 100.00% | 1.840 | 0.057 | 100.00% | 1.808 | 0.048 | 100.00% |
| 0 | SVR | 1.975 | 0.059 | 100.00% | 1.786 | 0.055 | 100.00% | **1.693** | 0.048 | 100.00% |
| | UNIF-LSR | 2.073 | 0.058 | 100.00% | 2.001 | 0.063 | 100.00% | 1.999 | 0.051 | 100.00% |
| | GMS-LSR | 2.051 | 0.053 | 100.00% | 1.995 | 0.060 | 100.00% | 1.994 | 0.048 | 100.00% |
| | LGS-LSR | 2.052 | **0.049** | 100.00% | 1.996 | 0.060 | 100.00% | 1.994 | 0.050 | 100.00% |
| | UNIF-KLSR | 2.026 | **0.060** | 89.85% | 2.820 | 0.115 | 90.00% | 2.838 | 0.104 | 89.96% |
| | MS-KLSR | 2.220 | 0.072 | 98.63% | 2.687 | 0.090 | 98.98% | 2.702 | 0.111 | 98.96% |
| | KRMS-KLSR | **1.981** | 0.061 | **100.00%** | **1.852** | 0.069 | **100.00%** | **1.804** | **0.055** | **100.00%** |
| 0.1 | SVR | 2.013 | 0.061 | 89.85% | 1.944 | **0.060** | 90.00% | 1.905 | 0.057 | 89.96% |
| | UNIF-LSR | 3.806 | 0.316 | 89.85% | 4.119 | 0.272 | 90.00% | 4.295 | 0.242 | 89.96% |
| | GMS-LSR | 2.417 | 0.135 | 99.24% | 2.578 | 0.136 | 99.18% | 2.723 | 0.153 | 99.10% |
| | LGS-LSR | 2.047 | 0.080 | 99.86% | 2.041 | 0.083 | 99.80% | 2.046 | 0.072 | 99.81% |
| | UNIF-KLSR | 2.116 | 0.073 | 80.12% | 3.157 | 0.105 | 80.23% | 3.149 | 0.093 | 80.03% |
| | MS-KLSR | 2.148 | 0.091 | 95.41% | 3.135 | 0.105 | 96.46% | 3.125 | 0.095 | 96.58% |
| | KRMS-KLSR | **1.994** | **0.067** | **100.00%** | **1.857** | **0.053** | **100.00%** | **1.808** | **0.051** | **100.00%** |
| 0.2 | SVR | 2.044 | 0.072 | 80.12% | 1.966 | 0.056 | 80.23% | 1.937 | 0.059 | 80.03% |
| | UNIF-LSR | 6.253 | 0.635 | 80.12% | 6.933 | 0.521 | 80.23% | 7.173 | 0.469 | 80.03% |
| | GMS-LSR | 2.960 | 0.237 | 97.17% | 3.358 | 0.172 | 96.80% | 3.436 | 0.138 | 96.76% |
| | LGS-LSR | 2.080 | 0.080 | 99.63% | 2.142 | 0.103 | 99.56% | 2.172 | 0.079 | 99.48% |
| | UNIF-KLSR | 2.148 | 0.085 | 70.09% | 3.133 | 0.119 | 70.10% | 3.308 | 0.109 | 70.08% |
| | MS-KLSR | 2.303 | 0.106 | 86.54% | 3.091 | 0.130 | 91.76% | 3.245 | 0.110 | 92.03% |
| | KRMS-KLSR | **1.975** | **0.074** | **100.00%** | **1.851** | **0.060** | **100.00%** | **1.807** | **0.061** | **100.00%** |
| 0.3 | SVR | 2.068 | 0.084 | 70.09% | 1.970 | 0.066 | 70.10% | 1.950 | 0.060 | 70.08% |
| | UNIF-LSR | 9.912 | 0.754 | 70.09% | 10.514 | 0.610 | 70.10% | 10.757 | 0.544 | 70.08% |
| | GMS-LSR | 3.623 | 0.210 | 92.40% | 3.826 | 0.225 | 91.91% | 3.859 | 0.168 | 91.95% |
| | LGS-LSR | 2.133 | 0.093 | 99.29% | 2.301 | 0.136 | 98.96% | 2.448 | 0.154 | 98.76% |
| | UNIF-KLSR | 2.303 | 0.101 | 59.79% | 3.062 | 0.095 | 60.07% | 3.095 | 0.099 | 59.89% |
| | MS-KLSR | 2.432 | 0.098 | 73.42% | 3.047 | 0.141 | 82.42% | 3.137 | 0.100 | 83.27% |
| | KRMS-KLSR | **1.993** | **0.083** | **100.00%** | **1.839** | **0.045** | **100.00%** | **1.817** | 0.065 | **100.00%** |
| 0.4 | SVR | 2.141 | 0.074 | 59.79% | 2.004 | 0.057 | 60.07% | 1.983 | **0.064** | 59.89% |
| | UNIF-LSR | 13.929 | 0.895 | 59.79% | 14.233 | 0.651 | 60.07% | 14.512 | 0.447 | 59.89% |
| | GMS-LSR | 4.522 | 0.384 | 84.24% | 4.859 | 0.336 | 83.98% | 4.889 | 0.260 | 84.20% |
| | LGS-LSR | 2.283 | 0.133 | 98.73% | 2.616 | 0.133 | 98.03% | 2.772 | 0.117 | 97.52% |

## D.3 REAL EXAMPLES

We illustrate the application of the proposed kernel-based robust Markov subsampling method to two real-world datasets: the NASDAQ stock dataset with economic indicators and an air quality dataset.

**Example 1**. To evaluate the performance of the proposed subsampling method on real-world financial data, we conduct an empirical analysis using a dataset comprising historical trading information from the NASDAQ market. The dataset is sourced from a public repository `https://www.kaggle.com/datasets/sail4karthik/nasdq-dataset` and integrates data from major financial providers, including Yahoo Finance, Federal Reserve Economic Data (FRED), Alpha Vantage, and Quandl. It encompasses the period from January 4, 2010 to October 25, 2024, containing daily open-high-low-close (OHLC) prices, trading volume, and key macroeconomic and market sentiment indicators for a designated NASDAQ-listed stock. The raw dataset contains a total of 3,914 daily observations. The primary objective of our analysis is to predict the next trading day's daily percentage return for this NASDAQ stock. Accordingly, the target variable is taken as the daily return $R_{t+1}$, calculated as $R_{t+1} = (P_{t+1} - P_t)/P_t$, where $P_t$ and $P_{t+1}$ denote the closing

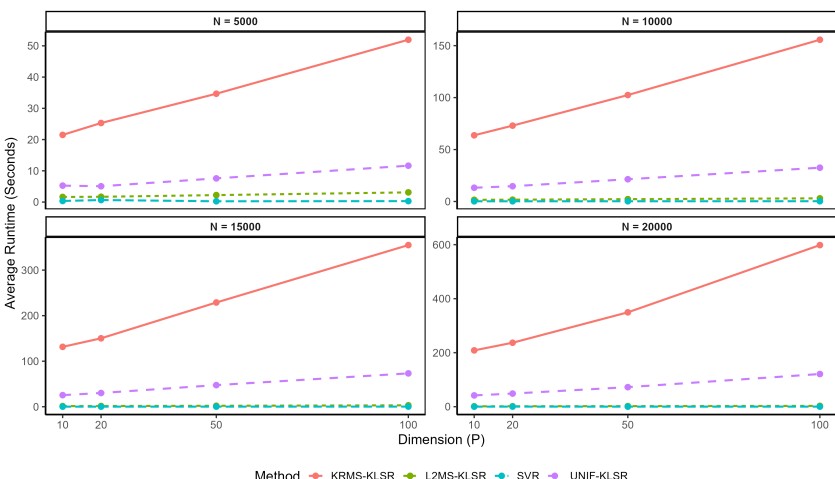

Figure 3: Runtime comparison of different methods with varying sample sizes N and dimensions P

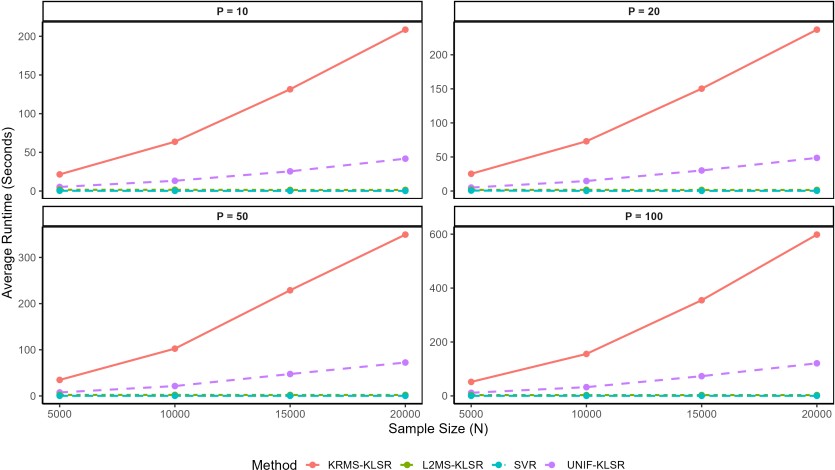

Figure 4: Runtime comparison of different methods with varying sample sizes N and dimensions P

prices on trading day $t$ and $t+1$, respectively. To construct a predictive model, we filter a set of predictors based on the established financial economic theory and common practices in empirical finance. These features are designed to capture diverse aspects of market dynamics and are broadly categorized as follows.

(A) Historical Market Behavior. We include the daily returns from the five preceding trading days (i.e., $R_t, \ldots, R_{t-4}$) to capture short-term momentum effects or potential mean-reversion patterns. A 5-day moving average ($MA_5$) and a 20-day moving average ($MA_{20}$) of closing prices are incorporated to represent short- and medium-term price trends, respectively. (B) Macroeconomic Conditions and Market Sentiment. We incorporate daily-frequency macroeconomic indicators and market sentiment proxies, including the CBOE volatility index (a measure of market risk expectations), a benchmark interest rate, the Effective Federal Funds Rate, the TED spread, an exchange rate, and commodity prices of Gold and Oil. These variables are widely recognized in the literature as external factors that may influence asset prices. After constructing these features, we remove observations with missing values. The remaining dataset is then divided into a training set (70% of observations) and a test set (the remaining 30%). We conduct regression analysis to predict the next-day return $R_{t+1}$, and compared the performance of the proposed KRMS-KLSR method with five competing methods: MS-KLSR, UNIF-KLSR, UNIF-LSR, GMS-LSR, and LGS-LSR. Results for AMSE and SD values over $M = 100$ replicates are summarized in Table 8.

Table 8: AMSE and SD values of six subsampling methods in NASDAQ stock data analysis

| $\theta$ | Method | $n=500$ AMSE | SD | $n=1000$ AMSE | SD | $n=1500$ AMSE | SD | $\theta$ | $n=500$ AMSE | SD | $n=1000$ AMSE | SD | $n=1500$ AMSE | SD |
|---|---|---|---|---|---|---|---|---|---|---|---|---|---|---|
| | UNIF-KLSR | 0.016 | 0.001 | 0.016 | 0.001 | 0.016 | 0.001 | | 0.042 | 0.007 | 0.039 | 0.006 | 0.039 | 0.005 |
| | MS-KLSR | 0.016 | 0.001 | 0.016 | 0.001 | 0.016 | 0.001 | | 0.021 | 0.003 | 0.021 | 0.002 | 0.020 | 0.002 |
| 0.0 | KRMS-KLSR | 0.016 | 0.001 | 0.016 | 0.001 | 0.016 | 0.001 | 0.2 | **0.016** | **0.001** | **0.016** | **0.001** | **0.016** | **0.001** |
| | UNIF-LSR | 0.016 | 0.001 | 0.016 | 0.001 | 0.016 | 0.001 | | 0.101 | 0.014 | 0.093 | 0.010 | 0.088 | 0.008 |
| | GMS-LSR | 0.016 | 0.001 | 0.016 | 0.001 | 0.016 | 0.001 | | 0.079 | 0.025 | 0.073 | 0.016 | 0.073 | 0.016 |
| | LGS-LSR | 0.016 | 0.001 | 0.016 | 0.001 | 0.016 | 0.001 | | 0.061 | 0.029 | 0.062 | 0.021 | 0.063 | 0.018 |
| | UNIF-KLSR | 0.025 | 0.004 | 0.024 | 0.003 | 0.024 | 0.003 | | 0.060 | 0.011 | 0.056 | 0.007 | 0.055 | 0.007 |
| | MS-KLSR | 0.017 | 0.001 | 0.017 | 0.001 | 0.017 | 0.001 | | 0.027 | 0.004 | 0.026 | 0.003 | 0.026 | 0.003 |
| 0.1 | KRMS-KLSR | **0.016** | **0.001** | **0.016** | **0.001** | **0.016** | **0.001** | 0.3 | **0.016** | **0.001** | **0.016** | **0.001** | **0.016** | **0.001** |
| | UNIF-LSR | 0.097 | 0.020 | 0.089 | 0.015 | 0.101 | 0.011 | | 0.110 | 0.018 | 0.095 | 0.011 | 0.092 | 0.009 |
| | GMS-LSR | 0.064 | 0.032 | 0.062 | 0.020 | 0.079 | 0.017 | | 0.085 | 0.023 | 0.080 | 0.013 | 0.078 | 0.011 |
| | LGS-LSR | 0.045 | 0.027 | 0.045 | 0.020 | 0.061 | 0.017 | | 0.066 | 0.025 | 0.071 | 0.019 | 0.071 | 0.014 |

We first evaluate the considered six subsampling methods on the original dataset without artificial contamination (i.e., contamination proportion $\theta = 0.0$). As shown in Table S7, in this uncontaminated scenario, six methods yield nearly identical AMSE values with low SD, indicating that the original dataset contains minimal extreme outliers. To investigate the robustness of the subsampling strategies, we artificially corrupt the training data. Specifically, for predictors $x_k$, we replace a proportion $\theta$ of observations with random values drawn from $w_k \sim U(2,3)$; for corresponding response variable $y$, we replace its observation with that drawn from $O_i \sim \mathcal{N}(1,3)$. As an illustration, we here consider three contamination proportions: $\theta \in \{0.1, 0.2, 0.3\}$, representing mild to severe data corruption scenarios.

The results under artificial contamination are presented in Table S7. Key findings include that (i) the proposed KRMS-KLSR method exhibits exceptional robustness, maintaining stable AMSE and SD values regardless of contamination levels and sample sizes, aligns with its uncontaminated performance, demonstrating its strong ability to mitigate contamination effects. (ii) The MS-KLSR method demonstrates consistent robustness, consistently outperforming the UNIF-KLSR method regardless of contamination levels and sample sizes, while effective, exhibits slightly less stability compared to the KRMS-KLSR method. (iii) the LGS-LSR performs best among linear methods. The GMS-LSR and UNIF-LSR methods suffer from significant performance deterioration under contamination, yielding higher AMSE values. (iv) The KRMS-KLSR and MS-KLSR methods maintain consistent performance regardless of sample sizes. Less robust methods show minor AMSE improvements with larger sample sizes at a high contamination level, but remain inferior to the KRMS-KLSR method. (v) While all six methods perform similarly on uncontaminated data, contamination scenarios clearly show KRMS-KLSR method's superiority in maintaining both accuracy and stability.

**Example 2**. To demonstrate the proposed method, we employ the Air Quality dataset, which comprises 9358 hourly averaged responses from an array of 5 metal oxide chemical sensors collected between March 2004 and February 2005. This dataset includes ground truth measurements for carbon monoxide (CO), non-methane hydrocarbons (NMHC), benzene, total nitrogen oxides (NOX), and nitrogen dioxide ($NO_2$), obtained from a co-located certified reference analyzer. Due to the high proportion of missing values in the raw data, we utilize a preprocessed version of the dataset curated by "cmertin" `https://github.com/cmertin/Machine_Learning` to ensure reliability for modeling.

The dataset is split into training set (70%) and test set (30%). In this example, we focus on predicting the hourly averaged $NO_2$ concentration (in $\mu g/m^3$), using the following predictor variables: month, hour, the five sensor responses (hourly averaged), temperature, relative humidity, and absolute humidity. We assess the proposed KRMS-KLSR method against several competing approaches: (i) kernel-based subsampling techniques: UNIF-KLSR and MS-KLSR, and (ii) linear regression-based subsampling techniques: UNIF-LSR, GMS-LSR, and LGS-LSR. Performance metrics: AMSE and SD values for three subsample sizes as well as four contamination levels are given in Table 9.

Similarly to Example 1, we first evaluate the considered six subampling methods for the original air quality dataset (i.e., uncontaminated, $\theta = 0.0$). From Table 9, we observe the following findings: (i) kernel-based subsampling methods usually outperform linear regression-based subsampling models

Table 9: AMSE and SD values of six subsampling methods in Air Quality data analysis

| | | $n = 500$ | | $n = 1000$ | | $n = 1500$ | | | $n = 500$ | | $n = 1000$ | | $n = 1500$ | |
|---|---|---|---|---|---|---|---|---|---|---|---|---|---|---|
| $\theta$ | Method | AMSE | SD | AMSE | SD | AMSE | SD | $\theta$ | AMSE | SD | AMSE | SD | AMSE | SD |
| 0.0 | UNIF-KLSR | **0.410** | 0.011 | **0.393** | **0.009** | **0.386** | **0.009** | 0.2 | 0.499 | 0.015 | 0.495 | **0.012** | 0.493 | **0.012** |
| | MS-KLSR | 0.418 | 0.014 | 0.400 | 0.011 | 0.395 | 0.010 | | 0.487 | 0.014 | 0.485 | 0.014 | 0.485 | 0.015 |
| | KRMS-KLSR | 0.443 | 0.018 | 0.425 | 0.014 | 0.419 | 0.013 | | **0.444** | 0.021 | **0.425** | **0.012** | **0.421** | 0.014 |
| | UNIF-LSR | 0.472 | **0.010** | 0.469 | 0.010 | 0.468 | 0.010 | | 0.573 | 0.026 | 0.569 | 0.020 | 0.568 | 0.018 |
| | GMS-LSR | 0.471 | 0.011 | 0.469 | 0.011 | 0.468 | 0.010 | | 0.502 | 0.016 | 0.504 | 0.015 | 0.504 | 0.015 |
| | LGS-LSR | 0.471 | **0.010** | 0.469 | 0.010 | 0.468 | 0.010 | | 0.476 | **0.012** | 0.476 | **0.012** | 0.475 | **0.012** |
| 0.1 | UNIF-KLSR | 0.475 | 0.016 | 0.470 | **0.012** | 0.468 | **0.011** | 0.3 | 0.518 | 0.016 | 0.514 | 0.014 | 0.513 | 0.013 |
| | MS-KLSR | 0.456 | 0.014 | 0.458 | 0.014 | 0.458 | 0.016 | | 0.506 | 0.015 | 0.506 | 0.015 | 0.505 | 0.017 |
| | KRMS-KLSR | **0.443** | 0.016 | **0.425** | 0.015 | **0.421** | 0.013 | | **0.450** | 0.017 | **0.455** | 0.018 | **0.458** | 0.017 |
| | UNIF-LSR | 0.532 | 0.024 | 0.528 | 0.019 | 0.526 | 0.017 | | 0.613 | 0.032 | 0.609 | 0.027 | 0.610 | 0.023 |
| | GMS-LSR | 0.489 | 0.015 | 0.489 | 0.014 | 0.488 | 0.012 | | 0.518 | 0.018 | 0.520 | 0.016 | 0.520 | 0.015 |
| | LGS-LSR | 0.477 | **0.012** | 0.474 | **0.012** | 0.474 | **0.011** | | 0.479 | **0.011** | 0.477 | **0.011** | 0.477 | **0.012** |

in that the former has smaller AMSE values than the latter regardless of sample sizes, and (ii) UNIF-KLSR method consistently achieves the lowest AMSE values regardless of subsample sizes in the presence of uncontaminated cases, demonstrating strong performance on uncontaminated data. The proposed KRMS-KLSR method yields slightly higher AMSE value, likely because its robustness leads to the exclusion of some informative observations in this contamination-free setting. These results indicate that UNIF-KLSR method behaves satisfactorily when applied to relatively uncontaminated datasets.

To assess the performance of the considered six subsampling methods in the presence of contaminated data, we artificially introduce outliers into the training dataset by replacing a proportion $\theta$ of observations. The outliers are generated as follows: predictors $x_k$ are drawn from the normal distribution $\mathcal{N}(-10, 3)$, and their corresponding responses $y$ from the normal distribution $\mathcal{N}(-3, 3)$. Mirroring Example 1, we consider three contamination levels: $\theta \in \{0.1, 0.2, 0.3\}$. The results for artificially corrupted datasets are given in Table 9. From Table 9, we have the following key findings. First, the proposed KRMS-KLSR method outperforms other methods in that the former has smaller AMSE values and the relatively small SD values than the latter regardless of contamination levels and subsample sizes, while the UNIF-KLSR and MS-KLSR methods perform better than linear regression-based three subsamplers in that the former consistently has smaller AMSE values than the latter regardless of contamination levels and subsample sizes. Second, the linear regression-based LGS-LSR method performs better than the UNIF-LSR and GMS-LSR approaches in terms of AMSE and SD values regardless of contamination levels and subsample sizes, which perform poorly under the considered settings. Third, the KRMS-KLSR method demonstrates exceptional stability, its AMSE values remain nearly unchanged even as contamination level $\theta$ increases, closely matching its performance on uncontaminated data ($\theta = 0$). This resilience is further confirmed by its low SD values, particularly at higher contamination levels $\theta$. Fourth, larger subsample size $n$ generally enhance or stabilize the performance of all subsampling methods. In summary, the KRMS-KLSR method demonstrates outstanding robustness to contamination, maintaining near-optimal accuracy across contamination levels while significantly outperforming competing subsampling methods.

# E  CONVERGENCE ANALYSIS AND PARAMETER SENSITIVITY ANALYSIS

To address concerns related to the convergence of the iterative optimization (Algorithm 1), the validity of subsampling, and the sensitivity to hyperparameters, we provide a comprehensive empirical analysis in this section. These simulation studies complement the theoretical discussion and further validate the robustness of the KRMS method. In the subsequent subsections, we generate datasets under case M1 of Experiment 2, with a contamination level of $\theta = 0.4$, and repeat the experiment 100 times.

## E.1  CONVERGENCE ANALYSIS OF THE ITERATIVE PROCEDURE

Figure 5 presents a robust evaluation of the iterative KRMS process under heavy data contamination, displaying the mean and 95% Confidence Interval (CI) across iterations ($\kappa$). The analysis

is structured along two independent aspects to simultaneously monitor model stability and performance. Figure 5a, which focuses on algorithmic convergence (parameter stability), employs the Mean Squared Prediction Change (MSPC), defined as $\left\| \hat{\mathbf{y}}^{(\kappa)} - \hat{\mathbf{y}}^{(\kappa-1)} \right\|_2^2 / n$, as the key metric. Figure 5b tracks generalization performance via RMSE computed on a clean testset. As shown in Figure 5, both the MSPC and the RMSE drop sharply within the first 3–5 iterations and stabilize thereafter. This empirical evidence strongly indicates that the proposed recursive updating strategy effectively corrects the initial pilot estimate $\alpha^{(0)}$, preventing divergence even when the initial sample contains outliers.

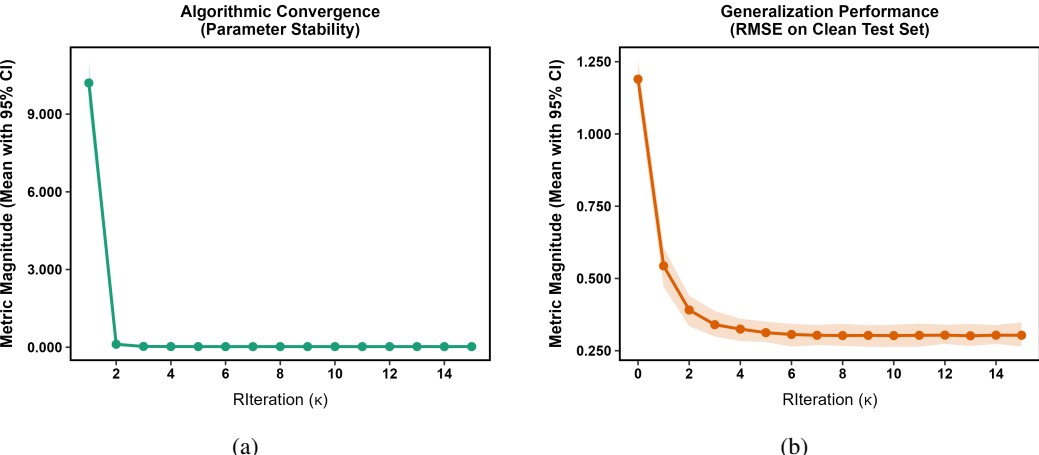

Figure 5: Convergence diagnosis of the Algorithm 1.

### E.2 VALIDATION OF THE SAMPLING MECHANISM

To illustrate the target distribution of our Markov subsampling procedure and demonstrate how the weights $w(\tilde{z}, \alpha)$ effectively down-weight contaminated observations, we visualize the sampling behavior in both the metric space and the feature space. Figure 6a displays the density distribution of the sampling metric $\log(w)$ for clean versus contaminated subsamples. Figure 6b visualizes the spatial distribution of the selected subsamples in a two-dimensional feature space, overlaid on the full contaminated dataset.

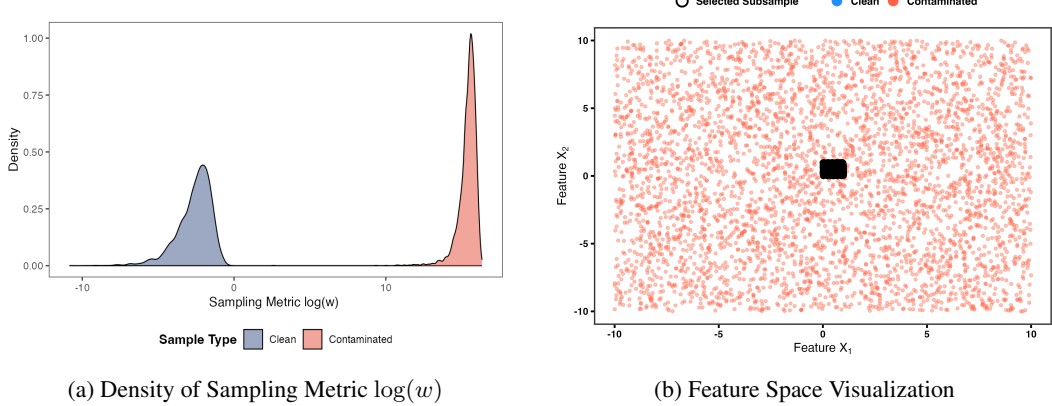

(a) Density of Sampling Metric $\log(w)$      (b) Feature Space Visualization

Figure 6: Visualization of the Subsampling Mechanism.

Figure 6a demonstrates a clear distinction between the weight distributions of clean and contaminated samples. Since the acceptance probability is proportional to $1/w(\tilde{z}, \alpha)$, the algorithm inherently favors selecting "clean" data. Furthermore, as depicted in Figure 6b, even under heavy

contamination, the subsampling algorithm predominantly selects nearly pure "clean" data points (shown in black). This provides additional evidence that the Markov chain effectively converges to the "clean" data distribution.

### E.3 PARAMETER SENSITIVITY ANALYSIS

We conduct sensitivity analyses on the subsample size $n_0$ and the burn-in period $t_0$. Figure 7a illustrates that as subsample size increases, the RMSE on the "clean" testset gradually decreases and eventually stabilizes. This indicates that a moderate number of subsamples is sufficient to achieve reliable performance, thereby substantially lowering computational cost. Figure 7b shows that the algorithm's performance remains highly stable across different burn-in periods. This observation suggests rapid mixing of the Markov chain, and demonstrates that the method is insensitive to the specific choice of $t_0$, which simplifies parameter tuning in practice.

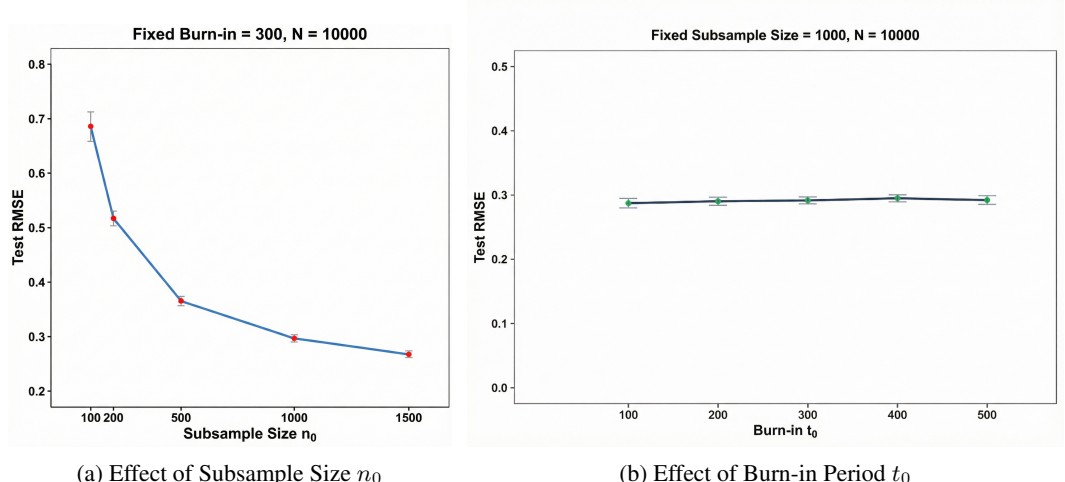

(a) Effect of Subsample Size $n_0$        (b) Effect of Burn-in Period $t_0$

Figure 7: Sensitivity Analysis

We assess the performance of KRMS with three alternative kernel functions: the Laplacian kernel $K(\boldsymbol{x}, \boldsymbol{y}) = \exp(-|\boldsymbol{x} - \boldsymbol{y}|/\sigma)$, the linear kernel, and the polynomial kernel $K(\boldsymbol{x}, \boldsymbol{y}) = (\boldsymbol{x}^\top \boldsymbol{y} + c)^d$. As shown in Table 10, while the Gaussian kernel achieves the best overall results, the Laplacian kernel remains competitive. In contrast, non-stationary kernels (linear and polynomial) perform notably worse. This is likely because non-stationary kernels produce values that depend on the absolute position of data points; as a result, outliers with large norms may be incorrectly selected, compromising robustness. We also investigate sensitivity to the bandwidth parameter $\sigma$ of the Gaussian kernel. Figure 8 shows the performance of KRMS across $\sigma = c/p$ for $c \in [0.1, 10]$. The results indicate that KRMS remains highly stable over a wide range of $c$, while the other two methods are noticeably sensitive to the bandwidth selection. In our experiments, the bandwidth of the Gaussian kernel is set according to the dimension-dependent rule: $\sigma = 1/p$ (Chang & Lin, 2011), which reflects the linear growth of squared Euclidean distances in high-dimensional space. The chosen value ($c = 1$) falls within the observed high-performance plateau, confirming that our parameter selection is both principled and non-arbitrary.

Table 10: Performance comparison of KRMS with different kernel for corrupted mechanism M1 in Experiment 2

| $\theta$ | Method | $n = 500$ | | | $n = 1000$ | | | $n = 1500$ | | |
|---|---|---|---|---|---|---|---|---|---|---|
| | | AMSE | SD | PSR | AMSE | SD | PSR | AMSE | SD | PSR |
| 0.1 | KRMS-Gaussian | 1.137 | **0.041** | **100.00%** | 1.098 | **0.036** | **100.00%** | 1.087 | **0.035** | **100.00%** |
| | KRMS-Laplacian | **1.134** | **0.041** | **100.00%** | **1.093** | 0.039 | **100.00%** | **1.085** | 0.039 | **100.00%** |
| | KRMS-Polynomial | 27.276 | 10.036 | 50.63% | 38.847 | 12.791 | 56.42% | 46.036 | 14.027 | 62.14% |
| | KRMS-Linear | 25.631 | 0.964 | 75.58% | 25.245 | 0.900 | 78.07% | 25.055 | 0.711 | 79.82% |
| 0.2 | KRMS-Gaussian | 1.153 | **0.044** | **100.00%** | 1.099 | **0.038** | **100.00%** | 1.091 | **0.035** | **100.00%** |
| | KRMS-Laplacian | **1.152** | 0.046 | **100.00%** | **1.104** | 0.040 | **100.00%** | **1.100** | 0.041 | **100.00%** |
| | KRMS-Polynomial | 29.191 | 22.964 | 29.15% | 40.328 | 30.571 | 37.06% | 56.794 | 32.144 | 44.62% |
| | KRMS-Linear | 29.833 | 0.548 | 41.96% | 29.746 | 0.547 | 47.15% | 29.708 | 0.472 | 51.71% |
| 0.3 | KRMS-Gaussian | 1.145 | 0.050 | **100.00%** | 1.104 | **0.041** | **100.00%** | 1.101 | **0.038** | **100.00%** |
| | KRMS-Laplacian | **1.135** | **0.048** | **100.00%** | **1.103** | 0.046 | **100.00%** | **1.095** | 0.042 | **100.00%** |
| | KRMS-Polynomial | 19.498 | 13.403 | 18.85% | 25.048 | 27.75 | 26.68% | 20.857 | 27.660 | 32.96% |
| | KRMS-Linear | 30.734 | 0.499 | 25.19% | 30.712 | 0.450 | 30.98% | 30.732 | 0.475 | 36.01% |
| 0.4 | KRMS-Gaussian | 1.149 | 0.045 | **100.00%** | **1.104** | 0.042 | **100.00%** | 1.106 | **0.042** | **100.00%** |
| | KRMS-Laplacian | **1.139** | **0.041** | **100.00%** | 1.107 | **0.033** | 99.99% | **1.104** | 0.043 | **100.00%** |
| | KRMS-Polynomial | 21.616 | 10.487 | 11.88% | 18.749 | 12.201 | 19.05% | 19.390 | 15.188 | 24.91% |
| | KRMS-Linear | 31.086 | 0.518 | 16.69% | 31.087 | 0.483 | 22.57% | 31.084 | 0.475 | 27.13% |

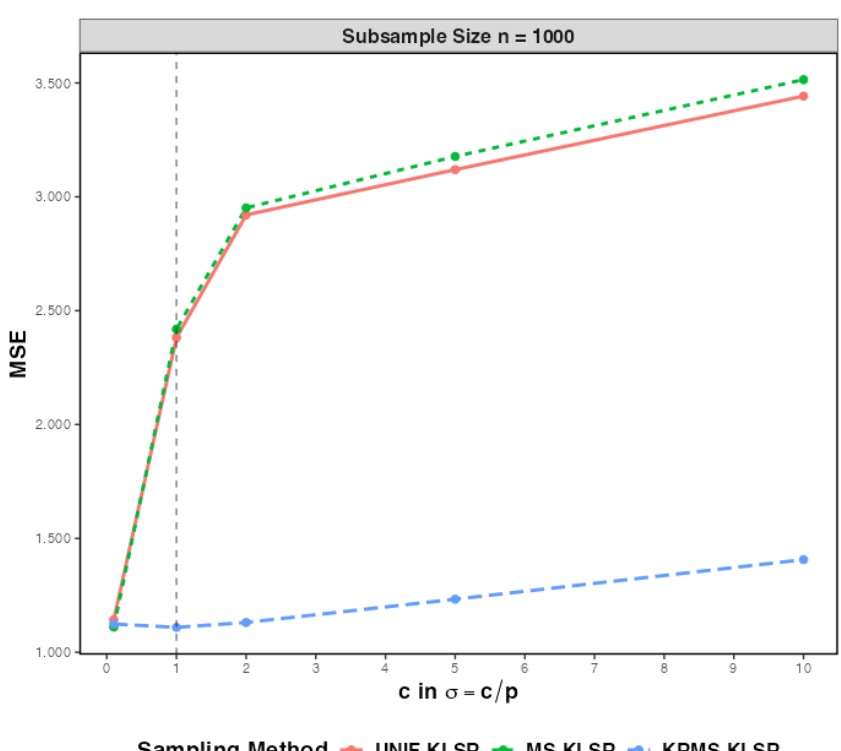

Figure 8: Bandwidth Sensitivity Analysis

