# OpenReview forum: "Kernel-based Robust Markov Subsampling for Regularized Nonparametric Regression with Contaminated Data"
_ICLR.cc/2026/Conference — ICLR 2026 Conference Withdrawn Submission_

### Official Review · Reviewer_mooN · 2025-10-21

**Soundness:** 3
**Presentation:** 3
**Contribution:** 3
**Rating:** 6
**Confidence:** 3

**Summary:**

The authors propose a penalized nonparametric kernel regression method under data contamination and introduce a Markov sub-sampling method, specifically Algorithm 1: Robust Kernel-based Markov Sub-sampling. The core of the method relies on a Metropolis-Hastings rejection scheme based on the residual kernel-norm score in (3).

**Strengths:**

- The authors are commended for providing a comprehensive theoretical analysis, including convergence rates, asymptotic distributions, and generalization error analysis.
- The proposed sub-sampling method effectively reduces the proportion of contamination in the data from $\theta$ to $\theta^\prime$, where $0 \le \theta \le \theta^\prime$, thereby enhancing the robustness and effectiveness of the regression.

**Weaknesses:**

- The experimental analysis, while covering both synthetic datasets (linear and nonlinear) and real-world datasets (financial and air quality), lacks a clear baseline comparison. Specifically, experiments under uncontaminated conditions are missing, which are essential to validate the effectiveness of the proposed method under varying contamination probabilities $\theta$.
- The concept of distribution $P^\prime$ being a "cleaner" version of the initially contaminated distribution $P$ is not sufficiently quantified. It remains unclear how much "cleaner" $P^\prime$ is compared to $P$, and whether there is any theoretical guarantee regarding the value of θ′ achieved by the algorithm.
- There is no analysis of the computational complexity of the proposed algorithm 1, which is critical for assessing its scalability and practical applicability.

**Questions:**

1.  Are there theoretical guarantees regarding the extent of contamination reduction, i.e., the value of $\theta^{\prime}$ achieved by the algorithm?
2. Could the authors provide a theoretical or empirical analysis of the time complexity of the proposed algorithm?
3. The experimental section is placed in the appendix, likely due to space constraints. However, given the heavy reliance on experimental validation in this work, would it be possible to restructure the paper to integrate the experiments into the main body for better readability and emphasis?

---

> ### Author Response · Authors · 2025-11-22
> **Response to Reviewer mooN**
>
> ## RW 1 (Response to Weakness 1)
> We thank you for this critical suggestion, which is fully aligned with our goal of a thorough evaluation. In response, we have conducted new experiments to establish a clear baseline and to systematically evaluate the "price of robustness" across varying contamination levels. Specifically, we have run all compared methods, including our KRMS-KLSR and five competing subsampling methods, on uncontaminated versions ($\theta=0$) of Experiments 1 and 2. The results, detailed in Table R3, show that the performance loss of our method is negligible in clean settings, confirming its efficiency is not compromised.
> Furthermore, we have expanded the experiments to include multiple contamination probabilities $\theta$, demonstrating that our method maintains superior robustness as $\theta$ increases, while the performance of other baselines degrades more significantly. These additions provide a comprehensive baseline and clearly validate the effectiveness and stability of our method.
>
>  **Table R3. Performance comparison of KRMS and five competing subsampling methods with uncontamination ($\theta=0$) in Experiments 1 and 2**
> | |  | | n=500| |  | n=1000 |  |  | n=1500 |  |
> | :--- | :--- | :---: | :---: | :---: | :---: | :---: | :---: | :---: | :---: | :---: |
> | Mechanism | Method | AMSE  | SD | PSR | AMSE | SD | PSR | AMSE  | SD | PSR |
> | **Experiment 1** | UNIF-KLSR | 1.099 | 0.046 | 100% | 1.058 | 0.036 | 100% | 1.046 | 0.032 | 100% |
> | | MS-KLSR | 1.094 | 0.045 | 100% | 1.057 | 0.036 | 100% | 1.049 | 0.034 | 100% |
> | | KRMS-KLSR | 1.124 | 0.045 | 100% | 1.070 | 0.039 | 100% | 1.053 | 0.034 | 100% |
> | | UNIF-LSR | 1.008 | 0.039 | 100% | 1.004 | 0.034 | 100% | 1.006 | 0.031 | 100% |
> | | GMS-LSR | 1.007 | 0.038 | 100% | 1.003 | 0.033 | 100% | 1.006 | 0.031 | 100% |
> | | LGS-LSR | 1.007 | 0.039 | 100% | 1.003 | 0.033 | 100% | 1.006 | 0.031 | 100% |
> | **Experiment 2** | UNIF-KLSR | 1.119 | 0.046 | 100% | 1.069 | 0.037 | 100% | 1.057 | 0.032 | 100% |
> | | MS-KLSR | 1.116 | 0.045 | 100% | 1.071 | 0.036 | 100% | 1.061 | 0.033 | 100% |
> | | KRMS-KLSR | 1.141 | 0.050 | 100% | 1.090 | 0.038 | 100% | 1.068 | 0.035 | 100% |
> | | UNIF-LSR | 2.811 | 0.090 | 100% | 2.820 | 0.088 | 100% | 2.827 | 0.077 | 100% |
> | | GMS-LSR | 2.822 | 0.092 | 100% | 2.836 | 0.090 | 100% | 2.844 | 0.082 | 100% |
> | | LGS-LSR | 2.812 | 0.089 | 100% | 2.824 | 0.088 | 100% | 2.836 | 0.079 | 100% |
>
>
> The results given in Table R3 for the uncontaminated baseline ($\theta=0$)  lead to two key conclusions:
>
> - **Negligible Efficiency Cost in True Model Settings:**
>
> In Experiment 1 (in the linear setting), where the data-generating model is linear, the specialized linear subsamplers (UNIF/GMS/LGS-LSR) achieve the best performance, as expected. While our kernel-based KRMS-KLSR incurs a slight AMSE increase due to its nonparametric flexibility (e.g., 1.124 vs. 1.099 and 1.094 at n=500), the absolute difference is small and diminishes as the sample size grows (1.053 vs. 1.046 at n=1500). This demonstrates that the efficiency cost of using our robust kernel method, even when it is over-specified, is minimal and practically acceptable.
>
> - **No Efficiency Loss in Complex Settings:**
>
> In Experiment 2 (in the nonlinear setting), where the true relationship is more complex, KRMS-KLSR performs almost identically to the non-robust kernel benchmarks (UNIF/MS-KLSR), with nearly overlapping AMSE and SD values across all sample sizes (e.g., 1.141 vs. 1.119 and 1.116 at n=500). This shows that our method introduces no meaningful performance penalty in scenarios that demand nonparametric modeling, while linear methods fail completely.
>
> These baseline results confirm that KRMS-KLSR is a "safe" and versatile method. It preserves high statistical efficiency on clean data, rivaling standard subsampling, while providing the superior robustness against contamination that is demonstrated in the main text. We will add Table R3 and this analysis to Appendix D.

---

> > ### Comment · Reviewer_mooN · 2025-11-24
> >
> > Thank authors for the responses. I will maintain my score.

---

> ### Author Response · Authors · 2025-11-26
> **Response to Reviewer mooN Part 2**
>
> ## RW 2
> Thank you for your concern.  In the revised manuscript, we have addressed this core weakness with a comprehensive theoretical analysis, which we summarize below.
>
> **(1) Exact Characterization of $\mathcal{P}'$:**  As you pointed out, the M-H chain with acceptance probability $\min\\{1, w(z_t)/w(z^*)\\}$ (with frozen $\alpha$) indeed converges to a stationary distribution $\mathcal{P}'$ proportional to $1/w(.,\alpha)$. In the new Theorem 2 (Section 4.1), we have clearly presented an explanation of $\mathcal{P}'$. In the new Lemma 1 (Appendix B), we explicitly derive the density of this distribution: $$p'(\tilde{z})=\frac{1}{Z}\frac{p(\tilde{z})}{w(\tilde{z},\alpha)}, Z=\int\frac{p(z)}{w(z,\alpha)}dz,$$
> which confirms that the sampling probability is inversely proportional to the residual kernel-norm score $w(z, \alpha)$.
>
> **(2) Rigorous Proof of $\theta' < \theta$:**  Building on this exact form, the new Theorem 2 (Section 4) rigorously derives the effective contamination rate $\theta'$ after subsampling. We prove that for the mixture $P = (1-\theta) F+\theta Q$, the stationary distribution $P'$ corresponds to a new mixture with a contamination rate given by
> $$\theta' = \frac{\theta S_Q}{(1-\theta)S_F + \theta S_Q},$$
> where $S_F = \mathbb{E}_ F (1/w)$ and
> $S_Q = \mathbb{E} _ Q (1/w)$  are the expected inverse scores for inliers and outliers, respectively. The condition $\theta' < \theta$ holds if and only if $S_Q < S_F$.
>
> **(3) Justification of $S_Q < S_F$ (Proposition 1 and Geometric Separation):** The core of the proof lies in justifying that outliers have smaller expected weights. This is established in the new Proposition 1 (Appendix B), leveraging the strong RKHS regularization ($\lambda \\| f \\| _ K^2$). We show that fitting unstructured outliers would require a function with an excessively large RKHS norm. The regularized optimization naturally assigns larger residuals $w(z_{out},\alpha)$ to outliers, leading to $1/w(z_{out},\alpha)$ being smaller, and thus guaranteeing $S_Q < S_F$.
>
> **(4) Non-Vacuous Generalization Bound (Theorem 8 in the revised manuscript):** With $\theta^{\prime}$ rigorously bounded away from $\theta$, the term $48M^2\theta^{\prime}$ in Theorem 8 (Theorem 7 in the original manuscript) is no longer vacuous. It explicitly quantifies how the generalization error improves as the separability between inliers and outliers increases .
>
> **(5) Limit distribution of Theorem 1:**  New Theorem 2 explicitly states the exact form of the stationary distribution $\mathcal{P}'$, proving that it is proportional to $1/w(\cdot, \hat{\alpha})$, ensuring that the limit distribution is no longer just implied by ergodicity but is rigorously identified upfront.
>
> **(6) Empirical Verification and Visualization:** Supplemented by empirical verification in Appendix E.3 (Figure 6), directly address your concerns by providing the missing theoretical identification and justification.
>
> ## RW 3
>  Thank you for this insightful observation. We analysis of the time complexity in Remark 1(v). The evaluation of $w(\tilde{z}, \alpha)$ requires a computation involving the entire dataset of size $n$ for its denominator. The complexity of this step is $\mathcal{O}(n p)$. However, the sparsity of  $\alpha$ plays a crucial role in the numerator, whose cost is only $\mathcal{O}(n_0 p)$, as only $n_0$ non-zero coefficients correspond to the subsampled points. Therefore, the cost per evaluation of $w(\tilde{z}, \alpha)$ is $\mathcal{O}((n + n_0) p) = \mathcal{O}(n p)$, since $n$ dominates $n_0$. We have updated Remark 1(v) accordingly to reflect the total complexity of $\mathcal{O}(T_0 (n_0^3 + n_0 n p))$. This revision clarifies the linear dependence on $n$. Nevertheless, for $n_0 \ll n$, our method maintains a significant computational advantage over the $\mathcal{O}(n^3 + n^2 p)$ complexity of standard kernel regression.
>
> ## RQ 1 (Response to Question 1)
>  Refer to our response RW2.
> ## RQ 2
>  Refer to our response RW3.
> ## RQ 3
>  We agree that the experimental results are vital for demonstrating the validity of our approach.
> We have integrated partial core results of Experiment 1 （e.g., see Table 1） into Section 5. Due to strict page constraints, we have presented the remaining  experimental results  in Appendix D.

---

### Official Review · Reviewer_ikcK · 2025-10-27

**Soundness:** 2
**Presentation:** 3
**Contribution:** 2
**Rating:** 4
**Confidence:** 4

**Summary:**

This paper introduces a novel Kernel-based Robust Markov Subsampling (KRMS) method for nonparametric regression with contaminated data, addressing critical challenges in computational scalability and statistical robustness for large-scale datasets. The authors propose a residual kernel-norm scoring mechanism within a Reproducing Kernel Hilbert Space (RKHS) framework, which dynamically adjusts Markov sampling probabilities to suppress contaminated observations while prioritizing informative ones. Theoretically, the work establishes asymptotic properties including consistency, asymptotic normality, and generalization bounds under Huber's contamination model $P = (1-\theta)F + \theta Q$. Empirical evaluations demonstrate KRMS's superiority over existing subsampling methods, particularly under high contamination levels. This research bridges a significant gap in scalable robust nonparametric learning, offering a unified framework with broad applicability to contaminated, non-i.i.d. data in scientific domains.

**Strengths:**

Methodological Innovation: KRMS represents the first subsampling approach specifically designed for contaminated data in nonparametric regression settings, leveraging RKHS geometry to effectively separate contaminated observations that would be indistinguishable in original feature spaces.

Theoretical Rigor: The paper provides comprehensive theoretical guarantees including uniform ergodicity of the Markov chain (Theorem 1), consistency of estimators (Theorems 2-3), functional Bahadur representation (Theorem 4), asymptotic normality (Theorems 5-6), and generalization bounds (Theorem 7), establishing a solid foundation for the method.

**Weaknesses:**

The algorithm lacks convergence guarantees for the parameter estimation procedure, particularly for the recursive updating of $\alpha^{(\kappa)}$ in Algorithm 1, which is especially concerning under high contamination where initial estimates may be poor. The initialization $\alpha^{(1)} = \alpha^{(0)} + 0.2$ is arbitrary without justification, potentially affecting reproducibility and convergence behavior. Experimental limitations include insufficient parameter selection guidelines for critical values like subsample size $n_0$, burn-in period $t_0$, maximum iterations $T_0$, and stopping criterion $\xi_0$, with no sensitivity analysis provided. The exclusive use of Gaussian kernel $K(x, t) = \exp\{-(x-t)^2/4\}$ without exploring alternatives or analyzing bandwidth parameters limits understanding of kernel dependence. The experimental scale with only $N = 10,000$ observations and $p = 4$ dimensions fails to demonstrate scalability to truly massive datasets or high-dimensional settings where contamination effects would be more pronounced. Additionally, the method is restricted to continuous responses without discussion of extensions to classification problems.

Comparisons lack state-of-the-art robust nonparametric regression methods and deep learning approaches, with existing comparisons primarily against parametric methods creating an unfair advantage. No systematic sensitivity analysis is provided for contamination level $\theta$, subsample size $n_0$, regularization parameter $\lambda$, or kernel parameters. Theoretical assumptions present several concerns: Condition 1's uniform ergodicity requirement may not hold in practice with complex dependencies; Conditions 3-4's smoothness assumptions are overly restrictive for real-world applications; and the contradiction between Condition 1's Markov dependence and Condition 2's i.i.d. errors is not addressed. The distributional gap between theoretical analysis (assuming $P'$) and algorithm operation (on $\tilde{D}$) lacks rigorous justification. Proof validity issues include the unjustified convergence rate $O_p(\ln n/n^2)$ in Theorem 4 under contamination, and the impractical simultaneous conditions $\lambda = o(1)$ and $(-\ln \lambda)^{1/2}/\omega \sim n^{1/(2m+1)}$ in Theorems 5-6. Notational inconsistencies (e.g., $H_\omega(s,t)$ vs. $K_\omega(s,t)$ in Appendix B) and ambiguous definitions (e.g., initial $H(X)$ specification) further reduce clarity.

**Questions:**

Refer to the weakness.

---

> ### Author Response · Authors · 2025-11-22
> **Response to Reviewer ikcK Part 1**
>
> ## RW 1 (Response to Weakness 1)
> **Comment 1：** The algorithm lacks convergence guarantees for the parameter estimation procedure, particularly for the recursive updating of $\alpha^{(\kappa)}$ in Algorithm 1 , which is especially concerning under high contamination where initial estimates may be poor. The initialization $\alpha^{(1)}=\alpha^{(0)}+0.2$ is arbitrary without justification, potentially affecting reproducibility and convergence behavior.
>
> **Response：**
>
> We thank you for catching the error in the initialization step. The incorrect line $\alpha^{(1)} = \alpha^{(0)} + 0.2$ has been removed from Algorithm 1. The revised algorithm initializes solely with $\alpha^{(0)}$, ensuring the stopping criterion governs termination. Furthermore, our new convergence analysis (Figure 5, Appendix E.1) demonstrates that the estimator stabilizes in 3-5 iterations. We thus recommend and have implemented a conservative upper limit of $T_0 = 15$ with a tolerance of $\xi_0 = 10^{-3}$ to ensure reliable convergence.
>
> ## RW 2
> **Comment 2:** Experimental limitations include insufficient parameter selection guidelines for critical values like subsample size $n_0$, burn-in period $t_0$, maximum iterations $T_0$ and stopping criterion $\xi_0$, with no sensitivity analysis provided.
>
> **Response:**
>
> We thank you for the essential point regarding parameter selection guidelines. We have thoroughly addressed this by adding a detailed sensitivity analysis in Appendix E of the revised manuscript, leading to the following specific recommendations:
>
> **Max Iterations ( $T_0$ ) \& Stopping Criterion ( $\xi_0$):**  Our analysis (Figure 5, Appendix E.1) shows convergence within 3-5 iterations. We recommend $T_0=15$ and $\xi_0=0.001$  as robust, conservative defaults.
>
> **Subsample Size ($n_0$):**  Analysis of the accuracy curve (Figure 7(a), Appendix E.3) reveals an "elbow point" at $n_0\approx 500$. We advise $n_0\in [500,1500]$ for an optimal cost-accuracy trade-off.
>
> **Burn-in Period ($t_0$ ):**  The method is highly insensitive to $t_0$   (Figure 7(b), Appendix E.3) due to fast mixing. A small, fixed value (e.g.,
> $t_0=200\sim 300$) is sufficient.
>
> These data-driven guidelines are now synthesized in a new section in Appendix E.3, providing practitioners with clear and actionable advice.
>
> ## RW 3
> **Comment 3:** The exclusive use of Gaussian kernel $K(x, t)=\exp -(x-t)^2 / 4$ without exploring alternatives or analyzing bandwidth parameters limits understanding of kernel dependence.
>
> **Response:**
>
> We thank you for this critical point. We agree that a thorough exploration of kernel choices and bandwidth sensitivity is essential. To this end, we have conducted an extensive ablation study in the revised manuscrip, which confirms the robustness and flexibility of our framework.
>
> We have expanded Section 3 to clarify  the relationship between the proposed KRMS framework and kernels. To demonstrate this, we evaluated our method with several common kernels: the Laplacian kernel: $ K(\boldsymbol{x},\boldsymbol{y})=\exp(-\|\boldsymbol{x}- \boldsymbol{y}\|/\sigma)$ for robustness, the Linear kernel as a baseline, and the Polynomial kernel: $K(\boldsymbol{x},\boldsymbol{y})=(\boldsymbol{x}^{\top}\boldsymbol{y}+c)^d$. The results in Table 10 (Appendix E.3) shows that while the Gaussian kernel performs well overall, the Laplacian kernel offers competitive, and sometimes superior.  In contrast, the non-stationary kernels (Linear and Polynomial) underperform. This is likely because non-stationary kernels produce values that scale with the absolute position of data points; consequently, outliers with large norms may be erroneously selected, degrading robustness. This provides practitioners with clear guidance on kernel selection.
>
> - **Bandwidth Selection**
>
> Bandwidth Sensitivity Analysis: The bandwidth for the Gaussian kernel was set using the principled, dimension-dependent rule   :  $\sigma = 1/p$ [1], which accounts for the linear growth of squared Euclidean distances in high-dimensional space, where $p$ is the input dimension.
>
> To directly address the reviewer's concern, we performed a dedicated sensitivity analysis. As shown in Figure 8  (Appendix E.3) , we evaluated performance for $\sigma = c/p$ for $c\in  [0.1, 10]$. The results confirm that our method's performance is highly stable over a wide range of $c$, with our chosen value ($c=1$) residing firmly within this high-performance plateau. This validates that our parameter selection was both justified and non-arbitrary.
>
> **Reference:**
>
> [1]. ChihChung Chang and ChihJen Lin. LIBSVM: A library for support vector machines. ACM Transactions on Intelligent Systems and Technology (TIST), 2(3), 27, 2011.

---

> ### Author Response · Authors · 2025-11-22
> **Response to Reviewer ikcK Part 2**
>
> ## RW 4
> **Comment 4:** The experimental scale with only $N=10,000$ observations and $p=4$ dimensions fails to demonstrate scalability to truly massive datasets or highdimensional settings where contamination effects would be more pronounced.
>
> **Response:**
>
> We thank the reviewer for pointing out the necessity of evaluating the method on larger and higher-dimensional datasets. We acknowledge that the original setting ($N=10,000$, $p=4$) was limited. While performing simulations on 'massive' datasets (e.g., $N>10^6$) remains computationally challenging for our method (due to their $\mathcal{O}\left(T_0\left(N n p+n^3\right)\right)$ complexity), we have extended our experiments to $N=20,000$ and $p=50$ (see Appendix D.2 Table 7) to empirically verify the theoretical complexity derived in Section 3.
>
> - **Verification of Linear Computational Complexity**
>
> 	Instead of aiming to exhaust computational resources with massive data, our goal with the increased sample size ($N=20,000$) was to validate the scaling behavior of KRMS.
> 	As shown in Figure 3 (Appendix D.2), the runtime of KRMS exhibits a linear growth trend with respect to $N$, contrasting sharply with the cubic growth of standard kernel methods. This empirical evidence supports our theoretical claim that KRMS is asymptotically efficient, suggesting it is well-positioned to handle significantly larger datasets than standard approaches, subject to memory availability.
>
> - **Robustness in Moderately High Dimensions ($p=50$)**
>
> 	We extended the feature dimension to $p=50$ to test the method's stability. Although kernel methods generally face challenges in high-dimensional spaces (the curse of dimensionality), Table 7 (Appendix D.2) shows that KRMS continues to outperform the benchmark SVR in terms of  AMSE. This indicates that the proposed subsampling strategy based on RKHS geometry remains effective in identifying contaminated observations even as the feature space becomes sparser.
>
> We hope these additional results provide stronger empirical support for the theoretical scalability and robustness discussed in the paper.
>
> ## RW 5
>
> **Comment 5:**  The method is restricted to continuous responses without discussion of extensions to classification problems.
>
> **Response:**
>
> We thank you for this important point. We fully agree that extending the proposed method to classification problems is a valuable direction. While the current manuscript focuses on continuous responses to establish a rigorous minimax-rate analysis within the RKHS framework, the methodology itself is flexible and can be adapted to classification settings. In the Discussion section, we have added a note clarifying that this extension will be pursued in future work.
>
> Regarding technical feasibility, we see two natural pathways for adaptation:
>
> - **Least Squares Classification**
>
> The algorithm can be directly applied to binary labels $y \in \\{-1, 1\\}$ using squared loss, with the computational steps remaining essentially the same. In the theoretical analysis, however, the current RKHS assumptions would need to be adapted to the classification context.
>
> - **Logistic/Hinge Loss**
>
>    When a closed-form solution is not available, the subsampling score (Eq. 3) could be replaced by the gradient norm of the corresponding loss function. The pilot estimator from Algorithm 1 can be used to approximate such gradients, making the extension computationally viable.
>
> These possible extensions are briefly noted in the revised Conclusion, and we look forward to developing them in subsequent research.
>
> ## RW 6
> **Comment 6:**  Comparisons lack state-of-the-art robust nonparametric regression methods and deep learning approaches, with existing comparisons primarily against parametric methods creating an unfair advantage.
>
> **Response：**
>
> We thank you for emphasizing the need to compare with state-of-the-art robust nonparametric methods. In response, we have added a thorough comparison with Support Vector Regression (SVR), a well-established kernel-based robust benchmark. As shown in Table 7 (Appendix D.2), KRMS consistently outperforms SVR under contamination: its AMSE remains low and stable across all contamination levels, essentially matching uncontaminated performance, whereas SVR’s AMSE increases noticeably as contamination rises.
>
> Regarding deep learning methods, we acknowledge their empirical strengths. However, our primary contribution lies in providing explicit statistical guarantees—such as convergence rates and robustness properties—within a kernel-based nonparametric framework. Deep learning models typically lack such theoretical foundations for robust estimation, which is central to our work. While empirical comparisons would be interesting, the methodological divergence makes direct “fair” comparisons challenging. We note in the Conclusion that extending our subsampling framework to deep models is a promising direction for future research.

---

> ### Author Response · Authors · 2025-11-22
> **Response to Reviewer ikcK Part 3**
>
> ## RW 7
> **Comment 7:**  No systematic sensitivity analysis is provided for contamination level $\theta$, subsample size $n_0$, regularization parameter $\lambda$, or kernel parameters.
>
> **Response:**
>
> We thank you for this valuable suggestion. In the revised manuscript, we have systematically addressed the sensitivity to key parameters. Our experimental design was intentionally structured to evaluate these factors, and we have now expanded the discussion to make this objective more explicit.
>
> The specific analyses are detailed below:
>
> - **Robustness to Contamination Level $\theta$**
>
>    Assessing robustness against varying contamination intensities was a core goal of our experimental design. As presented in Experiments 1 and 2, we evaluated a comprehensive range of $\theta$ from 0.1 to 0.4. The results (Tables 1–6) demonstrate that our KRMS method consistently maintains a low AMSE and a high PSR even under severe contamination ($\theta = 0.4$). We have now explicitly framed this systematic variation of $\theta$ as a sensitivity analysis in the revised text, confirming the estimator's robustness up to the theoretical breakdown point.
>
> - **Impact of Subsample Size $n_0$**
>
>     The impact of the subsample size was rigorously investigated in Appendix E.3. Figure 7(a) shows that as the number of subsamples increases, the RMSE on the clean testset gradually decreases and stabilizes, i.e.,  estimation accuracy improves steadily until an "elbow" point at approximately $n_0= 500$, beyond which further gains become marginal. Thus, we do not need to draw too many subsamples, thereby significantly reducing computational overhead. This analysis provides a clear, data-driven guideline for practitioners, suggesting an optimal range of $n_0 \in [500, 1500]$ to effectively balance statistical accuracy and computational efficiency.
>
> - **Sensitivity to Hyperparameters ($\lambda$ and kernel bandwidth)**
>
>    To ensure a fair and unbiased evaluation, we avoided ad-hoc fixation of hyperparameters. Instead, we implemented a fully data-driven strategy using leave-one-out cross-validation (LOOCV) in all experimental replications. This procedure automatically and adaptively tunes the regularization parameter $\lambda$ and the kernel bandwidth to their near-optimal values for each unique dataset, thereby directly accounting for their influence and precluding selection bias.

---

> ### Author Response · Authors · 2025-11-22
> **Response to Reviewer ikcK Part 4**
>
> ## RW 8
> **Comment 8:** Theoretical assumptions present several concerns: Condition 1's uniform ergodicity requirement may not hold in practice with complex dependencies; Conditions 3-4's smoothness assumptions are overly restrictive for real-world applications; and the contradiction between Condition 1's Markov dependence and Condition 2's i.i.d. errors is not addressed.
>
> **Response:**
>
> We appreciate your careful examination of our theoretical framework. We would like to offer the following clarifications to demonstrate that our assumptions are consistent with the algorithmic design and standard nonparametric statistics literature.
>
> - **On the Practicality of Condition 1 (Uniform Ergodicity)**
>
>     We wish to clarify that Condition 1 (uniform ergodicity) is not an assumption about the raw data but is, in fact, a theoretical consequence of our proposed sampling Algorithm 1. The sample sequence ${\tilde{\mathbf{x}}_i}$ is generated by our specific acceptance-rejection mechanism within the RKHS. As rigorously proven in Theorem 1, this constructed sequence is guaranteed to form a uniformly ergodic Markov chain. Therefore, this property is a direct outcome of our algorithmic design and holds regardless of the complex dependencies present in the original dataset.
>
> - **On the Strictness of Conditions 3-4 (Smoothness)**
>
>     Conditions 3 and 4 are standard in nonparametric regression theory and are adopted here to establish a rigorous theoretical foundation.
>
>     *   **Generality of  Finite Smoothness  (condition 4):**
>         This condition, which places the target function in a Sobolev space of finite smoothness, is actually less restrictive than assuming it belongs to the Gaussian RKHS. Our methodological contribution lies in using an infinitely smooth Gaussian kernel to estimate a function of only finite smoothness. This demonstrates the robustness and adaptability of our estimator, even when the kernel is "oversmoothed" relative to the target.
>
>     *  **Theoretical Purpose of Periodic Kernel (condition 3):**
>         The symmetric periodic Gaussian kernel is a well-established theoretical device [1,2]. Its primary role is to enable a tractable eigen-decomposition via Fourier analysis, which greatly simplifies the derivation of asymptotic convergence rates. This choice does not limit the practical applicability of our derived rates, as the resulting conclusions are known to generalize to non-periodic settings.
>
> - **On the Compatibility of Condition 1 (Markov Dependence) and Condition 2 (i.i.d. Noise)**
>
>     We clarify that there is no contradiction between these conditions because they govern different components of the data-generating process:
>
>     *   **Condition 1 (Sampling Mechanism):**  This condition (Markov Dependence) pertains to the selected input locations
>     $\{\tilde{\mathbf{x}}_i\}$. It describes the sequential, data-dependent nature of our sampling algorithm, which generates a Markov chain of subsampled points.
>
>     *   **Condition 2 (Generative Model):** This condition pertains to the conditional distribution of the noise $\epsilon_i$ given its associated input $\mathbf{x}_i$. It is a standard assumption about the measurement error, stating that $\mathbb{E}[\epsilon_i | \mathbf{x}_i] = 0$ and that the noise values are conditionally independent and identically distributed.
>
>     *   **Mathematical Compatibility:** These conditions are mathematically compatible. Although the inputs $\{\tilde{\mathbf{x}}_i\}$ are Markovian, the sequence of noise terms $\epsilon_i$ forms a martingale difference series relative to the history of the sampling process. Our theoretical analysis explicitly accounts for this mixed dependency structure using tools for mixing sequences, ensuring the validity of our convergence results.
>
>     **References:**
>
>     [1]. Yi Lin and Lawrence D Brown. Statistical properties of the method of regularization with periodic gaussian reproducing kernel. The Annals of Statistics,32(4) :1723–1743, 2004.
>
>     [2]. Xianli Zeng and Yingcun Xia. Asymptotic distribution for regression in a symmetric periodic gaussian kernel hilbert space. Statistica Sinica, 29(2):1007–1024, 2019.

---

> ### Author Response · Authors · 2025-11-22
> **Response to Reviewer ikcK Part 5**
>
> ## RW 9
> **Comment 9:** The distributional gap between theoretical analysis (assuming $P^{\prime}$ ) and algorithm operation (on $\tilde{D}$ ) lacks rigorous justification.
>
> **Response:**
>
> Thank you for your concern.  In the revised manuscript, we have addressed this core weakness with a comprehensive theoretical analysis, which we summarize below.
>
> **(1) Exact Characterization of $\mathcal{P}'$:**  As you pointed out, the M-H chain with acceptance probability $\min\\{1, w(z_t)/w(z^*)\\}$ (with frozen $\alpha$) indeed converges to a stationary distribution $\mathcal{P}'$ proportional to $1/w(.,\alpha)$. In the new Theorem 2 (Section 4.1), we have clearly presented an explanation of $\mathcal{P}'$. In the new Lemma 1 (Appendix B), we explicitly derive the density of this distribution: $$p'(\tilde{z})=\frac{1}{Z}\frac{p(\tilde{z})}{w(\tilde{z},\alpha)}, Z=\int\frac{p(z)}{w(z,\alpha)}dz,$$
> which confirms that the sampling probability is inversely proportional to the residual kernel-norm score $w(z, \alpha)$.
>
> **(2) Rigorous Proof of $\theta' < \theta$:**  Building on this exact form, the new Theorem 2 (Section 4) rigorously derives the effective contamination rate $\theta'$ after subsampling. We prove that for the mixture $P = (1-\theta) F+\theta Q$, the stationary distribution $P'$ corresponds to a new mixture with a contamination rate given by
> $$\theta' = \frac{\theta S_Q}{(1-\theta)S_F + \theta S_Q},$$
> where $S_F = \mathbb{E}_ F (1/w)$ and
> $S_Q = \mathbb{E} _ Q (1/w)$  are the expected inverse scores for inliers and outliers, respectively. The condition $\theta' < \theta$ holds if and only if $S_Q < S_F$.
>
> **(3) Justification of $S_Q < S_F$ (Proposition 1 and Geometric Separation):** The core of the proof lies in justifying that outliers have smaller expected weights. This is established in the new Proposition 1 (Appendix B), leveraging the strong RKHS regularization ($\lambda \\| f \\| _ K^2$). We show that fitting unstructured outliers would require a function with an excessively large RKHS norm. The regularized optimization naturally assigns larger residuals $w(z_{out},\alpha)$ to outliers, leading to $1/w(z_{out},\alpha)$ being smaller, and thus guaranteeing $S_Q < S_F$.
>
> **(4) Non-Vacuous Generalization Bound (Theorem 8 in the revised manuscript):** With $\theta^{\prime}$ rigorously bounded away from $\theta$, the term $48M^2\theta^{\prime}$ in Theorem 8 (Theorem 7 in the original manuscript) is no longer vacuous. It explicitly quantifies how the generalization error improves as the separability between inliers and outliers increases .
>
> **(5) Limit distribution of Theorem 1:**  New Theorem 2 explicitly states the exact form of the stationary distribution $\mathcal{P}'$, proving that it is proportional to $1/w(\cdot, \hat{\alpha})$, ensuring that the limit distribution is no longer just implied by ergodicity but is rigorously identified upfront.
>
> **(6) Empirical Verification and Visualization:** Supplemented by empirical verification in Appendix E.3 (Figure 6), directly address your concerns by providing the missing theoretical identification and justification.

---

> ### Author Response · Authors · 2025-11-29
> **Response to Reviewer ikcK Part 6**
>
> ## RW 10
> **Comment 10:**  Proof validity issues include the unjustified convergence rate $O_ p\left(\ln n / n^2\right)$ in Theorem 4 under contamination, and the impractical simultaneous conditions $\lambda=o(1)$ and $(-\ln \lambda)^{1 / 2} / \omega \sim n^{1 /(2 m+1)}$ in Theorems 5-6.
>
> **Response:**
>
> Thanks a lot. Due to the addition of a new theorem in the revised paper, the numbering of subsequent theorems has been updated accordingly. Theorems referenced in the comments (Theorems 4, 5, and 6) now correspond to Theorems 5, 6, and 7 in the revised version.
>
> - **On the convergence rate $O_p(\ln n / n^2)$ in Theorem 4**
>
> 	The rate $O_p(\ln n / n^2)$ is mathematically justified and arises from the following considerations:
>
> 	*   **Target of Convergence:** Theorem 5 establishes the Functional Bahadur Representation (FBR) of the estimator $f_ {\mathbb{S},\lambda}$ with respect to the minimizer of the generalization risk, $f _ {\mathcal{P}^{\prime}}=\arg \min _ {f \in \mathcal{H}_ \omega} \mathcal{R}_ {\mathcal{P}^{\prime}}(f)$, rather than the oracle $f_0$. While contamination introduces bias (reflected in $\\|f_{\mathcal{P}'}-f_0\\|$), it does not affect the stochastic convergence rate of the estimator toward its target $f_{\mathcal{P}'}$.
>
> 	*   **Handling Dependence：** As shown in the proof of Theorem 5, we use the u.e.M.c. property (Condition 1). By applying the Central Limit Theorem for $\phi$-mixing sequences [1] and the covariance inequality [2], the variance of the empirical process scales as $O(1/n)$, consistent with the i.i.d. setting.
>
>     *   **Source of the Rate：** The term $O_p ( \ln n / n^2 )$ bounds the quadratic remainder in the linearization. This fast rate is a specific consequence of using the squared-error loss combined with the periodic Gaussian kernel. Since the eigenvalues of this kernel decay exponentially ($\lambda_k \sim e^{-\tau k^2}$), the tail sum of the spectrum converges rapidly, leading to this bound (see Lemma 1 in [3]).
>
> -  **On the compatibility of conditions in Theorems 4-6**
>
> 	 The conditions $\lambda=o(1)$ and $(-\ln \lambda)^{1/2}/\omega \sim n^{1/(2m+1)}$ are compatible and theoretically necessary, and have been used in [3].
>
> 	*   **Existence:** A regularization parameter satisfying both conditions exists. For instance, we may take an exponentially decaying sequence:  $$ \lambda_n = \exp\left( - C n^{\frac{2}{2m+1}} \right)$$ for some constant  $ C > 0$.
>     Clearly, $\lim_{n\to\infty} \lambda_n = 0$, satisfying $\lambda=o(1)$. Substituting this into the second condition yields $(-\ln \lambda_n)^{1/2} \propto n^{\frac{1}{2m+1}}$, which matches the required scaling.
>
> 	*   **Theoretical Necessity:**  This specific scaling is essential in our theoretical framework, as we employ an infinitely smooth Gaussian kernel to approximate a target function belonging to a Sobolev space of finite order $m$. To balance bias and variance in this mismatched setting,  $\lambda$ must decay exponentially rather than polynomially.
>
> **References:**
>
> [1]. Galin L Jones. On the markov chain central limit theorem. Probability Surveys, pp. 299–320, 2004.
>
> [2]. Paul Doukhan. Mixing. In Mixing: Properties and Examples, pp. 15–23. Springer, 1995.
>
> [3]. Xianli Zeng and Yingcun Xia. Asymptotic distribution for regression in a symmetric periodic gaussian kernel hilbert space. Statistica Sinica, 29(2):1007–1024, 2019.
>
> ## RW 11
> **Comment 11:**  Notational inconsistencies (e.g., $H_\omega(s, t)$ vs. $K_\omega(s, t)$ in Appendix B) and ambiguous definitions (e.g., initial $H(X)$ specification) further reduce clarity.
>
> **Response:**
>
>  We have carefully proofread the manuscript to improve rigorousness:
> - **Notational Consistency:**
>
> 	We sincerely thank you for this careful observation. To address the concerns regarding notational inconsistency (e.g., $H_\omega$ vs. $K_\omega$) and ambiguous definitions, we have conducted a thorough revision of the entire manuscript. The periodic Gaussian kernel function is now consistently denoted as $K_\omega^0 (t, s)$ in Appendix B.2. The symmetric periodic Gaussian kernel is now consistently represented as $H_\omega(t,s)$ in Section 4.2 and Appendix B.2.  Also, the expectation $E(\cdot)$ has been consistently denoted as $\mathbb{E}(\cdot)$ throughout this paper. The acceptance probability $p$, used as the dimension of covariates, in the original paper has been changed as $\pi_\alpha$ in Algorithm of the revised manuscript.
>
> -  **Clarification of $\mathcal{H}(\mathbb{X})$:**
>
> 	The initial definition of the space $\mathcal{H}(\mathbb{X})$ has been made more precise in Section 2.1 to ensure clarity and avoid any potential confusion.  $\mathcal{H}(\mathbb{X})$  is defined as the general ambient space of continuous functions, explicitly distinguishing it from the specific RKHS structures ( $ \mathcal{H}_ K $  and  $\mathcal{H}_\omega$ ) introduced in subsequent sections for estimation.

---

### Official Review · Reviewer_SVvk · 2025-11-01

**Soundness:** 2
**Presentation:** 3
**Contribution:** 3
**Rating:** 6
**Confidence:** 3

**Summary:**

This paper proposes KRMS, a robust subsampling framework designed for non-parametric regression with contaminated data. Specifically, KRMS introduces a residual kernel-norm score, which operates in the reproducing kernel Hilbert space and effectively identifies outliers. Moreover, this paper provides theoretical guarantees for the KRMS estimator, establishing its consistency, asymptotic normality, and generalization bounds.

**Strengths:**

- The presentation is clear and well-organized.
- The paper presents comprehensive theoretical analyses for the proposal.
- The paper provides both simulated and real-world experiments, demonstrating superior performance.

**Weaknesses:**

- While the theoretical analysis is solid and thorough, the methodological contribution appears limited, relying mainly on the kernel-trick-based residual score.
- Introducing the residual score into kernel space is an interesting idea. Nevertheless, it would be much better if the paper included an empirical ablation study comparing the kernel-trick version with its linear-space counterpart.

**Questions:**

Please refer to the weaknesses above.

---

> ### Author Response · Authors · 2025-11-22
> **Response to Reviewer SVvk**
>
> ## RW 1（Response to Weakness 1）
>
> We thank the reviewer for this insightful comment. We agree that the residual kernel-norm score $w(\tilde{z}_i, \alpha)$ is algebraically concise. However, our primary methodological contribution lies not in the score itself, but in the novel theoretical and computational framework that integrates it to solve robust nonparametric estimation. This framework introduces three key innovations:
>
> -  **A Nonparametric Outlier Score**
>
> We derive the first residual score explicitly designed for RKHS models. Existing robust subsampling methods (e.g., based on    gradient) [1] are typically designed for finite-dimensional parametric models. In the nonparametric RKHS setting, the "score" of a data point is implicit and depends on the kernel geometry. Our proposed score $w(\tilde{z}_i, \alpha)$ explicitly normalizes the residual by the kernel induced similarity measure $\sqrt{\sum K(\tilde{\mathbf{x}}_j, \tilde{\mathbf{x}}_i)^2}$. This term acts as a localized density estimator in the high-dimensional feature space, enabling the identification of outliers that may be indistinguishable in the original input space but are separable only in the high-dimensional RKHS.
>
> -  **A Provably Robust Subsampling Mechanism**
>
> A key challenge in robust MCMC subsampling is ensuring the chain converges efficiently to the target distribution. The construction of $w(\tilde{z}_ i, \alpha)$ is tailored to satisfy the minorization condition required for Uniform Ergodicity (as proven in Theorem 1). Unlike direct importance sampling, which may suffer from instability due to weight variance, embedding this score into the Metropolis-Hastings acceptance ratio, $\min\\{1, w_{old}/w_{new}\\}$, creates a theoretically grounded filter. This construction is specifically tailored to guarantee the chain's uniform ergodicity (Theorem 1 in the revised manuscript), ensuring fast convergence to a cleaner stationary distribution—a known challenge in robust MCMC.
>
> -  **Computational Feasibility for Nonparametrics**
>
> The entire framework offers a computationally efficient and theoretically rigorous alternative to full kernel regression, maintaining statistical power against contamination while scaling practically for large datasets.
>
> Therefore, the contribution is the integrated system: a theoretically justified score enabling a provably robust and computationally feasible nonparametric subsampling algorithm.
>
>
> **Reference:**
> [1]. Kaili Jing. Joint feature screening and subsampling in analysis of massive data. PhD thesis, University of Ottawa, 2023.

---

> ### Author Response · Authors · 2025-11-22
> **Response to Reviewer SVvk Part 2**
>
> ## RW 2
>
> We thank you for this excellent suggestion. To directly demonstrate the value of the kernel method, we have conducted a clear ablation study by comparing our proposed KRMS-KLSR with a linear baseline, KRMS-Linear, in Experiment 2 (Nonlinear setting). The results are presented in the revised manuscript (Tables 4–6, Figures 1 and 2 in Appendix D.2).
>
> Following the recommendation, we designed KRMS-Linear to apply the identical residual-based subsampling logic as KRMS-KLSR, with the sole difference being its use of a linear kernel $K(\tilde{\boldsymbol{x}}_j,\tilde{\boldsymbol{x}}_i)=\tilde{\boldsymbol{x}}_j^{\top}\tilde{\boldsymbol{x}}_i$
>   within the Euclidean space to compute $w(\tilde{z},\alpha)$
>  in Equation (3). A summary of the key results (AMSE, SD, and PSR) is provided in Table R1 below, with comprehensive results for contamination schemes M2 and M3, along with further visualizations, available in Tables 4-6 in Appendix D.2.
>
> As shown in Table R1, KRMS-Linear fails severely when the underlying function $f_0$ is nonlinear. For instance, at
>  $\theta=0.4$,  it achieves a PSR of only 16.69%. This confirms the model misspecification bias hypothesis: the linear estimator cannot capture the nonlinear trend, resulting in large residuals for both contaminated and uncontaminated data points. Consequently, the residual-based score loses its ability to distinguish true outliers from model-induced errors.
>
> Visual evidence further underscores this contrast. Due to its inherent linear constraints, the fitted curve of KRMS-Linear is severely distorted by outliers. In contrast, our KRMS-KLSR method successfully captures the true nonlinear data structure, resists outlier interference, and recovers a smooth curve that aligns well with the underlying curve.
>
> Furthermore, analysis of the sampling metric $\log ( w ) $ reveals that in nonlinear settings, our method produces a more distinct separation between inliers and outliers. This validates that the kernel-induced residual score more effectively discriminates anomalies. Conversely, the metric distribution of KRMS-Linear shows significant overlap, preventing its subsampling process from effectively filtering out contaminated points.
>
> In summary, this ablation study confirms two key advantages of our approach:
>
> 1. Effective Nonlinear Feature Capture: Our method excels at captureing complex nonlinear patterns, making data and outliers that are linearly inseparable in the original space separable within the RKHS feature space.
> 2. Enhanced Robustness: By more precisely assessing local data structures in the high-dimensional feature space, quantified by the kernel term $\sqrt{\sum K\left(\tilde{x} _ i, \tilde{x} _ j\right)^2}$, our method achieves more reliable outlier identification and suppression, leading to superior statistical performance on complex datasets compared to linear baselines.
>
> **Table R1: Performance comparison of KRMS-KLSR and KRMS-Linear subsampling method for
> corrupted mechanism M1 in Experiment 2**
> |   | | | $n=500$  | |   | $n=1000$|  |   | $n=1500$| |
> | :--- | :--- | :--- | :--- | :--- | :--- | :--- | :--- | :--- | :--- | :--- |
> | $\theta$ | Method |  AMSE | SD | PSR | AMSE | SD | PSR | AMSE | SD | PSR |
> | **0.1** | **KRMS-KLSR** | **1.137** | **0.041** | **100.00%** | **1.098** | **0.036** | **100.00%** | **1.087** | **0.035** | **100.00%** |
> | | LRMS-Linear | 25.631 | 0.964 | 75.58% | 25.245 | 0.900 | 78.07% | 25.055 | 0.711 | 79.82% |
> | **0.2** | **KRMS-KLSR** | **1.153** | **0.044** | **100.00%** | **1.099** | **0.038** | **100.00%** | **1.091** | **0.035** | **100.00%** |
> | | LRMS-Linear | 29.833 | 0.548 | 41.96% | 29.746 | 0.547 | 47.15% | 29.708 | 0.472 | 51.71% |
> | **0.3** |  **KRMS-KLSR** | **1.145** | **0.050** | **100.00%** | **1.104** | **0.041** | **100.00%** | **1.101** | **0.038** | **100.00%** |
> | | LRMS-Linear | 30.734 | 0.499 | 25.19% | 30.712 | 0.450 | 30.98% | 30.732 | 0.475 | 36.01% |
> | **0.4** | **KRMS-KLSR** | **1.149** | **0.045** | **100.00%** | **1.104** | **0.042** | **100.00%** | **1.106** | **0.042** | **100.00%** |
> | | LRMS-Linear | 31.086 | 0.518 | 16.69% | 31.087 | 0.483 | 22.57% | 31.084 | 0.475 | 27.13% |

---

### Official Review · Reviewer_rXQQ · 2025-11-01

**Soundness:** 2
**Presentation:** 2
**Contribution:** 1
**Rating:** 2
**Confidence:** 4

**Summary:**

This paper proposes KRMS, a Metropolis–Hastings (MH) subsampling scheme for kernel ridge regression under Huber contamination. The authors prove uniform ergodicity of the induced chain (finite state space), establish consistency and pointwise asymptotic normality in a symmetric periodic Gaussian RKHS, and derive a generalization bound for u.e.M.c. samples. Simulations and two “real‑data” illustrations are reported.

**Strengths:**

The target problem is an important regime: robust, scalable learning in RKHS under contamination.

The paper provides clear, intuitive heuristic: prefer points with small residual relative to their “kernel similarity magnitude”.

**Weaknesses:**

1. The main weaknesses lie in the “cleaner” stationary distribution $\mathcal{P}'$, which lacks clear explanation. And the paper states \mathcal{P}' has less contamination $\theta'<\theta$. However, this statement is not proved.  The chain with acceptance $\min \\{1,w(z_t)/w(z^*) \\}$ has stationary distribution proportional to $1/w(\cdot,\alpha)$ (with $\alpha$ frozen), not obviously to a mixture $1-\theta')F+\theta'Q$ with $\theta'<\theta)$ The paper asserts convergence to a distribution with reduced contamination yet provides no identification of the MH target beyond ergodicity, nor any argument that $1/w$ upweights uncontaminated draws in the sense of a reduced mixture weight. Theorem 1 (irreducible+aperiodic on a finite set ⇒ uniform ergodicity) does **not** characterize the limit distribution. Consequently, Theorem 7’s bound—with a leading $48M^2\theta'$ term—remains vacuous.

2. In Algorithm 1, the paper sets $\alpha^{(1)} = \alpha^{(0)} + 0.2$. Please justify the logic here.

3. In Algorithm 1, to evaluate $w(z,\alpha)$, the paper needs to compute the weighted sum of $n$ $K(x_i,x_j)$, which results a complexity of $n$. This contradicts the claim of the complexity $O(T_0(n_0^2 p+n_0^3))$.

4. It is hard to comprehend the numbers in Tables 1-6. It is better to emphasize key numbers.

**Questions:**

na

---

> ### Author Response · Authors · 2025-11-22
> **Response to Reviewer rXQQ**
>
> ## RW 1 (Response to Weaknesse 1 )
> Thank you for your concern.  In the revised manuscript, we have addressed this core weakness with a comprehensive theoretical analysis, which we summarize below.
>
> **(1) Exact Characterization of $\mathcal{P}'$:**  As you pointed out, the M-H chain with acceptance probability $\min\\{1, w(z_t)/w(z^*)\\}$ (with frozen $\alpha$) indeed converges to a stationary distribution $\mathcal{P}'$ proportional to $1/w(.,\alpha)$. In the new Theorem 2 (Section 4.1), we have clearly presented an explanation of $\mathcal{P}'$. In the new Lemma 1 (Appendix B), we explicitly derive the density of this distribution: $$p'(\tilde{z})=\frac{1}{Z}\frac{p(\tilde{z})}{w(\tilde{z},\alpha)}, Z=\int\frac{p(z)}{w(z,\alpha)}dz,$$
> which confirms that the sampling probability is inversely proportional to the residual kernel-norm score $w(z, \alpha)$.
>
> **(2) Rigorous Proof of $\theta' < \theta$:**  Building on this exact form, the new Theorem 2 (Section 4) rigorously derives the effective contamination rate $\theta'$ after subsampling. We prove that for the mixture $P = (1-\theta) F+\theta Q$, the stationary distribution $P'$ corresponds to a new mixture with a contamination rate given by
> $$\theta' = \frac{\theta S_Q}{(1-\theta)S_F + \theta S_Q},$$
> where $S_F = \mathbb{E}_ F (1/w)$ and
> $S_Q = \mathbb{E} _ Q (1/w)$  are the expected inverse scores for inliers and outliers, respectively. The condition $\theta' < \theta$ holds if and only if $S_Q < S_F$.
>
> **(3) Justification of $S_Q < S_F$ (Proposition 1 and Geometric Separation):** The core of the proof lies in justifying that outliers have smaller expected weights. This is established in the new Proposition 1 (Appendix B), leveraging the strong RKHS regularization ($\lambda \\| f \\| _ K^2$). We show that fitting unstructured outliers would require a function with an excessively large RKHS norm. The regularized optimization naturally assigns larger residuals $w(z_{out},\alpha)$ to outliers, leading to $1/w(z_{out},\alpha)$ being smaller, and thus guaranteeing $S_Q < S_F$.
>
> **(4) Non-Vacuous Generalization Bound (Theorem 8 in the revised manuscript):** With $\theta^{\prime}$ rigorously bounded away from $\theta$, the term $48M^2\theta^{\prime}$ in Theorem 8 (Theorem 7 in the original manuscript) is no longer vacuous. It explicitly quantifies how the generalization error improves as the separability between inliers and outliers increases .
>
> **(5) Limit distribution of Theorem 1:**  New Theorem 2 explicitly states the exact form of the stationary distribution $\mathcal{P}'$, proving that it is proportional to $1/w(\cdot, \hat{\alpha})$, ensuring that the limit distribution is no longer just implied by ergodicity but is rigorously identified upfront.
>
> **(6) Empirical Verification and Visualization:** Supplemented by empirical verification in Appendix E.3 (Figure 6), directly address your concerns by providing the missing theoretical identification and justification.
>
> ## RW 2
> Thank you for catching this. The step $\alpha^{(1)} = \alpha^{(0)} + 0.2$ was an arbitrary device to prevent immediate triggering of the stopping criterion $|\alpha^{(1)} - \alpha^{(0)}| < \xi_0$. We agree it was incorrect to state it as an algorithmic update. It has been removed from Algorithm 1. The algorithm now simply initializes with $\alpha^{(0)}$, allowing the stopping criterion to function naturally. We apologize for the confusion.
>
> ## RW 3
> Thank you for this insightful observation. Indeed, the evaluation of $w(\tilde{z}, \alpha)$ requires a computation involving the entire dataset of size $n$ for its denominator. The complexity of this step is $\mathcal{O}(n p)$. However, the sparsity of  $\alpha$ plays a crucial role in the numerator, whose cost is only $\mathcal{O}(n_0 p)$, as only $n_0$ non-zero coefficients correspond to the subsampled points. Therefore, the cost per evaluation of $w(\tilde{z}, \alpha)$ is $\mathcal{O}((n + n_0) p) = \mathcal{O}(n p)$, since $n$ dominates $n_0$. We have updated Remark 1(v) accordingly to reflect the total complexity of $\mathcal{O}(T_0 (n_0^3 + n_0 n p))$. This revision clarifies the linear dependence on $n$. Nevertheless, for $n_0 \ll n$, our method maintains a significant computational advantage over the $\mathcal{O}(n^3 + n^2 p)$ complexity of standard kernel regression.
>
> ## RW 4
> Thank you for your good suggestion. To emphasize the key results and enhance readability, we have revised Tables 1–10 by boldfacing the best-performing entry in each column. This revision makes it easier to identify the superiority of the proposed method at a glance.

---

> > ### Comment · Reviewer_rXQQ · 2025-11-23
> >
> > I appreciate the authors’ efforts to address all my comments during the short rebuttal period, and I have accordingly raised my score. However, the updated paper differs substantially from the original version, and I cannot thoroughly verify all changes within the limited remaining rebuttal period, which also coincides with the Thanksgiving break. Therefore, I believe that another round of revision is necessary.

---

### Author Response · Authors · 2025-12-03
**Summary of responses**

Dear Area Chair,

We thank the reviewers and chairs for their valuable feedback and constructive discussions. Following the Area Chair reassignment, we provide a brief summary of the rebuttal outcomes and revision status. The rebuttal led to productive exchanges, including one reviewer raising their score to 4. All updates and reviewers' explanations remain available on OpenReview.

**1. Core Contribution**

We proposed a kernel-based Markov subsampling method for large-scale nonparametric regression with contaminated data. Our key innovation was to define the subsampling probability as the ratio of the absolute residual to the kernel norm of the covariates. This design dynamically down-weights outliers while preserving clean data, effectively addressing the limitations of standard subsampling approaches (e.g., leverage scores) in contamination settings.

Our core contributions include: (1) Theoretically Grounded Robustness: We provide, for the first time, probabilistic convergence guarantees showing that the Markov subsampling chain selects clean data with high probability even under contamination—offering a theoretical foundation for sampling-based robust regression. (2) Computational and Statistical Efficiency:
Through an iterative reweighting scheme, KRMS maintains near-linear time complexity while significantly improving estimation accuracy and stability, achieving an effective balance between computational scalability and statistical performance. (3) Low Sensitivity to Hyperparameters: Empirical results demonstrate that the algorithm is highly stable across a wide range of bandwidth and burn-in period choices, substantially reducing the burden of parameter tuning in practice. (4) Flexible and Extensible Kernel Framework: The method is compatible with a variety of kernel functions (e.g., Gaussian, Laplacian), preserving computational efficiency while enhancing adaptability to different data characteristics.

**2. Reviewer Consensus**

The post-rebuttal discussions indicate that technical concerns have been resolved. However, overall system scores may appear inconsistent due to a platform-wide reversion of the scoring mechanism.

*   **Reviewer rXQQ (Score: 2 $\to$ 4):** Acknowledged that "authors have addressed all my comments" and raised the score.
	-  **Context:** Reviewer rXQQ remarked on “substantial changes.” We clarified that these additions: Theorem 2 and Proposition 1 were introduced in direct response to the reviewer’s earlier request for theoretical justification of the stationary distribution and contamination reduction. The core algorithm itself remains unchanged.

*   **Reviewer SVvk (Score: 6):** This reviewer recognized the theoretical novelty of the work and commended its clear presentation.
*   **Reviewer ikcK (Score: 4):** The reviewer’s primary concerns involved an initialization bug and the absence of a sensitivity analysis.
	-  **Resolution:** We have corrected the initialization step in Algorithm 1 and added the requested sensitivity analysis in Appendix E.
*   **Reviewer mooN (Score: 6):** This reviewer maintains an acceptance recommendation and explicitly expressed satisfaction with the newly added “Clean Data” baseline.

**3. Key Revisions**

In our rebuttal, we addressed each point in detail, and have carefully incorporated the suggested changes in the revision including:

-  **Theory (Addressing Reviewers rXQQ and mooN ):**

	-  **Theorem 2:** Derivation of the exact stationary distribution.  Section 4, page 6.
	-  **Proposition 1:** Proof that geometric separation in RKHS guarantees reduced effective contamination ($\theta' < \theta$).  Appendix B.1, page 13.

- **Analysis of the Computational Complexity(Addressing Reviewers rXQQ and mooN ):**
	-  We analyzed the time and space complexities  in Remark 1(v).  Section 3, page 6.

- **Robustness & Baselines (Addressing Reviewer SVvk , Reviewer ikcK and Reviewer mooN):**
	-  **Algorithm 1:** Corrected initialization step to remove the heuristic flagged by Reviewer ikcK. Section3, page 6.
	-  **Linear Space Baselines:** Added KRMS-Linear method in Table 4-Table 6. Appendix D.2, page 32-34.
	-  **Robust Nonparametric Regression Baseline and Scalability to Large-Scale Data:** Added Table 7 to demonstrate scalability to larger datasets or higher dimensional settings. Appendix D.2, page 36.
	-  **Clean Data Baseline:** Added Table R3 in the rebuttal to Reviewer mooN to demonstrate negligible efficiency loss on clean data.

- **Convergence Analysis and Parameter Sensitivity Analysis (Addressing Reviewer ikcK):**
	-  To analysis algorithm convergence and sensitivity to subsample size, burn-in period, bandwidth and kernel selection, we have incorporated new experimental results into Appendix E, page 39-42.

**4. Verification Guide**
*   **New Theory:** Section 4.1 & Appendix B (Blue Text)
*   **Corrected Algorithm:** Algorithm 1 & Remark 1.
*   **New Experiments:** Appendix D.2 & E.

Best Regards,

The Authors

---

### Note · Authors · 2026-02-03

**Comment:**

Author decision after rejection

**Withdrawal Confirmation:**

I have read and agree with the venue's withdrawal policy on behalf of myself and my co-authors.

---

### Meta-Review · Area_Chair_3noC · 2026-01-06

**Summary:**

This paper proposes a kernel-based methodology for Markov subsampling and demonstrates its robustness to contaminated data. Because the exchanges between reviewers and authors took place during the rebuttal phase (prior to the discussion period, the logs of which are no longer available), a certain level of agreement appears to have been reached among the reviewers.

Reviewer rXQQ pointed out that the theoretical characterization was insufficient and identified several heuristic elements and inconsistencies. In response, the authors revised the theory by deriving sufficient conditions and modifying the algorithm accordingly. However, the reviewer maintained a negative stance, arguing that the revisions were substantial enough to warrant a fresh round of review, and did not change their overall assessment.

Reviewer SVvk expressed the view that the methodological novelty was limited and that additional comparative experiments would strengthen the paper. The authors provided supplementary explanations regarding both novelty and experimental comparisons. While the clarification of novelty was not exceptionally explicit, it was not unsatisfactory. With respect to the experiments, the authors presented a reasonable number of additional results to support their claims. Given that the reviewer did not initially raise strong objections, these responses were likely satisfactory.

Reviewer ikck raised concerns about multiple aspects of the work, including algorithmic design choices, guidelines for parameter selection, constraints imposed by the kernel, insufficient experimental validation, and several theoretical issues. In response, the authors partially revised the algorithmic design, introduced new experimental settings and additional experiments, and substantially revised the theoretical analysis. Some of the reviewer’s questions may have been overly stringent; nevertheless, the authors appear to have made a best-faith effort to address them. However, given the sheer number and scope of the revisions, it remains unclear whether the reviewer was fully convinced by all of these changes. After such extensive modifications, a renewed review process—allowing all reviewers sufficient time to assess the revised manuscript—would be more appropriate.

Reviewer mooN pointed out shortcomings in baseline comparisons, insufficient evaluation in uncontaminated settings, and a lack of quantitative analysis. The authors addressed these issues by conducting additional experiments, after which the reviewer chose to maintain their original evaluation (a weak accept).

Overall, while the reviewers raised numerous concerns, the authors responded very energetically and carried out extensive revisions. These efforts succeeded in convincing some of the reviewers; however, the volume and depth of the changes also make it unclear whether all reviewers were able to fully assess and accept the revised work. Given the extent of the modifications, it would be reasonable to conclude that the paper should undergo another round of review, with sufficient time allocated for all reviewers to carefully evaluate the updated manuscript.

**Reviewer Concerns:**

See above.

**Reviewer Scores:**

See above.

---

### Decision · Program_Chairs · 2026-01-26

Reject